# Energy transformation cost for the Japanese mid-century strategy

Shinichiro Fujimori [1,2,3]*, Ken Oshiro[1], Hiroto Shiraki [4] & Tomoko Hasegawa [2,3,5]

The costs of climate change mitigation policy are one of the main concerns in decarbonizing the economy. The macroeconomic and sectoral implications of policy interventions are typically estimated by economic models, which tend be higher than the additional energy system costs projected by energy system models. Here, we show the extent to which policy costs can be lower than those from conventional economic models by integrating an energy system and an economic model, applying Japan's mid-century climate mitigation target. The GDP losses estimated with the integrated model were significantly lower than those in the conventional economic model by more than 50% in 2050. The representation of industry and service sector energy consumption is the main factor causing these differences. Our findings suggest that this type of integrated approach would contribute new insights by providing improved estimates of GDP losses, which can be critical information for setting national climate policies.

[1] Department of Environmental Engineering, Kyoto University, C1-3 361, Kyotodaigaku Katsura, Nishikyoku, Kyoto city, Japan. [2] Center for Social and Environmental Systems Research, National Institute for Environmental Studies (NIES), 16–2 Onogawa, Tsukuba, Ibaraki 305–8506, Japan. [3] International Institute for Applied System Analysis (IIASA), Schlossplatz 1, A-2361 Laxenburg, Austria. [4] School of Environmental Science, The University of Shiga Prefecture, Hikone, Japan. [5] Department of Civil and Environmental Engineering, College of Science and Engineering, Ritsumeikan University, Kyoto, Japan. *email: sfujimori@athehost.env.kyoto-u.ac.jp

Climate change mitigation is one of the greatest societal challenges facing most countries as reduction of energy-related $CO_2$ emissions is key to reducing greenhouse gas (GHG) emissions. In 2015, more than 190 countries reached the Paris Agreement (PA)[1] and each country submitted their own Nationally Determined Contribution (NDC). Along with those targets, countries were also asked to engage in long-term planning, known as a mid-century strategy[2,3]. Under the long-term global goal in the PA of keeping the global mean temperature increase well below 2 °C compared with the pre-industrial level, the net $CO_2$ emissions in this mid-century must be close to neutral according to numerous studies carried out using Integrated Assessment Models (IAMs)[4].

Macroeconomic costs of climate change mitigation is a great concern for climate policy settings[5]. The Intergovernmental Panel on Climate Change (IPCC) fifth assessment report summarises climate mitigation costs, and GDP or consumption losses in 2050 are around 2–6%[4] to achieve the abovementioned 2 °C goal. There are multiple ways to interpret these numbers. It may be too expensive to pay for climate change prevention that delays GDP growth for a couple of years or low enough for avoiding widespread climate change impacts and irreversible risks associated with catastrophic events. To address macroeconomic mitigation costs, IAMs normally represent GHG emissions reduction costs either through an energy system model or an economic model, often termed bottom-up and top-down models, respectively. Although there are other ways to classify the IAMs, in this paper, we define economic model as the model that includes multi-sectoral CGE model within the IAM framework, and energy system model as the model that does not. Note that a power-dispatch model is also used in this study although that is not usually classified as IAMs. There are many global[6–8], and national energy system models[9,10] as well as the economic models[11,12], which are based on multi-sectoral CGE models.

Traditionally, CGE models tend to project higher policy costs than those of energy system models[13] (see also Supplementary Note 1 and Supplementary Table 1). One possible reason for this tendency is that parameters in CGE models are calibrated against a historical period in which it is difficult to decouple economic growth and $CO_2$ emissions. Some argue that aggregated energy system representation is disadvantageous to understanding drastic energy system changes and their macroeconomic implications. Thus, incorporating energy system model information into CGE models may lead lower macroeconomic costs than previously reported.

Integrating CGE and energy system model offers a great advantage in representing the feedbacks inherent across economic and energy systems. For the policy makers, macroeconomic implications including sectoral impacts provided by CGE models is more meaningful than energy system costs alone. To this end, several attempts have been made[14–16], whereas investigators such as Bohringer et al.[17–20] incorporated disaggregated information on power sectors. An extended literature list is shown in Supplementary Table 2 and there are more examples if we include non-multi-sector CGE models[21,22].

At the meantime, drastic energy transformation requires large-scale variable renewable energy penetration. The key issue of the variability in renewable energy is strongly dependent on national- and local-scale grid systems, availability of solar and wind power, battery technology, and other energy sources that can be used to balance demand and supply. Recently, some national modelling studies have addressed these issues[23–25] and integration of a power-dispatch model with an energy system model has been attempted[26]. In IAMs, they are represented to some degree[27,28], which are adequate to provide global-scale energy analyses. However, no studies showed macroeconomic implications of

consistently dealing with energy systems and the stability of power generation.

Here, we describe the macroeconomic implications of climate mitigation policy using an integrated modelling framework wherein an energy system model, Asian-Pacific Integrated Model/ Enduse (AIM/Enduse), and a power-dispatch model, AIM/Power, are inter-linked with the multi-sector economic model (AIM/ CGE). We call this new soft-linking modelling framework an integrated model, which allows us to assess the macroeconomic impacts of climate change mitigation with concrete specification of detailed energy technologies, ensuring a stable power supply with consideration of long-term (seasonal and daily) and short-term (less than hourly) power fluctuations.

The principle of this methodology is based on the concept that energy simulation from the energy system model is more reliable than that from the economic model, as energy supply and demand are technologically represented in detailed in the energy system model. Similarly, the technological representation of power supply in the power-dispatch model is more reliable than that in the energy system model. We overcome the disadvantages of these models by exchanging information and iterating it among models. We begin with the AIM/Enduse run, which provides energy system information to AIM/CGE and AIM/ Power. Then, these two models' outcomes are further fed into AIM/Enduse. Finally, we confirm whether the models reach sufficient convergence for our purposes (see Supplementary Information for more detailed discussion about reaching convergence). See the Methods for indicators exchanged among models. Note that for CGE results, we compare the stand-alone CGE model with the integrated model.

We applied this framework to Japan as a case study. The Japanese government has declared a long-term GHG emissions reduction target of 80% by 2050[29]. As mitigation costs in Japan estimated in previous studies vary significantly across IAMs[30–32], application of this framework would be beneficial for Japan's climate policies to communicate with the stakeholders. We analysed scenarios with and without climate mitigation policy, which are the mitigation and baseline scenarios, respectively.

As results, we found that the macroeconomic costs are not as high as previously reported when energy system information is appropriately reflected in the economic model. The critical determinants of mitigation costs that changed in the newly developed integrated model were identified as the representation of industry and service sectors' energy consumption, which is associated with production functions. These findings may change the general perception of climate change mitigation costs in terms of macroeconomic losses and provide important policy insights.

## Results

**Energy system in Japan's mid-century strategy.** An 80% reduction of GHG emissions requires substantial changes in the energy system compared to the current system or the baseline scenario (Fig. 1a). As a result of Japan's unique socioeconomic circumstances, with a decreasing population and modest economic growth (Supplementary Fig. 1), the overall energy system shows little changes in the future under the baseline scenario. The main changes of the baseline 2050 from the base year are the higher share of coal relative to other fossil fuels, and the decrease in the share of nuclear energy, which reflects the current societal attitude toward nuclear power that limits new construction (Fig. 1b). Regarding $CO_2$ emissions, the baseline level is stable or may even decline over time (Fig. 1d). Meanwhile, the mitigation scenario exhibits large-scale renewable energy penetration, slight energy demand reduction, compositional changes characterised by the use of more carbon-neutral energy sources, and

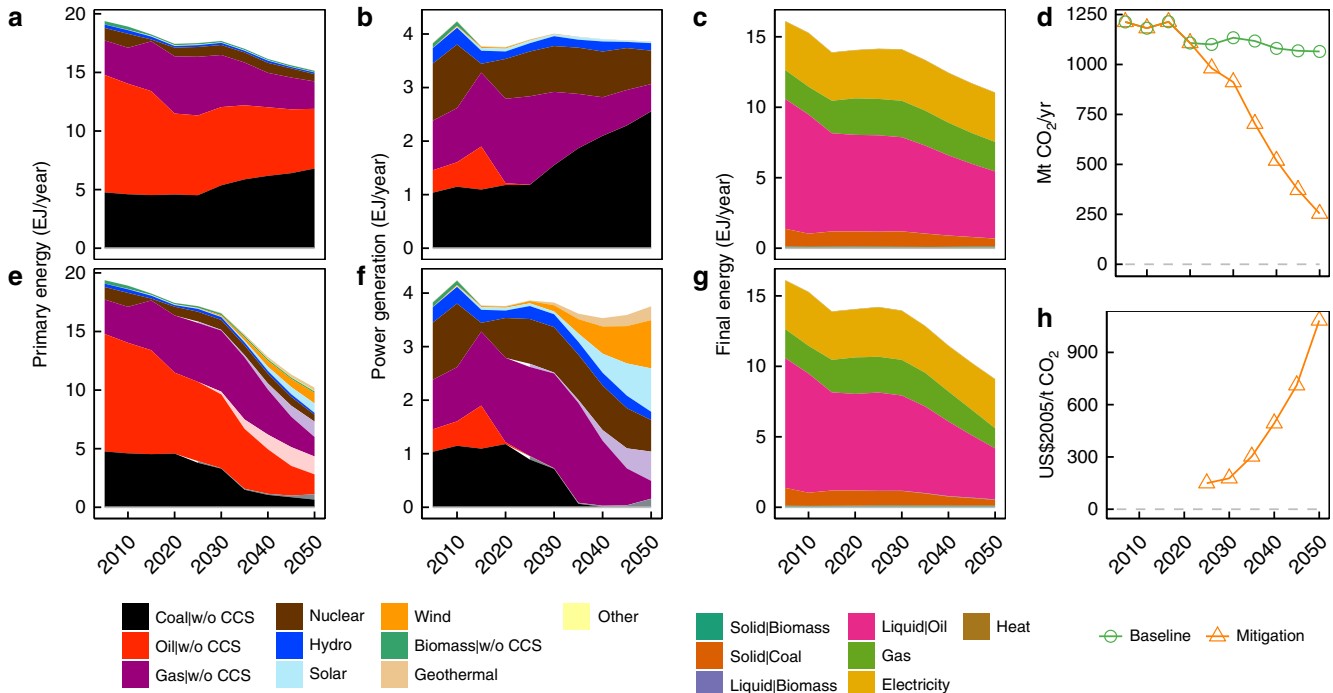

**Fig. 1** Main energy and emissions related variables. Primary energy source (**a**, **e**), power generation (**b**, **f**), final energy demand (**c**, **g**), $CO_2$ emissions (**d**), and carbon price (**h**) projections. **a**, **b**, and **c** show the baseline scenario, whereas **e**, **f**, and **g** show mitigation scenarios. The other in **a** and **e** includes secondary energy net imports

electrification (Fig. 1f). The price of carbon in the mitigation scenario increases over time and reaches ~1000$/tCO$_2$ in 2050, which is due to the low-carbon technological availability in the AIM/Enduse model.

The power system relying heavily on variable renewable energy requires measures to stabilise the power supply system and demand responses. Curtailment in onshore wind increases, particularly after 2020 when variable renewables start to expand (Fig. 2a). Furthermore, when coal-fired power is completely phased out around 2040, offshore wind also exhibits a clear curtailment increase. The battery requirements for short-term fluctuations also increase sharply after 2020, whereas the capacity factor of thermal power plants declines (Fig. 2b, c). We also show the daily electricity supply and demand profiles for selected days in 2050 (Fig. 2d).

**Mitigation costs.** Mitigation costs, as measured by GDP loss rates (hereafter GDP is accounted by the total final consumption), increase over time as emissions reductions become deeper, as illustrated in Fig. 3a. The CGE stand-alone results reach more than 2.5% after 2030, whereas the integrated model is lower, around 1.2% in 2050 (Fig. 3a). The equivalent variation also shows similar trend as GDP losses (Fig. 3b). The additional energy system costs in the AIM/Enduse stand-alone are plotted in the same figure, and are notably similar to the integrated model results (blue lines in Fig. 3a). The mitigation costs under such deep emissions reductions from CGE studies are usually not as low as our estimates (2–6% of GDP losses in 2050)[4]. Once the energy system model's results are reflected in the economic model, the integrated model would be able to estimate similar mitigation costs to those from energy system models.

We further implemented sensitivity scenarios with varying technological availability, which may lead to non-linear energy system responses, to investigate the robustness of our findings. For this purpose, we selected two technological variation scenarios wherein more power stability measures are needed;

namely, without nuclear and without carbon capture and storage (CCS). These results can be interpreted as a simple uncertainty analysis, but they have more meaningful policy implications because the perception of nuclear power in Japan has changed drastically since the Fukushima Daiichi accident, and there is limited geologically appropriate space for CCS on Japanese territory. Figure 3c illustrates the relationship of mitigation costs in the CGE stand-alone and integrated models for this sensitivity analysis. Here, we again see systematically higher costs in the stand-alone model than in the integrated model. Comparison of these integrated model's GDP losses and additional energy system costs derived from AIM/Enduse shows a similar trend to that in Fig. 3d. The overall energy and emissions trends for this sensitivity cases are provided in Supplementary Fig. 2 and Supplementary Fig. 3.

**Mechanism causing the differences in macroeconomic costs.** The central mechanisms for changing the macroeconomic implications are changes in the productivity of primary factors (labour and capital) constituting value-added, which is the GDP measure in production side. This is because the primary factor inputs are constrained exogenously for each year in our economic model[33,34] while the capital and labour inputs change dynamically with population development, and GDP growth. The straight-forward reason that GDP losses are lower in the integrated model than in the stand-alone model is that the parameter assumptions in CGE models differ between the stand-alone model and integrated model. The former relies on the existing literature and the latter on the energy system model outputs. Consequently, the primary factor productivity is higher in the integrated model than in the stand-alone model. Then, the differences in productivity are mainly driven by two things. One is the productivity decreases associated with emissions reductions in energy end-use sectors, such as industry, transport and service sectors (e.g. capital replacement by expensive but energy-efficient ones). The other is sectoral allocation changes in primary factors.

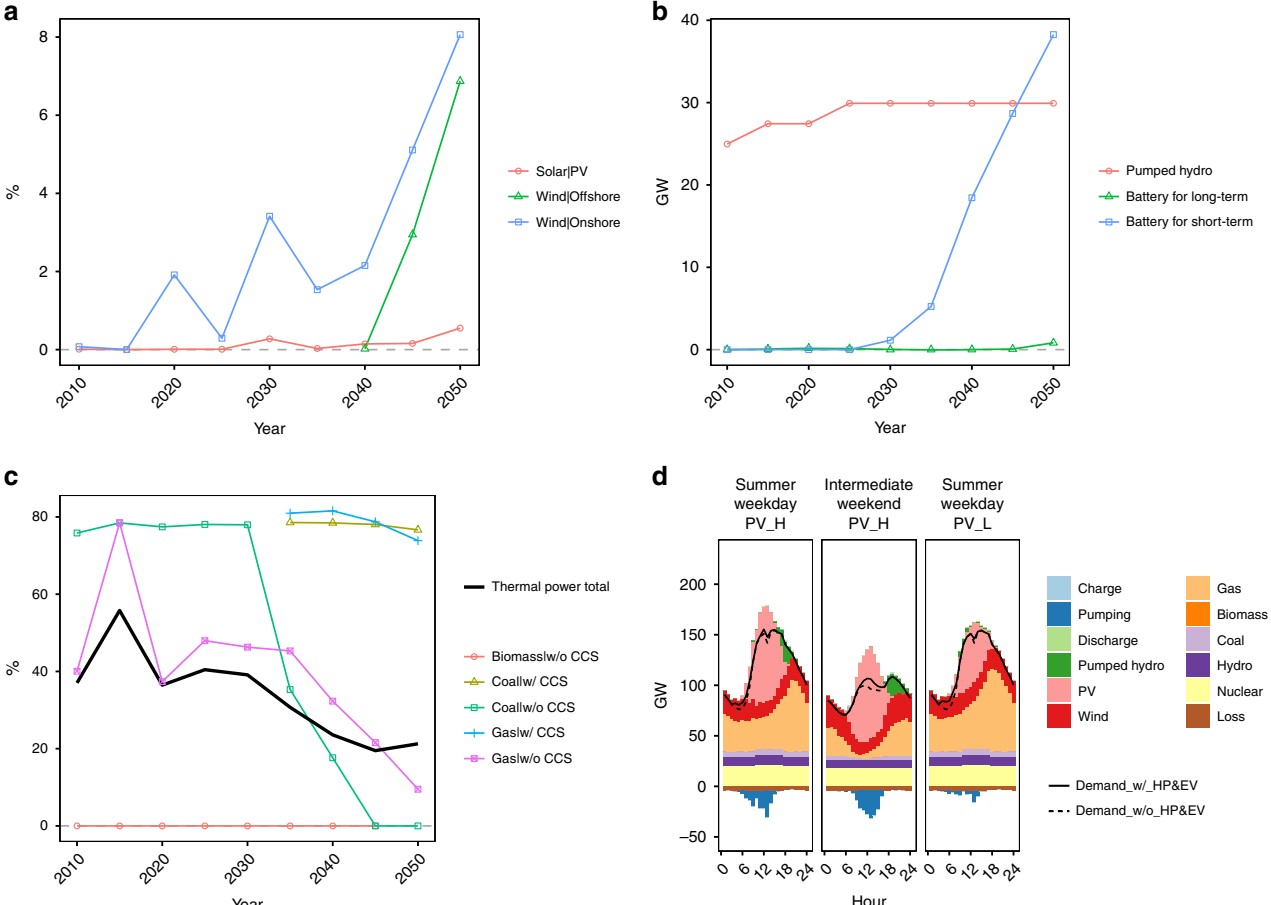

**Fig. 2** Power system reactions to large-scale renewable energy penetration. **a** Wind and solar curtailment rate, which is the unused energy divided by total power generation for each energy source, **b** the installed capacity of technologies to stabilise fluctuations in the electricity supply, **c** the capacity factor of thermal power, and **d** the profiles of electricity demand and supply on selected three typical days (PV_H and PV_L indicate sunny and cloudy days, respectively)

Regarding the first factor, Fig. 4a illustrates the capital input efficiency of major industrial sectors (top 10 industries, which account for 95% of GDP in the base year) in the mitigation scenario compared to the baseline scenario for stand-alone and integrated models in 2050. Here, we define the capital input efficiency as capital input per output for each sector, which is a model outcome. Higher values indicate that additional capital inputs are needed in the mitigation scenario compared to the baseline scenario. In general, the stand-alone model requires larger capital inputs than the integrated model in the mitigation scenario. We can roughly compute the value-added losses associated with these capital productivity losses by multiplying the value-added of each sector Fig. 4b, c). These eventually account for 1.3 percentage points of the total value-added (GDP). Then, the productivity differences between the stand-alone and integrated model are mainly caused by differences in the functional form and parameters particular to the value-added and energy bundle. Here, we use a CES function in which the substitution elasticity, share parameters and future autonomous energy efficiency are defined in the stand-alone model. The integrated model uses a function of the same form, but the additional investment and energy inputs are exogenously given by AIM/Enduse, whereas the CES shift parameters are determined endogenously (sector-wise additional investments are shown in Supplementary Table 3).

The second factor, namely the effect of sectoral primary factor allocation changes, is mainly driven by the power generation sector. The electricity generation in the mitigation scenario compared to the baseline scenario is about 20% higher in the stand-alone model, but almost the same in the integrated model (Fig. 4). There are certainly differences in technological shares between the stand-alone and integrated models, but, in summary, it seems that the difference in total electricity generation between the models is the dominant factor, where the stand-alone model requires additional capital and labour inputs, accounting for 0.4 percentage points of GDP, relative to the integrated model, which relies on the AIM/Enduse outputs (Fig. 4d). With respect to the representation of electricity demand, the total electricity demand is determined by energy consumption in the energy end-use sectors, which are represented by a CES function, as mentioned above. The fuel-wise share is determined using a logit function in both the stand-alone and integrated models. A parameter representing the preferences or technological choices in the logit function is determined endogenously in the integrated model, based on the AIM/Enduse results, whereas they are exogenous parameters in the stand-alone model. We describe the detailed mathematical formation and assumptions in the Supporting Information.

In addition to the two main mechanisms mentioned above, the productivity changes and sectoral shifts in other sectors certainly occur, but are relatively minor. In summary, the differences in GDP changes between the stand-alone and integrated models are

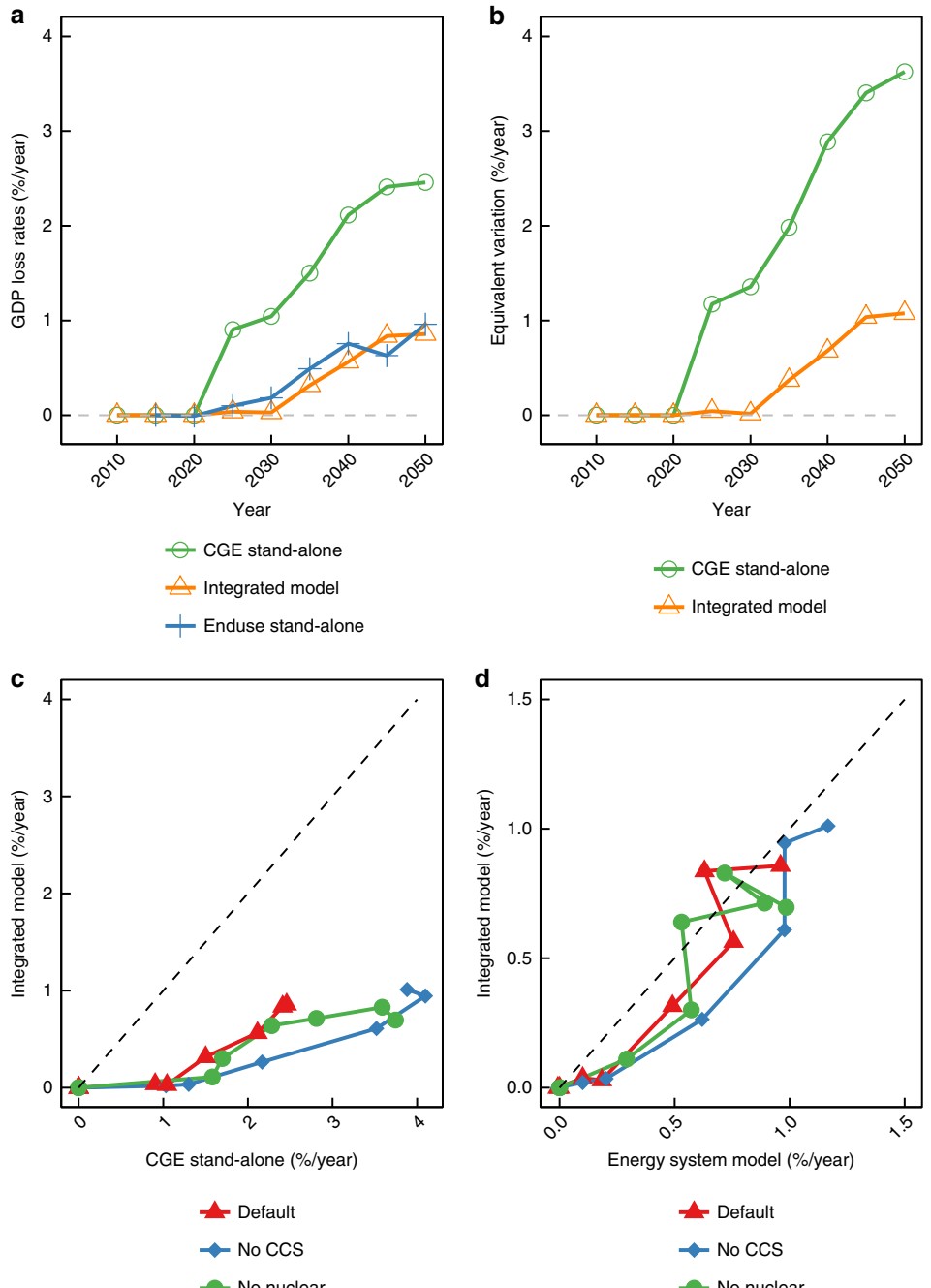

**Fig. 3** Climate change mitigation cost. **a**, **b** Time-series mitigation cost AIM/CGE results are represented as GDP loss rates and equivalent variation change rates relative to baseline scenarios. AIM/Enduse results are expressed as additional energy system costs of GDP relative to baseline scenarios. **c**, **d** 5-year mitigation costs with varying technological availability; **c** illustrates the relationship of GDP losses in the CGE stand-alone and integrated models, and **d** shows GDP losses in the integrated model and additional energy system costs in AIM/Enduse. The energy system model results shown here correspond to Enduse_results1 in Supplementary Fig. 5

explained above, but, generally speaking, many interactions simultaneously occur in the CGE model and sometimes the cause and consequences are not clear.

**Decomposition of mitigation costs and sectoral contributions.** To identify which sectors contribute to GDP losses, the value-added by each sector, as estimated by the economic model, is decomposed into three factors of output changes, value-added productivity (output per value-added), and residuals. Moreover, we compared the outputs of stand-alone CGE and integrated

model runs in Fig. 5. The stand-alone CGE model shows remarkable value-added decreases in the industry (IND) and service sectors (SER) in 2030, whereas the integrated model does not. These trends remained consistent for the year 2050, with the CGE stand-alone model showing large changes in the service sector. This result is consistent with those described in the previous section, wherein the industry and service sector's energy system information, i.e. the representation of production functions in those sectors, are critical factors for differentiating overall GDP losses between the two models. The output decrease in the service sector is the largest element to change the GDP in the

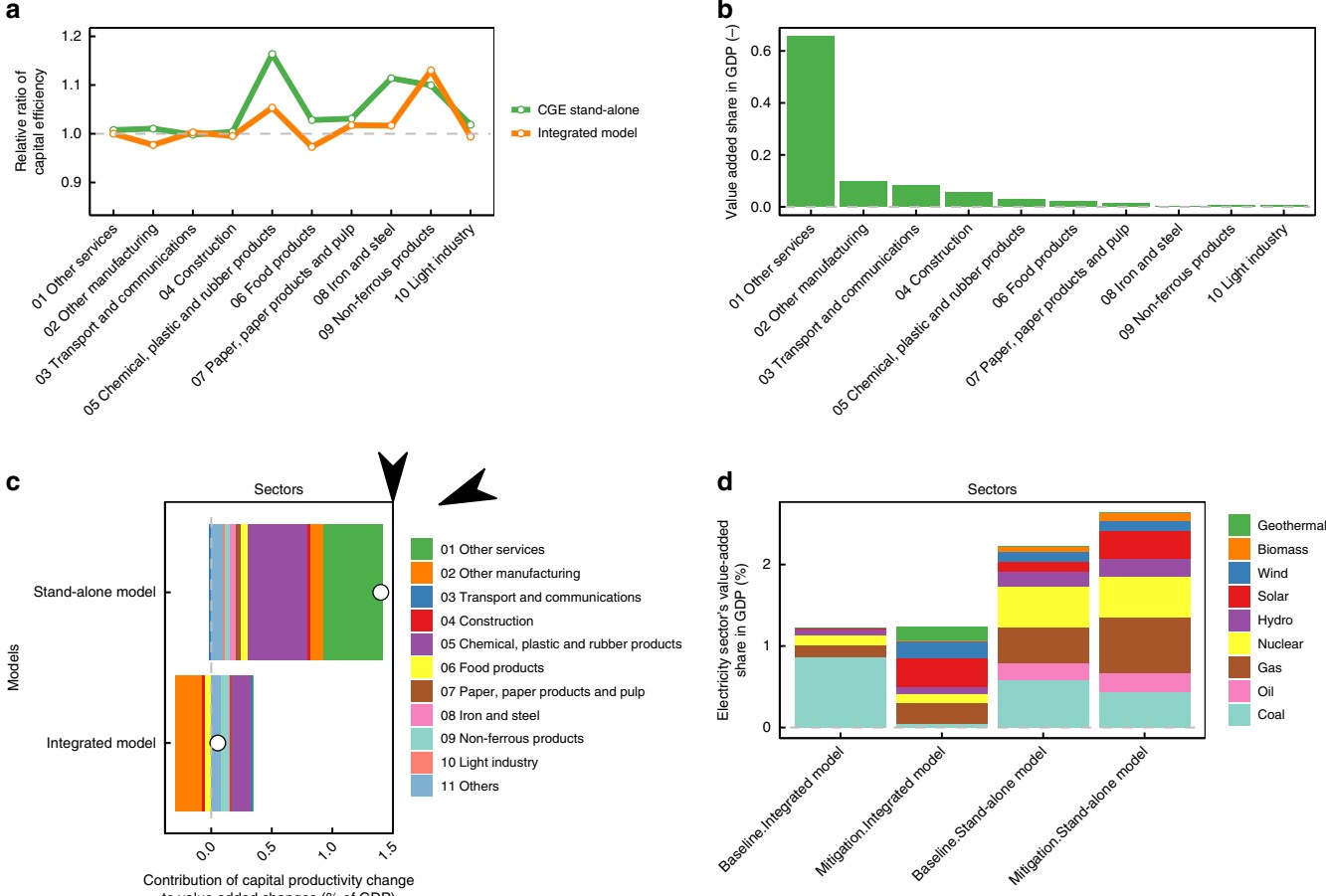

**Fig. 4** Valued-added differences between baseline and mitigation scenarios. **a** Sectoral capital input efficiency for the top 10 industrial activities in 2050. Capital efficiencies in baseline and mitigation scenarios are computed and then the capital efficiency in the mitigation scenario relative to the baseline scenario is shown. **b** Sectoral value-added share in the baseline scenario for the top 10 industrial activities. **c** The capital productivity value-added effects compared to the total value-added for the top 10 industrial activities. Dots means net total changes. Negative and positive values mean capital productivity gain and losses compared to baseline scenarios respectively. **d** The value-added share of power sectors in terms of total economy-wide value-added

CGE stand-alone model. This result may be driven by changes in household expenditures for services, which were around 3.4 and 0.0% in the CGE stand-alone and integrated models, respectively, in 2050. These differences may be due to changes in total income.

We ran further diagnostic scenarios with and without incorporating energy system information by sectors (see Methods for more details) to investigate the extent to which the energy system model's output information for each sector contributes to mitigation cost differences compared to the stand-alone CGE. Comparing scenarios that include a single sector's information from AIM/Enduse and the stand-alone model (Row 1–6 in Table 1, respectively), the inclusion of the industry and service sector information from AIM/Enduse makes a remarkable difference in the GDP loss rate (Row 5 and 4 in Table 1, respectively). From the opposite side, the scenarios taking out the AIM/Enduse information for each sector (Row 7–11 in Table 1) show that excluding the industry and service sectors consistently generates GDP loss differences compared to the integrated model (Row 12 in Table 1). Conversely, the incorporation of residential, transport and energy supply sector information given by AIM/Enduse has a small impact on GDP losses, or even has the opposite effect in some cases. Finally, we can see cross-sectional effects in other scenarios in Supplementary Table 4, which indicates the complexity of the results and shows that the influence of each sectoral impact is not additive. However, the

overall insights are clear, that the industry and service sectors are key in determining macroeconomic implications.

## Discussion

Our newly proposed integrated model approach implicitly assumes that the energy productivity in the CGE model is endogenized by using the energy system model information. This treatment is somewhat different from the conventional approach, in which CGE models use the same Autonomous Energy Efficiency Improvement (AEEI) and constant elasticity substitution parameters, with and without mitigation policies. Based on the results showing that the macroeconomic costs associated with climate change mitigation policies are lower than estimated using conventional approaches, we can interpret the energy productivities in the mitigation scenarios as being higher than in the conventional approach. This would imply that the AIM/Enduse model incorporates higher productivity technological information than the conventional CES approach.

Overall, as long as an energy system model is more reliable than the CGE model in terms of energy-related variables, the energy representation in the conventional CGE should be replaced by the energy system model outputs. The contributions of the industry and service sectors to GDP loss differences are caused by the production function form and its parameters. Basically, for most conventional

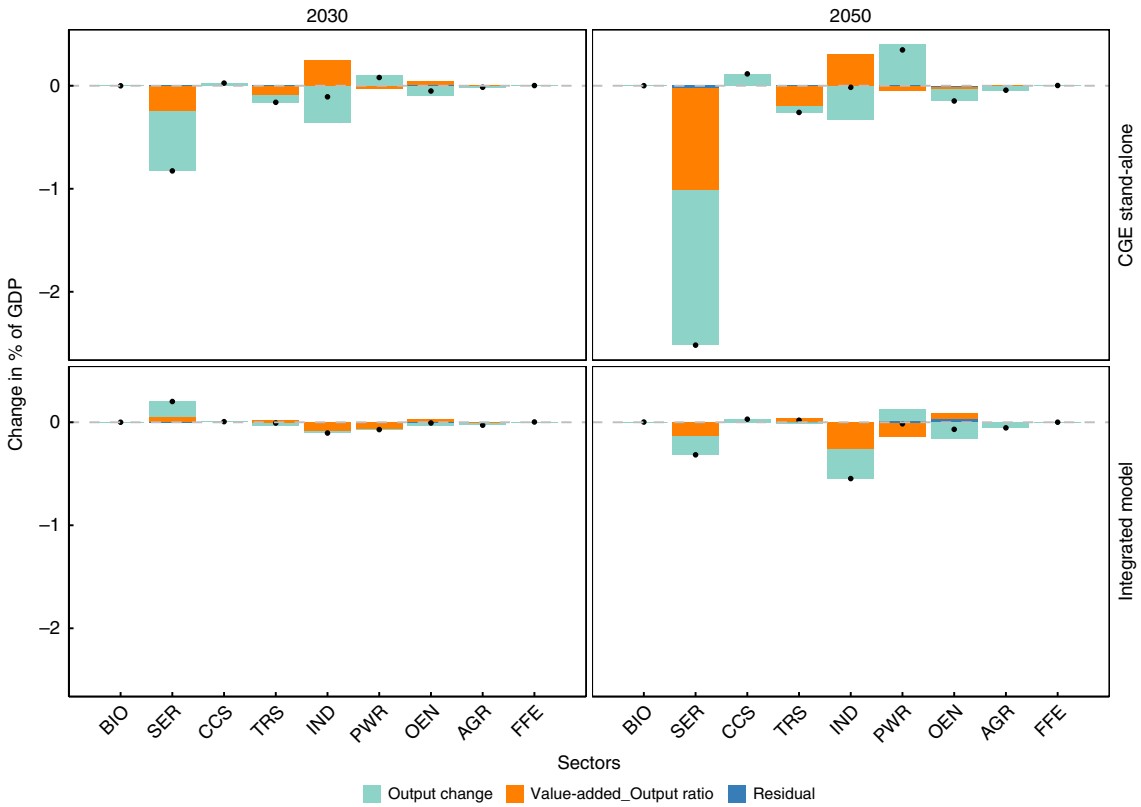

**Fig. 5** Decomposition analysis of GDP changes across sectors. Value-added changes relative to baseline scenarios are expressed as percentages of GDP. Legend entries Output change, Value-added_output ratio, and Residual refer to output changes, value-added productivity changes, and residuals, respectively. The top and bottom panels show CGE stand-alone and integrated model results, respectively. Sectors are BIO Bioenergy industry, SER service sector, CCS CCS industries, TRS Transportation, IND manufacturing and construction, PWR power, OEN other energy supply, AGR agriculture, and FFE fossil fuel extraction

| Table 1 Diagnostic scenarios and their GDP loss rates in 2050. | | | | | | | |
|---|---|---|---|---|---|---|---|
| | Energy supply | Industry | Service | Transport | Residential | GDP loss rate (%) | |
| | | | | | | 2030 | 2050 |
| 1 | off | off | off | off | off | 1.1 | 2.4 |
| 2 | off | off | off | off | on | 0.9 | 2.3 |
| 3 | off | off | off | on | off | 1.1 | 2.4 |
| 4 | off | off | on | off | off | 0.6 | 1.7 |
| 5 | off | on | off | off | off | 0.4 | 0.8 |
| 6 | on | off | off | off | off | 0.9 | 2.2 |
| 7 | off | on | on | on | on | 0.1 | 0.2 |
| 8 | on | off | on | on | on | 0.5 | 2.2 |
| 9 | on | on | off | on | on | 0.4 | 1.2 |
| 10 | on | on | on | off | on | −0.1 | 0.6 |
| 11 | on | on | on | on | off | 0.1 | 0.8 |
| 12 | on | on | on | on | on | 0.0 | 0.8 |

Column names are sectors, and on and off refer to whether AIM/Enduse information is incorporated. The red and blue rows indicate the stand-alone and integrated models, respectively. Yellow and green rows indicate scenarios that include and exclude information from a single sector given by AIM/Enduse, respectively

CGE models, the substitution elasticity of energy and value-added in these sectors use values referenced from the literature[34]. This representation has two possible disadvantages. First, historical price-induced energy and capital substitutability data are based on past events and limited to developed countries. Future technological availability, which is represented by the energy system model in this study, may change drastically. Second, the elasticity parameter is sometimes assumed to be uniform, but it should differ among sectors, and probably regions (this study uses the global model's uniform value for the stand-alone model).

There can be a discussion on the parameter choices in the conventional CGE models and a question whether our results are robust to the key parameter assumptions. To this end, we conducted a sensitivity analysis, varying the elasticity substitution between energy and value-added from 0.2 to 0.8, taking the range from the literature[35]. The results showed that the cost differences associated with variation in the substitution elasticity parameter are much smaller than the differences between the integrated and stand-alone models (see Supplementary Fig. 4). This implies that even if the wide range of values for the substitution elasticity

parameter (as seen historically in the literature) is considered, future technological changes represented by the energy system model cannot be expressed.

To represent the production functions, an alternative approach to CES-type methods already exists in the econometric method[36]. In contrast to this approach, our method relies on realistic representation of technological availability. Therefore, we can identify explicit technological changes that are consistent with the general equilibrium framework. Note that this process implicitly assumes that currently non-existent technologies are excluded, whereas the conventional approach using possible substitution could implicitly assume an infinite possibility to decrease energy consumption in response to energy price signals.

GDP loss differences associated with the household sector's representation in the conventional and integrated models were small, but we need to consider the disadvantages of measuring the mitigation cost as GDP loss. Household expenditure is a major component of GDP in the expenditure accounting system, and increases in household expenditure directly boost GDP. Hence, purchasing relatively expensive energy devices such as electric vehicles and heat pumps will not directly decrease GDP, but rather may offset the negative impacts of climate change mitigation costs. Notably, this GDP increase is attributed to the additional expenditure, which may not contribute to an increase in actual welfare. This finding may show one of the limitations of accounting for climate mitigation costs using this type of model.

An energy system model simply represents the reduction potential of energy-consuming devices, but numerous other possibilities exist to change the energy service itself. Artificial intelligence may maintain energy devices more efficiently, or transport demand could be reduced. Material consumption can also change through sharing of goods and services. From that perspective, the mitigation potential and associated cost may be underestimated. Meanwhile, these societal changes could have indirect effects in the opposite direction in terms of energy consumption, as information technology would require additional electricity. The monetary savings realised by decreasing energy usage could be spent on other things, and if it were spent on energy-intensive activities (e.g. tourism using air travel), energy consumption and emissions could increase.

The energy system model's representation of technological diffusion is based on linear programming with some constraints. Thus, this model may be interpreted as the extreme case where a single technology is selected at some point under certain price conditions, such as only electric vehicles being sold in a private car market. Meanwhile, the CES or logit formulations that are typically used in economic models allow multiple possibilities, implicitly assuming heterogeneity in goods and consumers, whose real behaviour should be represented by a utility function that accounts for non-monetary value[37]. This notion is important when interpreting household results derived from integrated model results, where some may select economically irrational technologies and non-monetary factors are present. However, according to our results, industrial activities have more influence over mitigation cost and our conclusions would hold true if we included such heterogeneity.

We achieved relatively fast convergence compared with existing studies. There are two possible reasons for the rapid convergence. First, on AIM/CGE side, the energy consumption is forced to be AIM/Enduse by endogenising parameters that are exogenous in the conventional CGE formula. Second, the major information provided by AIM/CGE to AIM/Enduse that changes the AIM/Enduse response is the energy service changes (output of sectors and total household consumption), but the difference from the previous iteration is less than 1%, which would not change AIM/Enduse results in terms of carbon price or power generation.

For now, this study's approach and the implications thereof are applicable only to Japan, within the context of our modelling framework. Application to other fields by different modelling teams is needed to demonstrate that our findings can be generalised.

For future researches, as reported in the results section, some variables show discrepancies between the two models in the base year. Although we think that this discrepancy does not affect our main conclusion, a more consistent understanding of this type of modelling framework is needed. This understanding may be accomplished by calibrating both models, but such calibration will require substantial additional efforts to fully harmonise the base year data. Although this calibration is not expected to change our conclusions, it is a worthwhile endeavour for future research. Another future potential research based on this modelling is that hard-linkage among the models and in particular, electricity market is now highly demanded to investigate in terms of intermittent supply of solar and wind power generation.

## Methods

**Overview of the method**. Here, we developed an integrated modelling framework that incorporates energy system, power-dispatch, and CGE models, as illustrated in Supplementary Fig. 5. Each model's output is exchanged with the others. We executed five model iterations and assessed the second iteration because the discrepancy improvements were sufficiently small at the second iteration. The calculation begins with an AIM/Enduse run and then uses AIM/CGE and AIM/Power. AIM/Enduse is run again, considering the AIM/CGE and AIM/Power outputs. The electricity demand and supply system under stringent emissions reduction targets would be highly dependent on fluctuations in the electricity supply and demand patterns, which requires operation on an hourly basis. Therefore, we used AIM/Power in this model. We conducted scenario-based simulations through 2050. The individual models were solved from 2010 to 2050, then the results from each were input to the other models. If models interact each other for each year, the convergence could be much faster since current approach can remain the gaps among the models each year, which can be amplified particularly latter period. However, fortunately we have already had good convergences with less iterations. The energy system and related $CO_2$ emissions are the scope of this study, as Japanese GHG emissions are associated with these factors. In this study, we excluded the effect of climate change damage on the economy to avoid complexity (e.g. isolating mitigation effects from the mixture of climate change mitigation and damage impact, and additional assumptions on other countries' emissions situations). The baseline socioeconomic assumptions are based on Shared Socioeconomic Pathways 2 described in Fujimori et al.[38].

**A computable general equilibrium model**. The CGE model used in this study is a recursive dynamic general equilibrium model that covers all regions of the world and is widely used in climate mitigation and impact studies[39–43]. The main inputs for the model are socioeconomic assumptions of the drivers of GHG emissions such as population, total factor productivity (TFP), which should reproduce the GDP assumptions in baseline scenarios, energy technology, and consumer preferences on diet. The production and consumption of all goods and GHG emissions are the main outputs based on price equilibrium. The base year is the year 2005.

One characteristic of our industrial classification is that energy sectors, including power sectors, are disaggregated in detail, because energy systems and their technological descriptions are crucial for the purposes of this study. Moreover, to appropriately assess bioenergy and land-use competition, agricultural sectors are highly disaggregated[44]. Details of the model structure and its mathematical formulas were provided by Fujimori et al. and wiki page[45].

Production sectors are assumed to maximise profits under multi-nested constant elasticity substitution (CES) functions at each input price. Energy transformation sectors (Supplementary Table 5) input energy and are value-added based on a fixed coefficient, whereas all energy end-use sectors (Supplementary Table 6) have elasticities between energy and the value-added (CES aggregation of capital and labor) amount. These sectors are treated in this manner to account for energy conversion efficiency in the energy transformation sectors. Power generation from several energy sources is combined using a logit function[46], although a CES function is often used in other CGE models. We chose this method to represent energy balance because the CES function does not guarantee a physical balance[47]. As discussed by Fujimori, Hasegawa[44], an energy or physical balance violation in the CES would not be critical if the power generation shares of each technology in total power generation were similar to the calibrated information. The hydrogen production sectors have similar structure as power generation. In this study, climate mitigation changes the power generation mix when compared to that of the base year, and therefore is a key treatment. The variable renewable energy cost assumption is shown in SI

section 2. Household expenditures on each commodity are described with a linear expenditure system (LES) function. The savings ratio is endogenously determined to balance savings and investment, and capital formation for each item is determined using a fixed coefficient. The Armington assumption, which assumes imperfect substitutability between domestically produced and traded goods[48], is used for trade, and the current account is assumed to be balanced.

To construct energy supply cost curves, we implemented multiple sources of information. Solar and wind supply curves are from a study considering urban distance[49]. Biomass potential and supply curve data is from a land-use allocation model[50].

**An energy system model**. The energy system model used in this study is a recursive dynamic partial equilibrium model based on detailed descriptions of energy technologies in the end-use and supply sectors. In this study, we used the multi-region version of AIM/Enduse [Japan][51], which divides Japan into 10 regions (see Supplementary Fig. 6) based on the power grid system. The model covers energy-related GHG emissions from both energy end-use and energy supply sectors. The end-use sectors are composed of industry, buildings and transportation sectors, and they are disaggregated into several subsectors with respect to types of products, buildings, and transportation mode based on the IEA energy balances. The $CO_2$ emissions constraint is assumed for every simulation year of the AIM/Enduse model under the mitigation scenario. Within this study, the carbon price trajectory is almost exponential as a consequence. Therefore, even if we adopt an inter-temporal optimisation scheme, it would not markedly affect the results. However, this might not be the case for other carbon constraints. Mitigation options are selected based on linear programming to minimize total energy system costs that include investments for mitigation options and energy costs subject to exogenous parameters such as cost and efficiency of technology, primary energy prices, energy service demands, and emission constraints. Detailed information on the model structure and parameter settings are provided in Kainuma et al.[52] and a list of technologies is given in Supplementary Table 7. As the models used in this study were recursive dynamic, we did not consider discounting the energy system costs. Nevertheless, the AIM/Enduse model annualises the capital costs of energy technologies using a discount rate in the range 5–33% (Oshiro et al.)[53]. The sectoral discount rate is 5% for power and industry, 10% for transportation, and 33% for other sectors. These individual discount rates are only applied to simulate technology selection in the energy system model. Consequently, the energy investment data fed into the economic model are not discounted by these rates.

The power sector is modelled in detail, considering the balances of electricity supply and demand in 3-h steps to assess the impacts of variable renewable energies (VREs). This sector also includes measures to integrate VREs into the grid, such as electricity storage, demand response (DR) using battery-powered electric vehicles and heat-pump devices, and interconnections. The total capacity was calculated based on the capacity of newly installed power plants, which was determined endogenously, as well as that of existing plants. In the AIM/Enduse model, the residual capacities of the existing power plants in operation in 2010 were calculated based on individual powerplant information, such as year constructed, capacity of each plant, and expected lifetime.

In the industry, building and transportation sectors, wide mitigation options are included, such as energy-efficient devices and fuel switching. The industrial sector also includes innovative technologies such as carbon capture and storage (CCS). However, the AIM/Enduse stand-alone model does not account for some mitigation options that contribute to reduction in energy service demands. The key power generation technoeconomic information is shown in Supplementary Table 8. The cost information is based on METI data (2015)[54], as they are consistent with the assumptions in Japan's NDC. Note that the estimated mitigation cost may become much lower under more optimistic assumptions regarding future cost reductions, especially for renewable energies. Moreover, powerplant information in 2010 and fuel assumptions are shown in Supplementary Table 9 and Supplementary Table 10.

**A power-dispatch model**. The power-dispatch model used in this study is a recursive dynamic partial equilibrium model focused on generation planning for the power sector. In other words, unlike the AIM/Enduse model covering all energy-related sectors, the AIM/Power model only covers the power generation sector. This model can simulate hourly or annual electricity generation, generation capacity, plant locations, and multiple flexible resources, and includes interregional transmission, dispatchable power, storage, and demand responses. These variables were selected based on linear programming while minimising the total system costs, including capital costs, operation and maintenance costs, and fuel costs under several constraints, including satisfying electricity demand and $CO_2$ emission reduction targets. In this study, we used a version of the model that classifies Japan into 10 regions (see Supplementary Fig. 6). Detailed information about this model can be found in Shiraki et al.[55]. Note that as AIM/Enduse provides power generation installed capacity for AIM/Power, AIM/Power does not make investment decisions, except for making additional investments in storage and power plants aimed at hourly and within hourly power demand-supply management

AIM/Power can explicitly simulate the hourly demand-supply balance of electricity, with consideration of daily variations in photovoltaic output caused by weather conditions as well as seasonal and weekday/weekend variations in demand.

In addition, the demand-supply balance of electricity within an hour is modelled using the fluctuations and flexible range of each generator. Although generators and flexible resources are modelled in detail, electricity demands are provided exogenously. Thus, the power-dispatch stand-alone model does not determine the total electricity consumption and installed capacity by technology, which are given parameters. Note that there are buffers to deal with seasonal fluctuations, such as fossil fuel CCS thermal plants, in the mitigation scenarios, and thus, even if we consider battery storage for seasonal fluctuations, it would remain unused due to the cost competitiveness. The no CCS scenario also uses gas thermal plants to adjust for seasonal differences.

**From the energy system model to the economic model**. The following information is given to AIM/CGE from AIM/Enduse outputs. First, Change ratio of final energy consumption by sector and energy type; second, power generation share by energy source; third, battery capacity for stabilising fluctuations of the power supply and its capacity factor, which is taken from AIM/Power (this capacity factor means the total hours that the battery used divided by a year); forth, CCS installation; fifth, investment in energy end-use sectors; sixth, carbon prices; seventh transmission losses.

Final energy consumption is classified into four sectors (industry, transport, service and residential) and fed into the CGE model. We exogenously represent these sectors, while autonomous energy efficiency improvement (AEEI) parameters are endogenised. This treatment maintains the same number of equations and variables as in the conventional CGE approach. To integrate household energy consumption and energy device purchase activities in the household, we divided the household expenditure into four categories, such as car-use activities and other energy consumption activities, as illustrated in the Supplementary Fig. 7 (see more detailed information in Supplementary Note 2 and Supplementary Table 11). Because the absolute value of energy consumption is not fully harmonised between these two models, we compare the change ratios of energy consumption with 2010 levels, which is the base year of the AIM/Enduse model, for final energy consumption determination. If the corresponding energy consumption was zero or very low in 2010 (less than 1 ktoe), the change ratio can lead to unrealistic projections; therefore, we use absolute values. The investment in energy end-use sectors is input as an incremental capital cost compared to the baseline case, where investment costs in the baseline is modelled by CES substitution. Moreover, the capital input coefficients are fixed at baseline levels so that additional energy investment is represented by AIM/Enduse information rather than CES substitution elasticity in the mitigation scenarios.

**From the economic model to the energy system model**. Because the sectoral disaggregation of AIM/Enduse basically complies with the IEA energy balance, there are inconsistencies in the AIM/CGE, which is based on an input-output table. Thus, in terms of data exchange from AIM/CGE to AIM/Enduse, the subsectors are aggregated so that the granularity of the sectors is in agreement. Nevertheless, given the large share of industrial GHG emissions in Japan's long-term low-carbon scenarios, iron, chemical, paper, non-metallic minerals, and non-ferrous metals are exempted from the sector aggregation. AIM/Enduse uses the following information generated by AIM/CGE: first, GDP changes; second, household consumption changes; third, industry and service sector outputs; fourth energy price changes

Economic information from AIM/CGE is input into AIM/Enduse as changes in energy service demand for each sector. Transport demand is associated with GDP projection in AIM/Enduse and we proportionally change the transport demand based on changes in GDP. The energy service demand in the industrial sectors, such as steel and cement production, and outputs of other industrial sectors, is altered by the outputs from AIM/CGE. Energy service demand in the household and industrial sectors could have low or high elasticities to relevant economic activity variables, such as household consumption and outputs of service sectors, but remains an uncertain factor. According to the Swedish econometric analysis[48], elasticity between monetary and physical units of energy services can be assumed to be ~1.0. This elasticity accounts for the percent change in physical energy services caused by a 1% change in monetary outputs. Furthermore, the GDP losses indicated in this study are relatively small, less than 3%, in the CGE stand-alone model, as shown in Fig. 4a. Thus, we tentatively applied an elasticity value of 1.0. Meanwhile, we varied the elasticity from 0.5 to 2.0 and observed that the policy costs change slightly, but the qualitative conclusion still holds (Supplementary Fig. 8).

**From the energy system model to the power-dispatch model**. AIM/Power's role is to present the feasibility of power-dispatch given an electricity demand and installed power capacity. Thus, AIM/Enduse provides the following items to AIM/Power: first, electricity demand; second, power generation installed capacity; third, demand response technological availability, such as heat-pump water heaters and electric vehicles

**From the power-dispatch model to the economic model**. AIM/Power provides more realism in terms of technologies to stabilise short-term fluctuations in the power system than the other two models used in this study. Moreover, the power system would respond to large-scale renewable energy installations by adjusting the capacity factor for conventional power generation systems (e.g. coal-fired power) in addition to curtailing the output from variable renewables. These measures for

balancing short-term fluctuations reduce the electricity output per installed capacity, and thus affect investment decisions. It is necessary to consider this feedback from AIM/Power to AIM/Enduse. In summary, the following AIM/Power information is given to AIM/Enduse: first, battery capacity needed to stabilise short-term electricity fluctuations; second, Curtailment ratio; third, capacity factors.

Note that generation capacity, although not directly related to balancing short-term fluctuations, is decided by AIM/Enduse. Thus, the battery capacity is determined by AIM/Enduse when considering long-term electricity fluctuations, whereas that for short-term fluctuations is provided by AIM/Power.

**Convergence of iterations**. We confirm fast convergence between the models. Detailed discussion related to the convergence is made by Supplementary Note 3 where Supplementary Figs. 9–20, Supplementary Table 12 and Supplementary Table 13 shows actual convergence situation. As stated, the discrepancy between the model almost reaches convergences in the second step.

**Scenario assumptions**. There are two basic assumptions for future scenarios, namely, baseline and mitigation, which are carried out with and without carbon pricing to reduce GHG emissions by 80% in 2050. Basic assumptions on technological conditions, such as nuclear scenarios and CCS capacities, are taken from previous studies[56]. In the results section, we describe how the mitigation cost differs from that in the stand-alone AIM/CGE run to identify each sector's contribution to the changing mitigation costs.

**Analytical method for the diagnostic scenario runs**. To investigate the extent to which AIM/Enduse output information for each sector contributes to mitigation cost adjustment compared to the conventional CGE approach, we ran diagnostic scenarios with and without incorporation of AIM/Enduse data by sector, as noted in the supplementary information. Ultimately, we conducted 32 scenarios with various combinations of AIM/Enduse information for energy supply, industry, service, transport, and residential sectors taken into account or excluded. The indicator shown below is adopted. i, t, and s are sets of variables (e.g. energy demand), years and scenarios, respectively. $X_{i,t,s}$ and $Y_{i,t,s}$ are AIM/CGE and AIM/Enduse outputs, respectively.

$$\text{ErrInd}_{i,s} = \sqrt{\frac{\sum_t \left( X_{t,i,s} - Y_{t,i,s} \right)^2}{\left[ \sum_t \left( \frac{X_{t,i,s} + Y_{t,i,s}}{2} \right) \right]^2}}$$

**Reporting summary**. Further information on research design is available in the Nature Research Reporting Summary linked to this article.

## Data availability
Scenario data are accessible online via HARVARD Dataverse (https://doi.org/10.7910/DVN/QE6ERU). The data which are derived from the original scenario database shown as figures but not in the above database is available upon requests. The source data underlying Figs. 1–5 and Supplementary Figs 1–20 are provided as a Source Data file.

## Code availability
The source code for generating figures used in the main text and supplementary information is available in HARVARD Dataverse (https://doi.org/10.7910/DVN/QE6ERU), which name is the Sourcecode.zip. The current code base of the AIM/Enduse, AIM/CGE and AIM/Power developed over more than two decades at Kyoto University, Shiga Prefecture University and National Institute for Environmental Studies. AIM/Enduse is available at AIM website (http://www-iam.nies.go.jp/aim/data_tools/index.html#enduse) and others are not available in a publicly shareable version. The code will continue to be developed and hosted by Kyoto University, Department of Environmental Engineering (http://www.athehost.env.kyoto-u.ac.jp/). Requests for code should be addressed to Shinichiro Fujimori.

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

## Acknowledgements
S.F., K.O., and T.H. are supported by the Environment Research and Technology Development Fund (2-1702, 2-1908) of the Environmental Restoration and Conservation Agency of Japan and JSPS KAKENHI Grant (No. 19H02273). S.F. is supported by COMMIT (Climate pOlicy assessment and Mitigation Modeling to Integrate national and global Transition pathways) financed by Directorate General Climate Action (DG CLIMA) and EuropeAid under grant agreement No. 21020701/2017/770447/SER/CLIMA.C.1 EuropeAid/138417/DH/SER/MulitOC. H.S. is supported by the Murata Science Foundation.

## Author contributions
S.F. designed the research; S.F carried out analysis of the modelling results; S.F. and H.S. created figures; S.F. wrote the draft of the paper; S.F., K.O. and H.S. provided economic, energy system, and power model data; and S.F., K.O., H.S., and T.H. contributed to the discussion and interpretation of the results.

## Competing interests
The authors declare no competing interests.
