## [Peer Review File · Nature Communications]

Reviewer #1 (Remarks to the Author):

Summary and major claim of the paper

The paper aims to assess future economic costs of climate change mitigation. To this end, it devises a new integrated approach and applies it to Japanese 80% decarbonization pathways to 2050. The new integrated approach combines/soft-links an economic (top-down) CGE model with a (bottom-up) energy sector model and a technically detailed power system dispatch model. The central result states that economic costs of decarbonization – measured in percentage losses in GDP compared to a baseline without any decarbonization measures – are smaller when derived using the new integrated approach than when derived from a standard stand-alone economic CGE model. The reason for that would be the following: a standard CGE model employs a constant elasticity of substitution between the input factors energy and all others. This value is derived based on historical data and may not capture conceivable future developments in the energy sector. The new approach determines the elasticity of substitution endogenously and could better represent future changes in the energy system and their impact on the economic production process.

Interest to others in the community and the wider field

The proposed approach is of high interest to the community and wider field. Ex-ante “evaluations” of the economic consequences of policy measures are of great relevance to the research community and policymakers when it comes to assess climate change mitigation measures. Also the identification of mechanisms that affect GDP loss estimates is timely and relevant.

Overall evaluation

Overall, I like the idea of the paper. It touches a relevant point but suffers from various weaknesses. It leaves me somewhat puzzled.

- In a nutshell, too many points are opaque to assess both the novelty compared to the existing literature and the soundness of the proposed integrated model.
- There is also a range of conceptual issues, especially concerning model iteration and convergence as well as the “econometric” analysis
- Moreover, the external validity is highly doubtful as the application is driven by a number of (partly unstated) assumptions. A focus on the internal validity and the methodological contribution how and why the soft-linking decreases GDP loss estimates would help to make the paper more credible.

The authors must devise good efforts to make the paper clearer in both methods and communication and satisfactorily address all major points. Only then, I can soundly judge the paper and potentially recommend publication. Please find my detailed review below.

Major points

Concerning structure

The authors could do a better job in coming to the core of the paper (as I understand it, the endogenization of the substitution elasticity between energy and other inputs in the economic production process by using an energy system model).

- The application to Japan is illustrative and should stay in the paper, but I see the contribution in the method. I highly doubt the external validity as numerical results are driven by numerous

assumptions on functional forms and input data.

- Likewise, the authors devote a lot of space to describing the differences between stand-alone CGE and integrated model. This can be shortened or longer parts relegated to an appendix. More space should be devoted to the central mechanism. The why is more relevant than the what

- Therefore, authors should streamline the paper and feature the central mechanism more prominently. In the current manuscript, it becomes clear only in lines 222 - 232

Concerning analysis

The exposition of the models is opaque and must be considerably improved. The model is highly complex but under-explained. It is difficult to deeply understand what drives the results. Please apologize that I go into details here, but all details together constitute a major issue.

- Overall language must be more precise. Especially, central concepts must have a clear and immutable name. E.g.

o "value-added", "energy service demand", ...

- CGE and energy system model:

o Clearly state which sectors you analyze. I was puzzled throughout the paper

- CGE model

o The description must be clearer. I have some experiences with CGE models, but lines 292 to 310 are opaque

→ Which production sectors? Is there a final consumption good or multiple?

→ Which energy transformation sectors? What do you mean here by "value-added" as an input?

Do you mean capital?

→ Which energy end-use sectors? What do you mean here by "value-added" as an input?

→ Please explain the concepts of energy and material balances in power generation more clearly!

→ Lines 299-300: Which share do you mean? Which calibrated information do you mean?

- Energy system model Enduse

o Clearly state the scope of the model!

o Line 314: Be more specific on energy end use and supply sectors! Are they the same as in the CGE? Are they a disaggregated representation of the sectors in the CGE?

o Line 318: Why are energy prices a constraint in the model? Which energy do you mean?

Electricity? In my understanding of an energy system model, they are a result.

o Line 325: Which energy-demanding sectors? Those in Table S3?

- Power dispatch model

o Clearly state whether investment decisions are part of that model! It seems not, but in line 332 you call it a generation expansion model

o Lines 336-337: What do you mean by "or both"? Is it possible that the model satisfies either only electricity demand or only emission reduction targets? This appears strange to me.

o Also explain the difference in the scopes of the power dispatch and energy system models more clearly! What do they cover?

The explanation of the communication between models is unsatisfactory and must be improved.

- Exchange from Enduse to CGE

o You state that both AIM/Enduse and AIM/Power decide on battery investments. Please clarify!

o Line 367: What do you mean by "baseline case"? What do you mean by the implicit assumption "that the baseline investment cost is inherent in the CES substitution"?

o Line 368: What do you exactly mean when you say that "capital input coefficients are fixed at baseline levels"? I think that this is the central point of the paper where you replace the

exogenously assumed substitution elasticity by results from the energy system model. If so, please explain way more clearly and understandably. This is at the heart of the contribution of the paper!

- Exchange from AIM/CGE to AIM/Enduse

o Line 382: What do you mean by "industry service demand"?

o Lines 384-386: Tentative guess of substitution elasticities being unity based on some Swedish econometric analysis is doubtful. Please more explanation here or argumentation why this is okay.

- Exchange from AIM/Enduse to AIM/Power and vice versa

o Please clarify in which model you determine which battery capacities!

o Please clarify the importance of the capacity factors! If you have a three-hour granularity in the Enduse model, why are they important (for investment decisions)?

- Iteration and convergence

o Line SI25: How do you apply an RMSE? If you compare two values (for the same variable in the two models), then the concept is not applicable. If you compare more than two values, please clarify!

o Table S2: In any case, the table is difficult to understand, please explain more clearly what the percentage numbers mean!

o Table S2: I do not see convergence. Basically, results for almost all variables differ by a large degree between CGE and Enduse, and only the magnitude of the difference stays roughly constant after the second iteration. In my understanding, convergence must imply that variables take on identical values between models.

o Related to that, two iterations seem unrealistically few.

o Please also comment on the computational burden

o You state that "the absolute value of energy consumption is not fully harmonized between" the CGE and energy system model (line 362). Do you refer to Table S4 here? If no, explain. If yes, this seems problematic for me.

Concerning novelty

As the analysis has many unclear points, it is hard to judge genuine novelty. The literature features a range of attempts to combine top-down CGE with bottom-up energy system models; among others:

- The Bohringer-Rutherford stream (e.g., Energy Economics 30 (2008), 574-596, Journal of Economic Dynamics and Control 33(9) (2009), 1648-1661 and further papers by the same author)

- Tuladhar et al. (Energy Economics 31(2), (2009), S223-S234)

- Abrell and Rausch (Journal of Environmental Economics and Management 79 (2016), 87-113)

- Tapia-Ahumada et al. (Economic Modelling 51 (2015), 242-262)

- Hwang and Lee (Energy Policy 83 (2015), 69-81)

- The IAM community (e.g., REMIND (<https://www.pik-potsdam.de/research/sustainable-solutions/models/remind/remindequations.pdf>))

Several of the above papers also explicitly analyze macro-economic effects (Tuladhar, Abrell and Rausch, Hwang and Lee). I did not carry out an exhaustive search, but there are, confidently, more.

As the exposition of the model in this manuscript is shallow and not sufficiently clear, it is hard to judge whether its contribution is sufficiently novel. While energy system-macro model integration is not entirely new, I would judge the integration of two highly detailed models sufficient. Likewise, the observation that CGE and energy system models differ in results concerning their

GDP impact is known, as the authors state. The new insight of this paper would be why they differ. The explanation given is plausible ("kind of" endogenous substitution elasticity), but does not become sufficiently clear from the analysis. This is a main point the authors must address.

Further major points

Diagnostic scenarios

As such, a good idea, but the regression analysis is awkward (Tables 1 and S4). Drop it.

- You have a small sample (N=32 in Table S4). Estimators cannot be assumed to be distributed normally. Did you account for that?
- Regression analysis is an appropriate method to estimate population effects from samples. Your data is deterministic. You know the statistical population. There is no need for regression analysis.
- You can just compare means. Take the mean of the GDP result of all scenarios with a certain feature (16 when for a single year) and the mean of the GDP result without this feature and compare.
- It would also be interesting to see whether the detected effect (e.g. representation of industry is relevant) is more pronounced in combination with other features. You could populate a Table like S3 with your GDP results. This would be more insightful
- All tables must state the sample size and the R-squared
- Line 175: as far as I see you do not have a fixed effect (=time-constant unobserved heterogeneity in the cross-sections)
- In any regression, you estimate parameters with given data for variables; adjust terminology

Throughout the paper, do not call it mitigation of GDP loss. Neither results from the CGE model nor results from the integrated model are the truth. Please consistently call it something like difference or wedge between results.

Line 166: It is unclear what you mean by "with and without incorporating energy system information by sector". Do you switch off the entire sector in Enduse or do you suppress transferring the information to CGE?

Lines 221-235: You clearly conclude that it is the industry and service sectors whose representation causes model results to be different. The question remains why. You mention that it is caused by the production function form and its parameters. Again, this central point must become crystal clear.

Model

How do you implement the temporal structure?

- Do you solve each model for 2010 until convergence and then proceed to 2015, 2020 and so on? Or do you solve Enduse for 2010 – 2050 and then hand over to CGE and iterate?
- Please be more specific on your CO2 budget. Is it one budget for the entire 2015-2050 period so that banking and borrowing, e.g., delayed decarbonization is possible?
- Please be more specific on (social) discounting
- Do you start with the existing Japanese power system? What are your assumptions on capital depreciation?

Please explain clearly why you do not account for climate damages on the economy, i.e., both production process and household utility (which hides behind linear expenditure system)!

When I work on forward-looking analyses, technological change is a key driver for results

- In both an energy-system and a macro-economic perspective, technological innovation and learning curves, i.e., decreasing costs are, of utmost importance. Think of solar PV or, currently,

batteries

- Your technology cost assumptions are missing. Be transparent on all of them (appendix)
- Do you incorporate decreasing technology costs? If not, defend result!
- Sector coupling is key for (very) low-carbon energy systems (e.g. see the literature by M.Z. Jacobson or C. Breyer). Please make it clearer in your model description how you deal, for instance, with hydrogen. Is it somehow included in your sectors? This would also be a credible long-term storage.

Minor points

Language must be improved

- Abstract: Try to shorten to at most 100-125 words
- Abstract: What do you mean by 100% lower losses? Are they zero?
- Line 24: Nationally
- Line 33: 5th IPCC reports states 2-6%
- Line 48: "Some argue..." Who?
- Figure 4c: Parentheses do not match
- Line 359: word "its" not appropriate
- Line SI57: check table heading

Further points

- Line 22: climate change is a particular challenge for developing countries
- Line 69: what about seasonal storage, which should become relevant under 80% decarbonization?
- Figure 1: the right column with arrows is hard to understand. Find a better format for representing exchange between models
- Line 103 and Figure S1 state that economic growth is a model input. How does this conform this GDP being the central output of interest?
- Line 109/Figure 2f: I don't see strong electrification in the figure
- Figure 2c: How can final energy demand be larger primary energy? Imports? Heat pumps?
- Figure 2a: What drives changes in the baseline?
- Figure 2b: Why is there a marked kink in 2015 Oil w/o CCS?
- Figure 2g: What are the low-carbon liquids, gases, and solids?
- Figure 2h: Please explain the extreme high CO2 price in 2050. Model-ending effects?
- Figure 2: Please use one clearly distinguishable color for one data point!
- Figure 3: Curtailment numbers are quite high; please comment on connection to (conceivable) sector coupling!
- Lines 136- 137: Please state the "usual" numbers!
- Line 211-212: If you include an energy system model in a CGE model, your results indicate lower GDP losses. This does not mean that GDP losses will be in this range. As said, external validity is doubtful for me.
- Table S1: Sample size, R-squared
- Table S3: Dispensable in this form; can be condensed to text

Reproducibility

Not given in this form. Basically all data and specific formulations on the model are missing. Also the description of the model is too vague to re-build the analysis from scratch.

Reviewer #2 (Remarks to the Author):

GENERAL COMMENTS

The main concern for this paper is robustness of results for macroeconomic cost. The claim that economic models “are known to project higher costs than energy system models” is too broad a statement, given the wide variety of economic models and how they are parameterized. I would expect that counter-examples exist.

The paper could be re-framed as an exploration of methods to better characterize technology evolution than is currently possible in existing Integrated Assessment Models (IAMs), especially for deep de-carbonization scenarios.

A major strength of the paper is the use of a power dispatch model that deals with hourly patterns of supply and demand, and requirements for electricity storage.

The paper can be improved with less emphasis on macroeconomic costs, and more emphasis on the challenges of model integration for deep decarbonization.

The key model linkage appears to be that from AIM/Enduse to AIM/CGE (discussion starting at line 347), where the autonomous energy efficiency parameters (AEEI) are endogenized (lines 357/358) to match final energy consumption. This essentially endogenizes productivity in the model, which can cause large changes to the cost of emissions mitigation. More thought and discussion are needed on the implications of this type of model linkage. How comparable are scenarios if productivity parameters are endogenous?

Comparison to other types of model integration would be helpful. In particular, the full integration method demonstrated by Böhringer and Rutherford (reference number 19) is an ideal goal, although hard to implement in large models.

The paper contains a lot of technical detail that will be difficult for a general audience to assess. Therefore, the paper may be better suited for a subject-specific journal than for Nature Communications.

SPECIFIC COMMENTS

Change in GDP is generally considered an inferior measure of policy cost in computable-general-equilibrium (CGE) models. Equivalent variation is the preferred measure and it should be possible to calculate both GDP loss and equivalent variation in AIM/CGE for comparison.

The sentence in the abstract at lines 12/13, “The GDP losses estimated with the integrated model were significantly lower than those in the conventional economic model by more than 100% in 2050,” is not supported by analysis in the paper. The reduction appears closer to 50%.

It would help to include discussion on the need for two energy system models, AIM/Enduse and AIM/Power. Does AIM/Enduse simulate electricity demand by hour, by month, or by year?

Reviewer #3 (Remarks to the Author):

Review of «Energy Transformation Cost for the Japanese Mid-century Strategy: Energy System

Feedback Effects in an Economic Model»

The paper investigates mitigation costs from reaching 80% greenhouse gas reductions in Japan in 2050. The study uses three models to analyze this question: a top-down CGE model (AIM/CGE), a bottom-up energy system model (AIM/endues) and a power dispatch model (AIM/power). Energy system results are taken as input into the other two models, and results are presented back as input to the energy system model.

This kind of study is demanding, since it requires insight in very different model types, covering broad sets of expertise. The analysis is based on repeated runs of the different models, exploring demanding challenges involving projections over several future decades.

The paper is interesting! Solid work has been done. I still think the manuscript may be improved on some aspects.

First, I have some general comments/questions:

- The paper focuses on mitigation costs, measured by GDP loss rates. The authors compare estimated costs from different model runs. It would be interesting to know if policy recommendations differ from an integrated model instead of stand-alone models (or alternatively whether policy assumptions or setup interpretation would differ in a stand-alone versus integrated setup). If policies do not differ between the model runs and different policy recommendations could not be inferred from model results, the question is which solution would be realized in real life. This does not depend on the model estimates, but on real-life action and real-life politics. The analysis focuses on reduced "model" mitigation costs from one model setup compared to another, and could be viewed as disconnected from real-life. If policy recommendations are similar, then real life outcomes following model result recommendations would not differ, even though different models estimate different costs. I am not convinced from the paper that the assertion in line 15 is justified, saying: "this type of integrated approach would be highly beneficial for setting national climate policies". This might indeed be true, but the paper does not substantiate the assertion.
- Models are usually meant to bring new insights. CGE models are criticized for being black-box models. The paper handles the models rather as black-box models. Instead of explaining cause and effects by utilizing the models, the paper uses statistical analysis of model results in order to explain model relations. I think the authors should comment or discuss whether assumptions for using a regression model are justified (distribution of variables, independent residuals, ...). Performing statistical analysis on model results and discussing statistical significance of impact from economic sectors seems unconventional.

Example:

Line 172: "The transport sector's effect (on GDP loss) is ambiguous, and its t-value is too small to reject the null hypothesis."

Line 382: "We proportionally change the transport demand based on changes in GDP."

If there is a clear model relation between transport demand and GDP, it might seem inappropriate to perform a statistical analysis on this relation.

- There is unfortunately no common agreement on classifications of hybrid models. The term "integrated model" could be interpreted by some modelers as if underlying models run together (simultaneously) as one integrated mathematical model (and possibly solved simultaneously by one solver). The term integrated model is used differently in this application. It would be interesting to know whether results are exchanged automatically (this might be referred to as hard-linked models) or manually (this might be referred to as soft-linked models). The low number of iterations (two iterations) could indicate soft-linking, but the high number of scenario setups (32 scenarios) could indicate hard-linking.

Furthermore, I have some detailed, minor comments.

Line 12-13: A shortened sentence would read: "The GDP losses were lower by more than 100%". This wording is unfortunate, and the sentence should be reformulated. (The percentage comparison might come from lines 134-135, since 2.5% is more than 100% larger than 1.2 %.)

Line 24: NDC  I think it is "nationally" (not "national").

Line 39: I think lines 39-44 should be further clarified/rewritten. According to the text, the first examples are global energy system models, and the last are CGE models. I don't think GCAM should be classified as an energy system model (line 39). MESSAGE could be classified as an energy system model, but MESSAGE-MACRO (line 40) represents a linking between a top-down macroeconomic model and a bottom-up energy system model. Line 42 reads: National models have applied "similar approaches". At this point global "energy system" models have apparently been presented, while the model examples following this point do not apply "similar approaches" as such models.

Line 48: "Some argue that ..." – please give references or describe sources.

Line 76: Consider a reformulation of the sentence.

Line 108: The mitigation scenario exhibits ... strong electrification (Figure 2f). How/where is the strong electrification evident? The level of power generation in the baseline (figure 2b) and mitigation scenario (figure 2f) seems rather similar, around 4 EJ/year in 2050 in both scenarios.

Line 120: Is the heat legend relevant for figure 2c and 2g? The color is not visible in the figures. The same comment applies to figures S2 and S9.

Line 137-138: "The mitigation costs under such deep emissions reductions are usually not as low as this study's estimates from the CGE stand-alone model." This sentence says that the CGE stand-alone model estimates very low mitigation costs. Do I read the sentence wrong? I do not follow the logic in the paragraph.

Line 149: "We see systematic overestimates in the stand-alone model." The authors are comparing results from models, but can hardly conclude that the stand-alone model overestimates mitigation costs. How do the authors know the true mitigation costs?

Line 149: Is the word "this" correct? (Comparison of this integrated model's ...)

Line 154 - Figure 4: Scenarios "no nuclear" and "no carbon capture and storage (CCS)" are introduced.

It seems counter-intuitive that "no CCS" has lower policy cost than the default mitigation scenario in the CGE stand-alone (Figure 4b). Why should policy mitigation costs decrease by removing technological options?

Why does the "no nuclear" scenario have lower policy costs than the default mitigation scenario in the integrated model (Figure 2b and 2c)? This seems counter-intuitive (but not impossible).

Line 168: I suggest that the estimates for industry and service sectors mitigating GDP loss rates by 0,40% and 0,50% should be commented also in view of the intercept estimated at 0.92% (and higher in Table S4).

Line 211: Is decarbonizing the energy system harmful to macroeconomic growth? Ultimately, we found that "this will not occur" if energy system information is appropriately reflected in the economic model. First: The authors mix model results and real-world matters. Second: "This will not occur" is imprecise.

Line 227: The authors state that "the elasticity parameter is normally assumed to be uniform, but it should differ among sectors, and probably regions". I may misunderstand the substitution elasticity of energy and value-added, but GTAP has sector specific elasticities and Koesler and Schymura (2012) estimates sector specific elasticities (Koesler, S. and M. Schymura (2012) "Substitution Elasticities in a CES Production Framework." Discussion Paper No 12-007).

Line 261: The meaning is unclear to me.

Line 274: "Because the discrepancy improvements were sufficiently small, we stopped the calculation after the second iteration." Was this the case for all 32 scenarios described in Table S3? Were the number of iterations predefined for the 32 scenarios, or was the convergence assessed in the same manner in all 32 scenarios – and all scenarios converged in two iterations? These questions matter with regards to the statistical analyses (ref Table 1 and Table S4).

Line 285: Gross domestic product (GDP) appears to be an input to the CGE model, and a GDP development is reported as an assumption in Figure S1. Still GDP losses are reported as results

from the CGE model. This seems contradictory, and should be clarified.

Line 287: Should the sentence read "One characteristic of OUR industrial classification is ..."?

Line 291: Incomplete reference? Reference 39 has three authors.

Line 321: ... considering the balances of electricity supply and demand in 3-h steps ... ?

Line 344: "... and increased capacity of demand ..." ?

Line 356: The sentence could be interpreted as if the energy demand is decided in AIM/Enduse, even though this model assumes energy service demand as exogenously provided. Is the point that AIM/endues provides energy consumption by energy types?

Line 419: Where is the set SJ defined?

Line 420-422: Is this description correct? Is α_j a dummy parameter representing a set of years and sectors? Is X_j the estimated variable?

Supplementary information

Line 1: Typo in the heading.

Line 27: "End1_CGE1" is not apparent in Table S2, there are two columns with heading CGE1_End1. Probably line 27 is correct and the table headings are wrong?

Line 58: What does "XXX results for each model" mean?

Line 69: You could explain the MER abbreviation (Market Exchange Rates) used in Figure S1b.

Line 72 Figure S2h: Why does the "no nuclear" scenario have baseline CO2 prices from 2025? Why are there no "no nuclear" mitigation CO2 prices? If the reported CO2 prices are mitigation prices, it seems counter-intuitive that they are lower in the "no nuclear" scenario than in the default scenario. Please clarify.

Comments to all reviewers

We would like to thank the referees for their remarks and helpful suggestions. They have helped us to further improve the quality of our manuscript. We have considered and responded to all referee comments. Based on the main arguments raised by the reviewers, we have made three main revisions to the manuscript and study.

- 1) We performed more iterations of model information exchanges, which we had previously repeated twice, but have now completed five times. We confirmed that the model variables converged sufficiently, as illustrated in the supplementary figures. We also changed the method slightly and updated all of the figures in the paper. However, the overall trends did not change, and the qualitative implications are the same as described in our previous manuscript.
- 2) We dropped the regression analysis of our simulation model outputs because two reviewers pointed out that the analysis was inadequate, and one of them suggested that we remove it. Instead, we described the macroeconomic impacts for all diagnostic scenarios and discussed the results. Basically, the qualitative conclusions are the same, but we added some discussion related to this.
- 3) We added substantial supplementary information to make our method as transparent as possible. Sectoral definitions, technological assumptions, and mathematical formulations with parameter assumptions are described.

Point-by-point responses to the referee comments are inserted below in blue.

Reviewer #1 (Remarks to the Author):

Summary and major claim of the paper

The paper aims to assess future economic costs of climate change mitigation. To this end, it devises a new integrated approach and applies it to Japanese 80% decarbonization pathways to 2050. The new integrated approach combines/soft-links an economic (top-down) CGE model with a (bottom-up) energy sector model and a technically detailed power system dispatch model. The central result states that economic costs of decarbonization – measured in percentage losses in GDP compared to a baseline without any decarbonization measures – are smaller when derived using the new integrated approach than when derived from a standard stand-alone economic CGE model. The reason for that would be the following: a standard CGE model employs a constant elasticity of substitution between the input factors energy and all others. This value is derived based on historical data and may not capture conceivable future developments in the energy sector. The new approach determines the elasticity of substitution endogenously and could better represent future changes in the energy system and their impact on the economic production process.

Thank you very much for your concise and comprehensive summary of this study.

Interest to others in the community and the wider field

The proposed approach is of high interest to the community and wider field. Ex-ante “evaluations” of the economic consequences of policy measures are of great relevance to the research community and policymakers when it comes to assess climate change mitigation measures. Also the identification of mechanisms that affect GDP loss estimates is timely and relevant.

Thank you very much for your positive assessment.

Overall evaluation

Overall, I like the idea of the paper. It touches a relevant point but suffers from various weaknesses. It leaves me somewhat puzzled.

- In a nutshell, too many points are opaque to assess both the novelty compared to the existing literature and the soundness of the proposed integrated model.
- There is also a range of conceptual issues, especially concerning model iteration and convergence as well as the “econometric” analysis
- Moreover, the external validity is highly doubtful as the application is driven by a number of

(partly unstated) assumptions. A focus on the internal validity and the methodological contribution how and why the soft-linking decreases GDP loss estimates would help to make the paper more credible.

The authors must devise good efforts to make the paper clearer in both methods and communication and satisfactorily address all major points. Only then, I can soundly judge the paper and potentially recommend publication. Please find my detailed review below.

Thank you very much for your overall evaluation. We will respond to each comment in detail below, but here we present a brief response to the above comments.

First, regarding opaqueness, we added substantial text and mathematical equations to make the approach as transparent as possible in this revision. Second, we made further iterations, instead of just two, and showed how the variables converged more clearly than before. Third, econometric regression analysis may not be the best option for our purposes, based on the reviewers' comments. Hence, we dropped these analyses. Fourth, the external validity cannot be proven exclusively based on this paper, although we have tried to present our methodology as clearly as possible. This approach will hopefully be tested by other modelling teams. Finally, we made substantial modifications to our explanation of our results, focusing on their internal validity and the mechanism of how the soft-linking approach affects the GDP loss compared to the conventional approach.

Major points

Concerning structure

The authors could should do a better job in coming to the core of the paper (as I understand it, the endogenization of the substitution elasticity between energy and other inputs in the economic production process by using an energy system model).

- The application to Japan is illustrative and should stay in the paper, but I see the contribution in the method. I highly doubt the external validity as numerical results are driven by numerous assumptions on functional forms and input data.

Thank you for pointing this out. We fully concur that the external validity depends on the functional forms and parameter assumptions. We have tried to make the assumptions and functional forms as transparent as possible in this revision. Meanwhile, we think that it will be difficult to prove the external validity within this paper because attempts by other modelling teams using the same (or similar) approach would be required. We added discussion related to this validity issue to the

corresponding section.

“For now, this study’s approach, and the derived implications are only applicable to Japan and within our modelling framework. We need further application fields and similar attempts from other teams, which can eventually show the external validity of our findings.”

- Likewise, the authors devote a lot of space to describing the differences between stand-alone CGE and integrated model. This can be shortened or longer parts relegated to an appendix. More space should be devoted to the central mechanism. The why is more relevant than the what
- Therefore, authors should streamline the paper and feature the central mechanism more prominently. In the current manuscript, it becomes clear only in lines 222 - 232

This is an excellent comment and we really appreciate it. The central mechanism for changing the macroeconomic implications is adjusting the productivity of the primary factors (labour and capital) which consist of a major part of value-added. This is because the total primary factor inputs are constrained exogenously each year. The reason why the GDP losses are lower in the integrated model than in the stand-alone model is that the primary factor productivity is higher in the integrated model than in the stand-alone model. Hence, the productivity shifts are mainly driven by two factors. One is the productivity decreases associated with emissions reductions in energy end-use sectors such as manufacturing, transport and service sectors (e.g. capital replacement by expensive but energy efficient ones). The other factor is sectoral allocation changes in primary factors.

For the first factor, Figure 5(a) illustrates the capital input efficiency of major industrial sectors (top 10 industries, which account for 95% of GDP in the base year) in the mitigation scenario compared to the baseline scenario for stand-alone and integrated models in 2050. Here, we define the capital input efficiency as capital input per output for each sector. Higher values indicate that additional capital inputs are needed in the mitigation scenario compared to the baseline scenario. In general, the stand-alone model requires larger capital inputs than the integrated model in the mitigation scenario. We can roughly compute to what extent the GDP losses are associated with these capital productivity losses by multiplying the value-added with the capital efficiency changes (Figure 5 (b, c)). They eventually account for 1.3 % of the total value-added (GDP). Hence, the productivity differences between the stand-alone and integrated model are mainly caused by the differences in the function form and parameters, particularly for the value-added and energy bundle. Here, we use a CES function wherein the substitution elasticity is 0.4 for the stand-alone model. The future autonomous energy efficiency is adopted. The integrated model uses a function of the same form, but the additional investment and energy inputs are provided by AIM/Enduse, by endogenizing the CES shift parameters (sector-wise additional investments are shown in Supplementary Table 9). The second factor, effects of sectoral primary factor allocation change, is mainly driven by the power

generation sector, where the electricity generation in the mitigation scenario compared to the baseline scenario is 20% higher in the stand-alone model, whereas it is almost same in the integrated model (Figure 5). There are certainly differences in power technological shares between the stand-alone and integrated models, but, in summary, it seems that the difference in total electricity generation between the models is the dominant factor. The stand-alone model requires incremental capital and labour inputs that account for 0.4% of GDP compared to the integrated model (Figure 5 (d)). The total electricity demand is determined by the energy consumption represented in the energy end-use sectors. The total energy consumption in each energy end-use sector is represented by a CES function, as mentioned above, and the fuel-wise share is determined by a logit function in both the stand-alone and integrated models. The share parameters in the logit function are endogenously determined by the integrated model obtained from the AIM/Enduse results, whereas they are exogenous parameters in the stand-alone model.

In addition to the two main mechanisms mentioned above, productivity changes and sectoral shifts in other sectors certainly occur, but are relatively minor.

In summary, we added one section to the results to discuss the points raised above and described the corresponding equations and parameter assumptions in more detail in the Supplementary Notes.

“Mechanism causing the differences in macroeconomic implications between the stand-alone and integrated models

The central mechanisms for changing the macroeconomic implications are changes to the productivity of primary factors (labour and capital) which constitute value-added. This is because the primary factor inputs are constrained exogenously for each year in our economic model. The reason that GDP losses are lower in the integrated model than in the stand-alone model is that the primary factor productivity is higher in the integrated model than in the stand-alone model. Hence, the differences in productivity are mainly driven by two factors. One is the productivity decreases associated with emissions reductions in energy end-use sectors, such as industry, transport and service sectors (e.g. capital replacement by expensive but energy efficient ones). The other factor is sectoral allocation changes in primary factors.

In the case of the first factor, Figure 5(a) illustrates the capital input efficiency of major industrial sectors (top 10 industries, which account for 95% of GDP in base year) in the mitigation scenario compared to the baseline scenario for stand-alone and integrated models in 2050. Here, we define the capital input efficiency as capital input per output for each sector. Higher values indicate that additional capital inputs are needed in the mitigation scenario compared to the baseline scenario. In general, the stand-alone model requires larger capital inputs than the integrated model in the mitigation scenario. We can roughly compute the value-added losses associated with these capital productivity losses by multiplying the value-added of each sector (Figure 5 (b, c)). These eventually

account for 1.3 % of the total value-added (GDP). Then, the productivity differences between the stand-alone and integrated model are mainly caused by differences in the functional form and parameters particular to the value-added and energy bundle. Here, we use a CES function in which the substitution elasticity, share parameters and future autonomous energy efficiency are defined in the stand-alone model. The integrated model uses a function of the same form, but the additional investment and energy inputs are exogenously given by AIM/Enduse, whereas the CES shift parameters are determined endogenously (sector-wise additional investments are shown in Supplementary Table 9).

The second factor, namely the effect of sectoral primary factor allocation change, is mainly driven by the power generation sector. The electricity generation in the mitigation scenario compared to the baseline scenario is about 20% higher in the stand-alone model, but almost the same in the integrated model (Figure 5). There are certainly differences in technological shares between the stand-alone and integrated models, but, in summary, it seems that the difference in total electricity generation between the models is the dominant factor, where the stand-alone model requires additional capital and labour inputs, accounting for 0.4% of GDP, relative to the integrated model (Figure 5(d)). With respect to the representation of electricity demand, the total electricity demand is determined by energy consumption in the energy end-use sectors, which are represented by a CES function, as mentioned above. The fuel-wise share is determined using a logit function in both the stand-alone and integrated models. A parameter representing the preferences or technological choices in the logit function is determined endogenously in the integrated model, based on the AIM/Enduse results, whereas they are exogenous parameters in the stand-alone model. We describe the detailed mathematical formation and assumptions in the Supplementary information.

In addition to the two main mechanisms mentioned above, the productivity changes and sectoral shifts in other sectors certainly occur, but are relatively minor.”

Figure 3. Valued-added differences between baseline and mitigation scenarios. Panel **a** illustrates sectoral capital input efficiency for the top 10 industrial activities in 2050. Panel **b** shows sectoral value-added share in the baseline scenario for the top 10 industrial activities. Panel **c** represents the capital productivity value-added effects compared to the total value-added for the top 10 industrial activities. Panel **d** illustrates the value-added share of power sectors in terms of total economy-wide value-added.

The mathematical description and parameter assumptions below are now included in the Supplementary Information.

3. Mathematical formula of CGE model

In this section, we describe the mathematical formula in the AIM/CGE model, which is particularly relevant to energy consumption and power generation.

3.1. Energy end-use sectors other than household sector in the CGE model

We begin with the representation of the energy end-use sectors (Table S5). The value-added and energy composite inputs are determined by multiplying a coefficient by the output from the energy end-use sectors (Equation 1). Then, value-added and energy composite are combined with the CES

function (Equation 2). Labour and capital inputs are further nested in the CES function, as shown in Equation 3.

$$QVAE_a^B = ivae_a \cdot QA_a^B \quad (1)$$

$$QVAE_a^B = \alpha ae_a (\beta ae_a \cdot QVA_a^B^{-\rho ae_a} + (1 - \beta ae_a) \cdot (at_a \cdot QENE_a^B)^{-\rho ae_a})^{-\frac{1}{\rho ae_a}} \quad (2)$$

$$QVA_a^B = tfp \cdot \alpha f_a (\beta f_a \cdot QF_{capital,a}^B^{-\rho f_a} + (1 - \beta f_a) \cdot QF_{labor,a}^B)^{-\frac{1}{\rho f_a}} \quad (3)$$

where

aCE is a set of production activities,

QA_a^B is an output of sector a in the baseline scenario,

$QVAE_a^B$ is the composite of the value-added and energy of sector a in the baseline scenario,

QVA_a^B is the value-added of sector a in the baseline scenario,

$QENE_a^B$ is the energy input of sector a in the baseline scenario,

$QF_{f,a}^B$ is the primary factor input of factor f and sector a in baseline scenarios (f = capital or labour),

$ivae_a$ is an input coefficient of the output of sector a ,

αae_a is the scale parameter of the CES function for value-added and energy aggregates,

βae_a is the share parameter of the CES function for value-added and energy aggregates,

ρae_a is the exponent parameter of the CES function for value-added and energy aggregates,

αf_a is the scale parameter of the CES function for primary factor aggregates,

βf_a is the share parameter of the CES function for primary factor aggregates,

ρf_a is the exponent parameter of the CES function for primary factor aggregates,

at_a is an autonomous energy efficiency improvement parameter, and

tfp is a parameter that represents the economy-wide total factor productivity, which is calibrated in the baseline scenarios by hitting the target GDP. The calibrated values are adopted in the mitigation scenarios.

Energy consumption by fuel type in energy end-use sectors is determined by logit sharing, as follows:

$$SHENE_{c,a} = \frac{ae_c \cdot \delta_{c,a}^{en} \cdot PQ_{c,a}^{-\beta^{el}}}{\sum_{c \in CEENE} ae_{cp} \cdot \delta_{cp,a}^{en} \cdot PQ_{cp,a}^{-\beta^{el}}} \quad c \in CEENE, a \in AEnd \quad (4)$$

Where

$aCEEnd$ is a subset of the production activity and energy end-use production activity (e.g. industry, transport and so on),

$cCEENE$ is set of energy commodities (coal, petroleum products and so on), which is a subset of all commodities,

$SHENE_{c,a}$ is the energy consumption share of energy commodity c and production activity a ,

$PQ_{c,a}$ is the price of commodity c in production activity a ,

$\delta_{c,a}^{en}$ and β^{el} are parameters for logit selection, and

ae_c is the fuel-wise energy preference change parameter of commodity c .

For the stand-alone model, βae_a^B and $\delta_{c,a}$ are calibrated by base year information. Autonomous Energy Efficiency Improvement (AEEI) in the stand-alone model at_a is one of the critical parameters determining energy consumption. We adopted a uniform AEEI across energy end-use sectors for each year, which is associated with GDP growth. For the years that assume more than 1% of GDP annual growth, the AEEI is 1%, and half of the annual GDP growth rates are assumed for the other the years. The fuel-wise energy preference change parameter ae_c is set annually to 1%, 0.5% and -0.5% for electricity, gas and coal, respectively, which represent the fuel shift from conventional solid and liquids to gas and electricity carriers.

With respect to the integrated model, the $SHENE_{c,a}$ and $QENE_a^B$ are fixed based on the AIM/Enduse model results and endogenise $\delta_{c,a}^{en}$ and at_a .

Then, we can derive the input coefficients of capital in the baseline scenarios after obtaining the simulation results of the baseline scenarios, as shown in Equation (5). For the mitigation scenarios in the integrated model, the derived input coefficients of the primary factors shown above are used to estimate primary factor inputs. For the capital inputs, additional investment costs associated with mitigation policy, which is given by the AIM/Enduse model, are added to the baseline inputs, as shown in Equation (6).

$$ifa_{f,a}^B = QF_{f,a}^B / QA_a^B \quad f \in Fcap \quad (5)$$

$$QF_{f,a}^M = ifa_{f,a}^B * QA_a^M + AddInv_{f,a} \quad f \in Fcap \quad (6)$$

where

$fCEcap$ is a set of capital and a subset of primary factors

ifa_a^B is an input coefficient of primary factor f and sector a in baseline scenarios.

$AddInv_{f,a}$ is the additional investment cost associated with mitigation costs provided by AIM/Enduse,

$QF_{f,a}^M$ is the primary factor input of factor f and sector a in the mitigation scenarios (f = capital or labour), and

QA_a^M is an output of sector a in mitigation scenarios.

3.2. Power generation in the CGE model

The total consumption of electricity is determined by the demand side representation shown in the previous subsection (and the latter for household). Then, the power generation is determined based on logit sharing, as below.

$$SHAC_{c,a} = \frac{\delta_{c,ap}^{el} \cdot PXAC_{c,a}^{-\beta^{el}}}{\sum_{ap \in AEly} \delta_{c,ap}^{el} \cdot PXAC_{c,ap}^{-\beta^{el}}} \quad c \in CEly, a \in AEly \quad (7)$$

where $aCEly$ is a subset of production activity and electricity production activity (e.g. coal, solar

PV and so on),

$cCEELY$ is a subset of commodity (electricity),

$SHAC_{c,a}$ is the electricity generation share of production activity a ,

$PXAC_{c,a}$ is the price of commodity c produced by production activity a ,

$\delta_{c,a}^{el}$ and β^{el} are parameters for the logit selection of power general technologies.

For the stand-alone model, $\delta_{c,a}^{el}$ is calibrated using base year information. The integrated model fixes the $SHAC_{c,a}$ as the AIM/Enduse model results and endogenises $\delta_{c,a}^{el}$. This treatment is the same between the baseline and mitigation scenarios. As the absolute amount of power generation is determined by the demand side, here we specify only the share. The transmission losses are also considered. Battery capacity is input as an absolute amount.

3.3. Household consumption in CGE model

Household consumption is formulated using a LES function and further nested in a CES function for energy and other manufacturing goods. Then, the total energy consumption is finally split out into fuel-wise consumption using a logit function

$$QCH_{ch} = \mu_{ch} + \theta_{ch} \cdot \left(\frac{EH}{PCH_{ch}} - \sum_{chp \in CH} \mu_{chp} \right) \quad ch \in CH \quad (8)$$

$$QCH_{ch} = \alpha h_{ch} \left(\beta h_{ch} \cdot QHE_{ch}^{-\rho h_{ch}} + (1 - \beta h_{ch}) \cdot QHM_{ch}^{-\rho h_{cp}} \right)^{\frac{1}{-\rho h_{ch}}} \quad (9)$$

$$SHHENE_{c,ch} = \frac{ae_c \cdot \delta_{c,ch}^h \cdot PQ_{c,h}^{-\beta^h}}{\sum_{cp \in CENE} ae_{cp} \cdot \delta_{cp,ch}^h \cdot PQ_{cp,h}^{-\beta^h}} \quad c \in CENE, ch \in CHE \quad (10)$$

$$QH_c = \begin{cases} \sum_{(c,ch) \in mapCCH} QCH_{ch} & c \in CNENE \\ \sum_{(c,ch) \in mapCCH} QHM_{ch} & c \in COMF \end{cases} \quad (11)$$

where

$chCEH$ is a set of household consumption goods, of which mappings with goods c are shown in Table S10,

$chCEHE$ is a set of energy-related household consumption goods (car usage and other energy-related consumption)

$(c, ch) \in mapCCH$ is a mapping from household consumption goods ch to general goods c , shown in **Error! Reference source not found.**,

PCH_{ch} and QCH_{ch} are the quantity and price of household consumption goods ch ,

$SHHENE_{c,ch}$ is the share of energy fuel c of household consumption goods ch ,

QHE_{ch} and QHM_{ch} are the quantity of energy and other manufacturing goods, respectively, for household consumption goods ch (car usage and other energy-related consumption),

θ_{ch} and μ_{ch} are LES function parameters,

α_{ch} is the scale parameter of the CES function for energy and other manufacturing goods aggregates,

β_{ch} is the share parameter of the CES function for energy and other manufacturing goods aggregates,

ρ_{ch} is the exponent parameter of the CES function for energy and other manufacturing goods aggregates,

$\delta_{c,ch}^h$ and β^h are parameters for logit selection of household energy fuel.

Household LES function parameters in the stand-alone model are updated recursively based on income elasticity. Electricity and biofuel used in transport are not accounted for in the base year social accounting matrix and, thus, we introduce the initial parameters for $\delta_{c,ch}^h$ and $\delta_{c,a}^{en}$ by calibrating 0.1 % of the share in each energy consumption in 2015 and 2020. They are then updated afterwards, to one-third of the value of petroleum products in 40 years. For the integrated model, the parameters β_{ch} and ρ_{ch} were determined endogenously based on the energy consumption and investment needs for other manufacturing goods computed by AIM/Enduse.

Concerning analysis

The exposition of the models is opaque and must be considerably improved. The model is highly complex but under-explained. It is difficult to deeply understand what drives the results. Please apologize that I go into details here, but all details together constitute a major issue.

- Overall language must be more precise. Especially, central concepts must have a clear and immutable name. E.g.

We highly appreciate you pointing this out. We have amended the text accordingly and read through the entire manuscript once again.

o “value-added”, “energy service demand”, ...

We use the terms “value-added” and “energy service demand” immutably rather than using “value added” and “service demand”

- CGE and energy system model:

o Clearly state which sectors you analyze. I was puzzled throughout the paper

- CGE model

o The description must be clearer. I have some experiences with CGE models, but lines 292 to 310 are opaque

→ Which production sectors? Is there a final consumption good or multiple?

→ Which energy transformation sectors? What do you mean here by “value-added” as an input? Do

you mean capital?

→ Which energy end-use sectors? What do you mean here by “value-added” as an input?

We have added a list of production sectors to the Supporting Information, which we quote here. We also clarified the meaning of value-added.

→ Please explain the concepts of energy and material balances in power generation more clearly!

→ Lines 299-300: Which share do you mean? Which calibrated information do you mean?

Our intended meaning was the share of each technological power generation method out of the total electricity generation. We have clarified this point.

- Energy system model Enduse

o Clearly state the scope of the model!

o Line 314: Be more specific on energy end use and supply sectors! Are they the same as in the CGE? Are they a disaggregated representation of the sectors in the CGE?

Regarding the scope and granularity of sectors in AIM/Enduse, we added the following sentences.

“The model covers energy-related GHG emissions from both energy end-use and energy supply sectors. The end-use sectors are composed of industry, buildings and transportation sectors, and they are disaggregated into several subsectors with respect to types of products, buildings, and transportation mode based on the IEA energy balances.”

As the difference between the sectoral granularity of AIM/Enduse and AIM/CGE should be elaborated in the data exchange section rather than in each model description, we added the corresponding sentences to the “Information from AIM/Enduse provided to AIM/CGE” section.

“Because the sectoral disaggregation of AIM/Enduse basically complies with the IEA energy balance, there are inconsistencies in the AIM/CGE, which is based on an input-output table. Thus, in terms of data exchange from AIM/CGE to AIM/Enduse, the subsectors are aggregated so that the granularity of the sectors is in agreement. Nevertheless, given the large share of industrial GHG emissions in Japan’s long-term low carbon scenarios, iron, chemical, paper, non-metallic minerals, and non-ferrous metals are exempted from the sector aggregation.”

o Line 318: Why are energy prices a constraint in the model? Which energy do you mean? Electricity? In my understanding of an energy system model, they are a result.

As Japan’s energy supply largely depends on imported fossil fuels, primary energy prices must be

defined exogenously in this model. We edited the text as follows.

“Mitigation options are selected based on linear programming to minimize total energy system costs that include investments for mitigation options and energy costs subject to exogenous parameters such as cost and efficiency of technology, primary energy prices, energy service demands and emission constraints.”

o Line 325: Which energy-demanding sectors? Those in Table S3?

Yes, it is consistent with those in Table S3. However, we avoid using this term to clarify the model description. We have edited the text as shown below.

“In the industry, building and transportation sectors, wide mitigation options are included, such as energy-efficient devices and fuel switching.”

- Power dispatch model

o Clearly state whether investment decisions are part of that model! It seems not, but in line 332 you call it a generation expansion model

As for the second sentence of your comment, unfortunately, we could not find the words “generation expansion model”. We used the term “generation planning model” on line 332. We think that the term “generation planning” can be used for planning operation schedules, frequency management, and storage management, which are modelled in AIM/Power. Thus, we did not modify the term used on line 332.

As for the first sentence of your comment, we added the following description of the investment decisions made by the power dispatch model in this study after the first paragraph of the section on the AIM/Power model.

“Note that as AIM/Enduse provides power generation installed capacity for AIM/Power, AIM/Power does not make investment decisions, except for making additional investments in storage and power plants aimed at hourly and within hourly power demand-supply management”

- Lines 336-337: What do you mean by “or both”? Is it possible that the model satisfies either only electricity demand or only emission reduction targets? This appears strange to me.

Thank you very much for your comment. It was a misleading sentence. It has been modified as follows.

“several constraints, including satisfying electricity demand and CO₂ emission reduction targets”.

- Also explain the difference in the scopes of the power dispatch and energy system models more

clearly! What do they cover?

Thank you very much for your comment. The following sentence has been added.

“In other words, unlike the AIM/Enduse model covering all energy-related sectors, the AIM/Power model only covers the power generation sector.”

The explanation of the communication between models is insatisfactory and must be improved.

- Exchange from Enduse to CGE

o You state that both AIM/Enduse and AIM/Power decide on battery investments. Please clarify!

We stated that AIM/Power determines the capacity factors of the batteries, and that capacity is determined by AIM/Enduse. Then, CGE incorporates both data. We have clarified this point.

o Line 367: What do you mean by “baseline case”? What do you mean by the implicit assumption “that the baseline investment cost is inherent in the CES substitution”?

We meant to say that the energy investment of energy end-use sectors in the baseline scenario is modelled by a typical CES nested function. We have modified the text to clarify this point.

o Line 368: What do you exactly mean when you say that “capital input coefficients are fixed at baseline levels”? I think that this is the central point of the paper where you replace the exogenously assumed substitution elasticity by results from the energy system model. If so, please explain way more clearly and understandably. This is at the heart of the contribution of the paper!

Thank you. We think that it would be better to use a mathematical formulation to provide a precise explanation. We have added this to the supplementary information, as mentioned above.

- Exchange from AIM/CGE to AIM/Enduse

o Line 382: What do you mean by “industry service demand”?

Thank you very much for pointing this out. We added a more detailed explanation, as indicated below.

“The energy service demand in the industrial sectors, such as steel and cement production, and outputs of other industrial sectors, is altered by the outputs from AIM/CGE.”

o Lines 384-386: Tentative guess of substitution elasticities being unity based on some Swedish econometric analysis is doubtful. Please more explanation here or argumentation why this is okay.

Thank you very much for raising this important point. We think that not asserting this tentative

assumption is okay because the elasticity between monetary and physical units is uncertain. However, we believe that the impact of this uncertainty on the major findings of this paper is negligible, because the GDP losses estimated in this paper are low enough (3% at maximum) in the CGE stand-alone model. To test the uncertainty in the elasticity, we ran scenarios in which we varied the elasticity from 0.5 to 2.0. Our results indicate that the elasticity assumption affects the numbers, but the same qualitative conclusion holds. We have edited the sentences as below.

“According to the Swedish econometric analysis⁴⁸, elasticity between monetary and physical units of energy services can be assumed to be approximately 1.0. Furthermore, the GDP losses indicated in this study are relatively small, less than 3 %, in the CGE stand-alone model, as shown in Figure 4a. Thus, we tentatively applied an elasticity value of 1.0. Meanwhile, we varied the elasticity from 0.5 to 2.0 and observed that the policy costs change slightly, but the qualitative conclusion still holds (Supplementary Figure 5).”

Supplementary Figure 5. Climate mitigation policy costs associated with variations in the elasticity in monetary outputs and physical energy service demand.

- Exchange from AIM/Enduse to AIM/Power and vice versa

o Please clarify in which model you determine which battery capacities!

Although we calculated the battery capacities in both AIM/Enduse and AIM/Power, we used the results from AIM/Enduse in our analysis. This is because the decision to invest in batteries for long-term fluctuations is part of capacity planning, as are investment decisions relating to pumped hydro power plants. Battery capacities for short-term fluctuations were determined by AIM/Power, which can model hourly electricity demand-supply balances.

The following sentence was added to the last section of “Information from AIM/Power provided to AIM/Enduse”.

“Note that generation capacity, although not directly related to balancing short-term fluctuations, is decided by AIM/Enduse. Thus, the battery capacity is determined by AIM/Enduse when considering long-term electricity fluctuations, whereas that for short-term fluctuations is provided by AIM/Power.”

o Please clarify the importance of the capacity factors! If you have a three-hour granularity in the Enduse model, why are they important (for investment decisions)?

Even AIM/Enduse has a three-hour granularity, and this time resolution is insufficient for assessing demand-supply balances for electricity (and this is the reason why we incorporated AIM/Power in the model integration framework). Curtailment is one effective method for increasing the time resolution of the model, because the curtailment ratio directly affects the capacity factor of each generation plant. Specifically, X% of curtailment ratio is equivalent to an X% reduction in the capacity factor, which is a crucial factor for investment decisions.

To clarify these points, we added the following sentence.

“Moreover, the power system would respond to large-scale renewable energy installations by adjusting the capacity factor for conventional power generation systems (e.g. coal-fired power) in addition to curtailing the output from variable renewables. These measures for balancing short-term fluctuations reduce the electricity output per installed capacity, and thus affect investment decisions. It is necessary to consider this feedback from AIM/Power to AIM/Enduse.”

- Iteration and convergence

o Line SI25: How do you apply an RMSE? If you compare two values (for the same variable in the two models), then the concept is not applicable. If you compare more than two values, please clarify!

o Table S2: In any case, the table is difficult to understand, please explain more clearly what the percentage numbers mean!

- o Table S2: I do not see convergence. Basically, results for almost all variables differ by a large degree between CGE and Enduse, and only the magnitude of the difference stays roughly constant after the second iteration. In my understanding, convergence must imply that variables take on identical values between models.
- o Related to that, two iterations seem unrealistically few.
- o Please also comment on the computational burden
- o You state that “the absolute value of energy consumption is not fully harmonized between” the CGE and energy system model (line 362). Do you refer to Table S4 here? If no, explain. If yes, this seems problematic for me.

Regarding the iterations, there are three points for discussion here. First, the definition of convergence depends on the study, and within this study, we define it as the iteration point where differences between models stop improving. Technically speaking, we cannot force all AIM/CGE variables into AIM/Enduse. At the same time, we reconsidered the method for inputting AIM/Enduse data into AIM/CGE and it turned out that we could use a slightly different approach, in which the absolute final energy consumption information is input directly (previously, the fuel share was input). The differences between AIM/Enduse and AIM/CGE were much smaller than in the previous approach, and we updated the description of our method accordingly. Non-energy use and LPG accounts are post-processed and agricultural energy is not fully harmonized. Consequently, small gaps remain, but these are small enough to stop iterating.

Second, as the differences would not improve, it would not be meaningful to iterate further. However, we did carry out further iterations, up to five, and present all of our results. The results confirm that most variables converge after two iterations. We did this for the default case as well as the no CCS and no nuclear cases for the baseline and mitigation scenarios, respectively. In the Supplementary Information, we show all comprehensive indicators by iteration for each scenario. In this response letter, we selected two examples, namely the 2050 baseline and mitigation scenarios, for default technological assumptions (see Supplementary Information for more scenarios and years).

Third, we admit that the root mean square error was not the exact metric used, and we define the error indicator below. Here, we regard the AIM/Enduse and AIM/CGE information as the true and estimated values, respectively. Although there is no true value for the future, we think that the metric itself is valuable. According to the comment, we changed the name of that indicator.

The indicator shown below is adopted. i , t and s are sets of variables of model (e.g. energy demand), years and scenarios, respectively. $X_{i,t,s}$ and $Y_{i,t,s}$ are AIM/CGE and AIM/Enduse outputs, respectively.

$$\text{Error indicator}_{i,s} = \sqrt{\frac{\sum_t (X_{t,i,s} - Y_{t,i,s})^2}{\left[\sum_t \left(\frac{X_{t,i,s} + Y_{t,i,s}}{2} \right) \right]^2}}$$

Figure S8. Main energy, emissions and economic indicators of the baseline scenario in 2050 by iteration. Each panel illustrates the individual variables, the codes and units of which are listed in Table S5.

Figure S9. Main energy, emissions and economic indicators in the mitigation scenario in 2050 by iteration. Each panel illustrates individual variables, the codes and units of which are listed in Table S5.

Table S1. List of variables, codes and units.

Code	Variable	Unit
Fin_Ene	Final Energy	EJ/yr
Prc_Car	Price Carbon	US\$2005/t CO2
Prm_Ene	Primary Energy	EJ/yr
Prm_Ene_Coa	Primary Energy Coal	EJ/yr
Prm_Ene_Fos	Primary Energy Fossil Fuel	EJ/yr
Prm_Ene_Gas	Primary Energy Gas	EJ/yr
Prm_Ene_Nuc	Primary Energy Nuclear	EJ/yr
Prm_Ene_Oil	Primary Energy Oil	EJ/yr
Sec_Ene_Ele	Secondary Energy Electricity	EJ/yr
Sec_Ene_Ele_Bio	Secondary Energy Electricity Biomass	EJ/yr
Sec_Ene_Ele_Coa	Secondary Energy Electricity Coal	EJ/yr
Sec_Ene_Ele_Gas	Secondary Energy Electricity Gas	EJ/yr
Sec_Ene_Ele_Nuc	Secondary Energy Electricity Nuclear	EJ/yr
Sec_Ene_Ele_Oil	Secondary Energy Electricity Oil	EJ/yr
Sec_Ene_Liq	Secondary Energy Liquids	EJ/yr
Fin_Ene_Ele	Final Energy Electricity	EJ/yr
Fin_Ene_Gas	Final Energy Gases	EJ/yr
Fin_Ene_Ind	Final Energy Industry	EJ/yr
Fin_Ene_Liq	Final Energy Liquids	EJ/yr
Fin_Ene_Res_and_Com	Final Energy Residential and Commercial	EJ/yr
Fin_Ene_Solids	Final Energy Solids	EJ/yr
Fin_Ene_Tra	Final Energy Transportation	EJ/yr
Prm_Ene_Hyd	Primary Energy Hydro	EJ/yr
Prm_Ene_Solar	Primary Energy Solar	EJ/yr
Prm_Ene_Win	Primary Energy Wind	EJ/yr
Sec_Ene	Secondary Energy	EJ/yr
Fin_Ene_Res	Final Energy Residential	EJ/yr
Fin_Ene_Com	Final Energy Commercial	EJ/yr
Emi_CO2_Ene	Emissions CO2 Energy	Mt CO2/yr

Concerning novelty

As the analysis has many unclear points, it is hard to judge genuine novelty. The literature features a

range of attempts to combine top-down CGE with bottom-up energy system models; among others:

- The Bohringer-Rutherford stream (e.g., Energy Economics 30 (2008), 574-596, Journal of Economic Dynamics and Control 33(9) (2009), 1648-1661 and further papers by the same author)
- Tuladhar et al. (Energy Economics 31(2), (2009), S223-S234)
- Abrell and Rausch (Journal of Environmental Economics and Management 79 (2016), 87-113)
- Tapia-Ahumada et al. (Economic Modelling 51 (2015), 242-262)
- Hwang and Lee (Energy Policy 83 (2015), 69-81)
- The IAM community (e.g., REMIND (<https://www.pik-potsdam.de/research/sustainable-solutions/models/remind/remindequations.pdf>))

Several of the above papers also explicitly analyze macro-economic effects (Tuladhar, Abrell and Rausch, Hwang and Lee). I did not carry out an exhaustive search, but there are, confidently, more.

We deeply appreciate you pointing this out. We agree that some references were missing from the previous manuscript and we have, therefore, made a table exhaustively listing studies linked to multi-sectoral CGE and energy system models in the Supporting Information. The above literature pointed out by the reviewer is either included in the list or discussed within the main text.

Meanwhile, we would like to remind the reviewers that the novelty of this paper is that the policy cost estimates are smaller than previously thought. We do think that this has not been discussed before, as we clearly compared the conventional stand-alone CGE and integrated models. To clarify this point, we modified the text as below.

“we identify the magnitude of the differences in macroeconomic costs for climate change mitigation using values derived from this newly integrated model and the conventional economic model approach, and determine which sector’s representation is an influential factor. This is the novelty of this study.”

As the exposition of the model in this manuscript is shallow and not sufficiently clear, it is hard to judge whether its contribution is sufficiently novel. While energy system-macro model integration is not entirely new, I would judge the integration of two highly detailed models sufficient.

Likewise, the observation that CGE and energy system models differ in results concerning their GDP impact is known, as the authors state. The new insight of this paper would be why they differ. The explanation given is plausible (“kind of” endogenous substitution elasticity), but does not become sufficiently clear from the analysis. This is a main point the authors must address.

Thank you very much. We fully concur. As mentioned above, the key message of this paper is that the macroeconomic implications differ between the integrated and standalone models rather than

methodological advances, although the three models (CGE, energy system and power dispatch model integration) could be new.

As suggested above, we significantly modified and supplemented the text, as described above, to explain why they differ.

Further major points

Diagnostic scenarios

As such, a good idea, but the regression analysis is awkward (Tables 1 and S4). Drop it.

- You have a small sample (N=32 in Table S4). Estimators cannot be assumed to be distributed normally. Did you account for that?
- Regression analysis is an appropriate method to estimate population effects from samples. Your data is deterministic. You know the statistical population. There is no need for regression analysis.
- You can just compare means. Take the mean of the GDP result of all scenarios with a certain feature (16 when for a single year) and the mean of the GDP result without this feature and compare.
- It would also be interesting to see whether the detected effect (e.g. representation of industry is relevant) is more pronounced in combination with other features. You could populate a Table like S3 with your GDP results. This would be more insightful
- All tables must state the sample size and the R-squared
- Line 175: as far as I see you do not have a fixed effect (=time-constant unobserved heterogeneity in the cross-sections)
- In any regression, you estimate parameters with given data for variables; adjust terminology

Based on the suggestion that there is no need for regression analysis, we decided to remove it. It was also recommended that we show a table like S3 with GDP changes, and the corresponding table has been added. We discussed the table and, although it may not be easy to say deterministically that industry and service sector energy use representation is the main driver affecting GDPs, we observed this tendency.

“We ran further diagnostic scenarios with and without incorporating energy system information by sector (see Methods for more details) to investigate the extent to which the energy system model’s output information for each sector contributes to mitigation cost differences compared to the stand-alone CGE. Comparing scenarios that include a single sector’s information from AIM/Enduse and the stand-alone model (yellows and red in Table S, respectively), the inclusion of the industry and service sector information from AIM/Enduse makes a remarkable difference in the GDP loss rate (Scenarios 9 and 5 in Table S, respectively). From the opposite side, the scenarios taking out the

AIM/Enduse information for each sector (green in Table S) show that excluding the industry and service sectors similarly generates GDP loss differences compared to the integrated model (scenario 32 in Table S). Conversely, the incorporation of residential, transport and energy supply sector information given by AIM/Enduse has a small impact on GDP losses, or even has the opposite effect in some cases. Further cross sectoral diagnostic scenarios are shown in Table S4. However, the overall insights are clear, that the industry and service sectors are key in determining macroeconomic implications.

Table S4. Full list of diagnostic scenarios and their GDP loss rates in 2050. Column names are sectors, and 'on' and 'off' refer to whether AIM/Enduse information is incorporated. The red and blue rows indicate the stand-alone and integrated models, respectively. Yellow and green rows indicate scenarios that include and exclude information from a single sector given by AIM/Enduse, respectively.

	Energy Supply	Industry	Service	Transport	Residential	GDP loss rate (%)	
						2030	2050
scenario 1	off	off	off	off	off	1.1	2.4
scenario 2	off	off	off	off	on	0.9	2.3
scenario 3	off	off	off	on	off	1.1	2.4
scenario 4	off	off	off	on	on	1.0	2.3
scenario 5	off	off	on	off	off	0.6	1.7
scenario 6	off	off	on	off	on	0.5	1.5
scenario 7	off	off	on	on	off	0.7	1.6
scenario 8	off	off	on	on	on	0.6	1.5
scenario 9	off	on	off	off	off	0.4	0.8
scenario 10	off	on	off	off	on	0.3	0.6
scenario 11	off	on	off	on	off	0.5	0.7
scenario 12	off	on	off	on	on	0.4	0.6
scenario 13	off	on	on	off	off	0.1	0.4
scenario 14	off	on	on	off	on	0.0	0.3
scenario 15	off	on	on	on	off	0.2	0.3
scenario 16	off	on	on	on	on	0.1	0.2
scenario 17	on	off	off	off	off	0.9	2.2
scenario 18	on	off	off	off	on	0.9	2.3
scenario 19	on	off	off	on	off	1.1	2.4
scenario 20	on	off	off	on	on	0.9	2.5

scenario 21	on	off	on	off	off	0.5	2.0
scenario 22	on	off	on	off	on	0.4	1.8
scenario 23	on	off	on	on	off	0.7	2.3
scenario 24	on	off	on	on	on	0.5	2.2
scenario 25	on	on	off	off	off	0.4	1.2
scenario 26	on	on	off	off	on	0.3	1.0
scenario 27	on	on	off	on	off	0.6	1.3
scenario 28	on	on	off	on	on	0.4	1.2
scenario 29	on	on	on	off	off	0.0	0.8
scenario 30	on	on	on	off	on	-0.1	0.6
scenario 31	on	on	on	on	off	0.1	0.8
scenario 32	on	on	on	on	on	0.0	0.8

”

Throughout the paper, do not call it mitigation of GDP loss. Neither results from the CGE model nor results from the integrated model are the truth. Please consistently call it something like difference or wedge between results.

Agreed. We have changed this throughout the manuscript.

Line 166: It is unclear what you mean by “with and without incorporating energy system information by sector”. Do you switch off the entire sector in Enduse or do you suppress transferring the information to CGE?

Thank you very much for pointing this out. We should have specified these things. We meant to say that the information obtained from AIM/Enduse includes data on all sectors, but here we generated scenarios that partially incorporate such information. For example, one scenario only uses the transport energy-related information provided by AIM/Enduse, and the rest of the sector’s model representation was maintained as in the conventional CGE, while another used residential and industrial energy information. We clarified this point by changing the text as below.

“we ran diagnostic scenarios with and without incorporating energy system information generated by AIM/Enduse by sector (e.g. one scenario only uses data on the transport sector’s energy use obtained from AIM/Enduse)”

Lines 221-235: You clearly conclude that it is the industry and service sectors whose representation

causes model results to be different. The question remains why. You mention that it is caused by the production function form and its parameters. Again, this central point must become crystal clear.

This point has already been discussed above. We would like to refer you to it.

Model

How do you implement the temporal structure?

- Do you solve each model for 2010 until convergence and then proceed to 2015, 2020 and so on? Or do you solve Enduse for 2010 – 2050 and then hand over to CGE and iterate?

Thank you for pointing this out. We solved each model for the entire period and then passed the model outputs to the other models. We have added a description to this effect.

“The individual models were solved from 2010 to 2050, then the results from each were input to the other models.”

- Please be more specific on your CO₂ budget. Is it one budget for the entire 2015-2050 period so that banking and borrowing, e.g., delayed decarbonization is possible?

Our approach is recursive dynamic and there is only an annual emissions constraint. We modified the corresponding text as follows.

“The CO₂ emissions constraint is assumed for every simulation year of the AIM/Enduse model under the mitigation scenario.”

- Please be more specific on (social) discounting

There is no social discounting rate because our models are solved as recursive dynamic. To compute the annualized cost, we use the discount rate for individual energy use devices. We added the following detail to the supporting information and methods.

“As the models used in this study were solved as recursive dynamic, we did not consider discounting of the energy system costs. Nevertheless, the AIM/Enduse model annualizes the capital costs of energy technologies using a discount rate in the range 5–33 % (Oshiro et al. 2016)¹. The sectoral discount rate represents 5 % for power and industry, 10 % for transportation, and 33 % for other sectors.”

- Do you start with the existing Japanese power system? What are your assumption on capital depreciation?

Yes, we begin with the existing Japanese power system. For large power generation plants, we included data on each individual power plant, such as year constructed, capacity, efficiency, *etc.* We have added corresponding text, as below.

“In the AIM/Enduse model, the residual capacities of the existing power plants in operation today were calculated based on individual power plant information, such as year constructed, capacity of each plant, and expected lifetime. The total capacity was calculated based on the capacity of newly installed power plants, which was determined endogenously, as well as that of existing plants.”

Please explain clearly why you do not account for climate damages on the economy, i.e., both production process and household utility (which hides behind linear expenditure system)!

First, we did not want to mix all of the information in a single study. Our primary focus was to identify the macroeconomic effects associated with climate change mitigation policies, and if we were to include these impacts, we would need to isolate these two effects. Second, the degree of global climate change and its impact is not determined exclusively by Japanese climate policy but is dependent on other countries’ future emissions. It would add more complexity to this study and make our primary focus unclear. Third, it is quite ordinary to exclude climate damage from climate change mitigation studies and, for example, most studies referred to in Chapter 6 of IPCC AR5 do not include climate damage. Thus, inclusion of climate damage is not a necessary condition for climate change mitigation studies. We have added the following sentences related to these discussions.

“In this study, we excluded the effect of climate change damage on the economy to avoid complexity (e.g. isolating mitigation effects from the mixture of climate change mitigation and damage impact, and additional assumptions on other countries’ emissions situations.”

When I work on forward-looking analyses, technological change is a key driver for results

- In both an energy-system and a macro-economic perspective, technological innovation and learning curves, i.e., decreasing costs are, of utmost importance. Think of solar PV or, currently, batteries
- Your technology cost assumptions are missing. Be transparent on all of them (appendix)
- Do you incorporate decreasing technology costs? If not, defend result!
- Sector coupling is key for (very) low-carbon energy systems (e.g. see the literature by M.Z.

Jacobson or C. Breyer). Please make it clearer in your model description how you deal, for instance, with hydrogen. Is it somehow included in your sectors? This would also be a credible long-term storage.

Thank you very much. We added cost information with respect to power generation, which also includes decreasing technology costs, to the Supporting Information.

Regarding hydrogen, we included it in the same manner as other energy carriers for energy demand sectors. On the production side, we also assumed that hydrogen production sectors will be introduced and provide input energy. However, within the Japanese context, our results indicate that hydrogen consumption will occupy a very small share of the final energy consumption (e.g. 0.02 %) in the mitigation scenarios up to 2050.

Variable	Unit	2010	2015	2020	2025	2030	2035	2040	2045	2050
Capital Cost Biomass w/ CCS	US\$2010/kW	7540	8063	8063	8063	8063	8063	8063	8063	8063
Capital Cost Biomass w/o CCS	US\$2010/kW	3813	4336	4336	4336	4336	4336	4336	4336	4336
Capital Cost Coal w/ CCS	US\$2010/kW	4338	4338	4338	4338	4338	4338	4338	4338	4338
Capital Cost Coal w/o CCS	US\$2010/kW	2704	2704	2704	2704	2704	2704	2704	2704	2704
Capital Cost Gas w/ CCS	US\$2010/kW	2174	2174	2174	2174	2174	2174	2174	2174	2174
Capital Cost Gas w/o CCS	US\$2010/kW	1122	1122	1122	1122	1122	1122	1122	1122	1122
Capital Cost Geothermal	US\$2010/kW	8716	8607	8607	8607	8607	8607	8607	8607	8607
Capital Cost Hydro	US\$2010/kW	8716	8716	8716	8716	8716	8716	8716	8716	8716
Capital Cost Nuclear	US\$2010/kW	4506	4721	4721	4721	4721	4721	4721	4721	4721
Capital Cost Solar PV	US\$2010/kW	5704	4031	3035	2588	2141	2065	1990	1914	1838
Capital Cost Wind Offshore	US\$2010/kW	5443	5704	5449	5194	4940	4797	4655	4513	4370
Capital Cost Wind Onshore	US\$2010/kW	3046	3145	2918	2725	2531	2531	2531	2490	2449
Lifetime Biomass w/ CCS	years	40	40	40	40	40	40	40	40	40
Lifetime Biomass w/o CCS	years	40	40	40	40	40	40	40	40	40
Lifetime Coal w/ CCS	years	40	40	40	40	40	40	40	40	40
Lifetime Coal w/o CCS	years	40	40	40	40	40	40	40	40	40
Lifetime Gas w/ CCS	years	40	40	40	40	40	40	40	40	40
Lifetime Gas w/o CCS	years	40	40	40	40	40	40	40	40	40
Lifetime Geothermal	years	40	40	40	40	40	40	40	40	40
Lifetime Hydro	years	80	80	80	80	80	80	80	80	80
Lifetime Nuclear	years	40	40	40	40	40	40	40	40	40
Lifetime Solar PV	years	15	15	15	15	15	15	15	15	15
Lifetime Wind Offshore	years	15	15	15	15	15	15	15	15	15
Lifetime Wind Onshore	years	15	15	15	15	15	15	15	15	15
OM Cost Fixed Biomass w/ CCS	US\$2010/kW/yr	982	982	982	982	982	982	982	982	982
OM Cost Fixed Biomass w/o CCS	US\$2010/kW/yr	290	290	290	290	290	290	290	290	290
OM Cost Fixed Coal w/ CCS	US\$2010/kW/yr	303	303	303	303	303	303	303	303	303
OM Cost Fixed Coal w/o CCS	US\$2010/kW/yr	106	106	106	106	106	106	106	106	106
OM Cost Fixed Gas w/ CCS	US\$2010/kW/yr	159	159	159	159	159	159	159	159	159
OM Cost Fixed Gas w/o CCS	US\$2010/kW/yr	34	34	34	34	34	34	34	34	34
OM Cost Fixed Geothermal	US\$2010/kW/yr	354	354	354	354	354	354	354	354	354
OM Cost Fixed Hydro	US\$2010/kW/yr	379	379	379	379	379	379	379	379	379
OM Cost Fixed Nuclear	US\$2010/kW/yr	200	200	200	200	200	200	200	200	200
OM Cost Fixed Solar PV	US\$2010/kW/yr	39	39	39	39	39	39	39	39	39
OM Cost Fixed Wind Offshore	US\$2010/kW/yr	241	241	241	241	241	241	241	241	241
OM Cost Fixed Wind Onshore	US\$2010/kW/yr	64	64	64	64	64	64	64	64	64

Minor points

Language must be improved

- Abstract: Try to shorten to at most 100-125 words

The Nature Communications guideline for authors says that abstracts should be approximately 150 words, and we think that our current abstract meets this standard.

- Abstract: What do you mean by 100% lower losses? Are they zero?

It should be 50% and we revised this accordingly.

- Line 24: Nationally

Thank you. We revised this.

- Line 33: 5th IPCC reports states 2-6%

Thank you. We revised this.

- Line 48: "Some argue..." Who?

We have added the corresponding reference.

Gherzi F. Hybrid Bottom-up/Top-down Energy and Economy Outlooks: A Review of IMACLIM-S Experiments. *Frontiers in Environmental Science* 2015, 3(74).

- Figure 4c: Parentheses do not match

Changed.

- Line 359: word "its" not appropriate

Thank you. We revised this.

- Line SI57: check table heading

Thank you. We revised this.

Further points

- Line 22: climate change is a particular challenge for developing countries

Revised

- Line 69: what about seasonal storage, which should become relevant under 80% decarbonization?

We agree that seasonal storage is an essential element. The actual results are, however, that there are buffers to deal with seasonal fluctuations, such as fossil fuel CCS thermal plants, in the 80 % reduction scenario, and thus, even if we consider battery storage for seasonal fluctuations, it would remain unused due to the cost competitiveness. The no CCS scenario also uses gas thermal plants to adjust for seasonal differences.

- Figure 1: the right column with arrows is hard to understand. Find a better format for representing exchange between models

We added numbers to the arrows so that we can specify what each arrow means.

- Line 103 and Figure S1 state that economic growth is a model input. How does this conform this GDP being the central output of interest?

This model input should be specified for the baseline scenario only.

- Line 109/Figure 2f: I don't see strong electrification in the figure

The term "strong electrification" is exaggerated. On the other hand, the electricity share in the final energy demand is higher in the mitigation scenario than in the baseline scenario because, with respect to the baseline scenario, the total final energy demand decreases in the mitigation scenario. Thus, we slightly modified the corresponding text.

- Figure 2c: How can final energy demand be larger primary energy? Imports? Heat pumps?

Indeed, secondary energy trade was not shown in that figure, but we have now added it.

- Figure 2a: What drives changes in the baseline?

As mentioned in the main text, the change in the total primary energy in the baseline scenario is due to decreasing population and modest economic growth. As for the share by fuel type, the phase-out of nuclear and lower energy price of coal drive changes to the baseline scenario. We added the following text.

"The main differences relative to the base year in the baseline 2050 model are the lower price of coal relative to other fossil fuels, and the share of nuclear energy, which reflects the current societal attitude toward nuclear power that limits new construction (Figure 2b)."

- Figure 2b: Why is there a marked kink in 2015 Oil w/o CCS?

This is due to the suspension of nuclear power generation after the accident at the Fukushima Daiichi plant. After 2020, power generation from oil will be replaced by restarting nuclear power and new construction of natural gas and coal plants, which are less costly than oil.

- Figure 2g: What are the low-carbon liquids, gases, and solids?

We have split these out.

- Figure 2h: Please explain the extreme high CO₂ price in 2050. Model-ending effects?

This is due to the technological availability in the AIM/Enduse model. We added an explanation of this.

“The price of carbon in the mitigation scenario increased over time and reaches approximately \$1000/tCO₂ in 2050. This high carbon price is due to the technological availability in the AIM/Enduse model”

- Figure 2: Please use one clearly distinguishable color for one data point!

Agreed.

- Figure 3: Curtailment numbers are quite high; please comment on connection to (conceivable) sector coupling!

We have added a comment.

- Lines 136- 137: Please state the “usual” numbers!

We have included these.

- Line 211-212: If you include an energy system model in a CGE model, your results indicate lower GDP losses. This does not mean that GDP losses will be in this range. As said, external validity is doubtful for me.

We concur with this point and appreciate it. We changed the text accordingly. As discussed above, external validity is impossible to prove in this paper and we need further studies based on results obtained by other modelling teams. As we have already responded we have added the following

statement

“For now, this study’s approach, and the derived implications are only applicable to Japan and within our modelling framework. We need further application fields and similar attempts from other teams, which can eventually show the external validity of our findings.”

- Table S1: Sample size, R-squared

We have added these.

- Table S3: Dispensable in this form; can be condensed to text

It might be possible to explain by using only text, but we prefer to keep this table to show precisely what we did. There are no strong disadvantages to keeping the table.

Reviewer #2 (Remarks to the Author):

GENERAL COMMENTS

The main concern for this paper is robustness of results for macroeconomic cost. The claim that economic models “are known to project higher costs than energy system models” is too broad a statement, given the wide variety of economic models and how they are parameterized. I would expect that counter-examples exist.

Thank you very much. We realize that the expression was overstated and the corresponding main text was written as “Traditionally, CGE models tend to project policy costs that are higher than those of energy system models”. We have revised this sentence as follows.

“However, such models are known to tend to project higher costs than energy system models.”

The paper could be re-framed as an exploration of methods to better characterize technology evolution than is currently possible in existing Integrated Assessment Models (IAMs), especially for deep de-carbonization scenarios.

A major strength of the paper is the use of a power dispatch model that deals with hourly patterns of supply and demand, and requirements for electricity storage.

The paper can be improved with less emphasis on macroeconomic costs, and more emphasis on the challenges of model integration for deep decarbonization.

We admit that we speculated about how we should summarize this study and there was an option to emphasise its methodological aspects rather than its implications. Meanwhile, although model integration is the core of the new method, the implications for macroeconomic costs make this study worth sharing with the research community, as well as other members of society, such as policymakers. The model itself is a tool to show something and we think that the scientific advances of this study are sustained by both the novel methodology and the new insights derived. Neither of these should be omitted. Therefore, we continue to emphasise the macroeconomic costs derived from the new methodology, but added a substantial description of the method and explained the mechanism by which we obtained different results to the conventional approach.

The key model linkage appears to be that from AIM/Enduse to AIM/CGE (discussion starting at line 347), where the autonomous energy efficiency parameters (AEEI) are endogenized (lines 357/358) to match final energy consumption. This essentially endogenizes productivity in the model, which

can cause large changes to the cost of emissions mitigation. More thought and discussion are needed on the implications of this type of model linkage. How comparable are scenarios if productivity parameters are endogenous?

We appreciate this comment. We concur that the essence of the method is to endogenize AEEI and change the energy productivity. At the same time, the conventional approach also endogenously determines energy productivity based on substitution elasticity, which is normally held constant. In that sense, the essential discussion point would be how the substitution elasticity and AEEI endogenized approaches differ. We have added a discussion on the comparability of the scenarios with endogenous productive parameters.

“Our approach implicitly assumes that the energy productivity in the CGE model is endogenized by using the energy system model information. This treatment is somewhat different from the conventional approach, in which CGE models use the same AEEI and constant elasticity substitution parameters, with and without mitigation policies. Based on the results showing that the macroeconomic costs associated with climate change mitigation policies are lower than estimated using conventional approaches, we can interpret the energy productivities in the mitigation scenarios as being higher than in the conventional approach. This would imply that the AIM/Enduse model incorporates higher productivity technological information than the conventional CES approach.”

Comparison to other types of model integration would be helpful. In particular, the full integration method demonstrated by Böhringer and Rutherford (reference number 19) is an ideal goal, although hard to implement in large models.

This is an excellent suggestion. Böhringer and Rutherford is obviously a remarkable model integration study; moreover, we realize that it would be good to have a more comprehensive comparison with other studies reviewed in this paper. Basically, we see that most model integration attempts only include the electricity sector. The studies that incorporate the full energy system rarely include power dispatch models, but instead take load duration curve approaches. These have the advantage of being relatively easy to reduce into a form for representing variability, whereas they have disadvantages in that they cannot guarantee hourly supply and demand matching. Thus, we added the table shown below to the Supplementary Information and discussed this.

Table Literature review of CGE and energy system integration studies.

Literature	Region	Integration method	Top-down model	Bottom-up model	Sector coverage of energy integration	Convergence confirmation	Electricity intermittency representation
Abrell and Rausch, 2016 ¹	Europe	Soft-linking with iterative procedure	Static CGE	Electricity	Electricity	Electricity price and quantity	LDC
Andersen et al. 2019 ²	Denmark	Soft-linking with iterative procedure	Static CGE	Energy system model (TIMES)	All energy	Energy price and quantity	LDC (32 time slices per annual)
Arndt et al. 2016 ³	South Africa	Soft-linking with iterative procedure	Static CGE (SAGE)	Energy system model (TIMES)	Electricity	Electricity price and quantity	No
Boeringer et al. 2008 ⁴	Global	Hard-linking	Static and dynamic CGE	Electricity	Electricity	-	No
Fortes et al. 2014 ⁵	Portugal	Soft-linking with iterative procedure	Static CGE (GEM-E3)	Energy system model (TIMES)	All energy	Energy service demand differences	No
Helgesen et al. 2018 ⁶	Norway	Soft-linking with iterative procedure	Static CGE (REMES)	Energy system model (TIMES)	All energy	Commodity price and sectoral output	LDC (260 time slices per annual)
Hwang and Lee, 2015 ⁷	Korea	Soft-linking with iterative procedure	Static CGE	Electricity	Electricity	Electricity price and quantity	LDC (12 time slices per annual)
Krook-Riekkola et al. 2017 ⁸	Sweden	Soft-linking with iterative procedure	Static CGE (EMEC)	Energy system model (TIMES)	All energy	Energy consumption	LDC (12 time slices per annual)
Lanzi et al. 2012 ⁹	Global	Parameter calibration	Recursive dynamic CGE	World Energy Outlook	All energy	None	No
Laurent et al. 2004 ¹⁰	Switzerland	Soft-linking with iterative procedure	Static CGE	Energy system model (MARKAL)	Houshold	Carbon price	No
Sue Wing et al. 2008 ¹¹	The US	Hard-linking	Static CGE	Electricity disaggregation	Electricity	-	No
Tapia-Ahumada et al. 2015 ¹²	Global	Soft-linking	Recursive dynamic CGE	Electricity	Electricity	-	Hourly
Tuladhar et al. 2009 ¹³	US	Soft-linking with iterative procedure	Dynamic CGE	Electricity	Electricity	Energy price and quantity	LDC (20 time slices per annual)
Vandyck et al. 2016 ¹⁴	Global	One-way Soft-linking	Recursive dynamic CGE	Energy system model (POLES)	All energy	None	No
Waisman et al. 2012 ¹⁵	Global	Recursive soft-linking	Recursive dynamic CGE	Energy system model (POLES)	All energy	None	No

The paper contains a lot of technical detail that will be difficult for a general audience to assess. Therefore, the paper may be better suited for a subject-specific journal than for Nature Communications.

We admit that there are technical details in this study, but there are several reasons to publish this

paper in Nature Communications. First, we would like to remind the reviewers of the aim of Nature Communications, which we refer to below, which clearly indicates that studies achieving important advances **within each field** are to be published:

Nature Communications is an open access, multidisciplinary journal dedicated to publishing high-quality research in all areas of the biological, health, physical, chemical and Earth sciences. Papers published by the journal aim to represent important advances of significance to specialists within each field.

We believe that we adequately present the scientific advances of the integrated assessment community. Second, the results of this study are meaningful outside of academia, as climate policymakers in Japan can refer to the results of this study when they consider Japan's long-term climate goals. Currently, the best available scientific knowledge regarding Japan's long-term climate mitigation goals are summarized in Sugiyama et al. ², but our study could largely affect those results. Third, we only applied this newly developed scientific approach to Japan, thus the results might not be relevant for many other countries. However, this methodology can be generalized and applied to other countries. We also believe that this study's method and derived implications have great merit for applications to other countries, and it will be valuable for all scientists within our research community to know about it. Moreover, it may be of interest to audiences beyond this community because climate change mitigation is receiving much attention from a range of scientific fields (e.g. materials science to invent new solar panels.)

In this regard, we revised the introduction and discussion to state these points more clearly.

“This study's approach, and the derived implications, are only applicable to Japan and within our modelling framework for now. Further applications and similar attempts by other teams will eventually confirm the external validity of our findings.”

SPECIFIC COMMENTS

Change in GDP is generally considered an inferior measure of policy cost in computable-general-equilibrium (CGE) models. Equivalent variation is the preferred measure and it should be possible to calculate both GDP loss and equivalent variation in AIM/CGE for comparison.

We fully concur that equivalent variation would be the preferred measure. The reason that we used GDP changes is that we wanted to compare the climate change mitigation costs derived from the energy system model, in this case AIM/Enduse. Within that context, change in GDP would be more relevant because the GDP captures total economic performance, whereas equivalent variation highlights changes in consumption. Meanwhile, we understand the reviewer's concern and therefore,

we have added results on equivalent variation as an additional index in Figure 4b, which shows an overall similar trend to GDP losses.

Figure 4. Climate change mitigation cost. **a** and **b** Time-series mitigation cost AIM/CGE results are represented as GDP loss rates and equivalent variation change rates relative to baseline scenarios. AIM/Enduse results are expressed as additional energy system costs of GDP relative to baseline scenarios. **c** and **d** show 5-year mitigation costs with varying technological availability; **c** illustrates the relationship of GDP losses in the CGE stand-alone and integrated models, and **d** shows GDP losses in the CGE stand-alone model and additional energy system costs in AIM/Enduse. The energy system model results shown here correspond to Enduse_results1 in **Error! Reference source not found.**

The sentence in the abstract at lines 12/13, “The GDP losses estimated with the integrated model

were significantly lower than those in the conventional economic model by more than 100% in 2050,” is not supported by analysis in the paper. The reduction appears closer to 50%.

Agreed. We revised this part.

It would help to include discussion on the need for two energy system models, AIM/Enduse and AIM/Power. Does AIM/Enduse simulate electricity demand by hour, by month, or by year?

AIM/Enduse’s temporal resolution is three hours, which is coarser than AIM/Power. This becomes extremely critical when determining battery capacity for short-term electricity fluctuations, curtailment and capacity factors under deep decarbonization scenarios because electricity variability is the key technological issue to resolve. We discussed this in the paper, as below.

“The electricity demand and supply system under stringent emissions reduction targets would be highly dependent on fluctuations in the electricity supply and demand patterns, which requires operation on an hourly basis. Therefore, we used AIM/Power in this model.”

Reviewer #3 (Remarks to the Author):

Review of «Energy Transformation Cost for the Japanese Mid-century Strategy: Energy System Feedback Effects in an Economic Model»

The paper investigates mitigation costs from reaching 80% greenhouse gas reductions in Japan in 2050. The study uses three models to analyze this question: a top-down CGE model (AIM/CGE), a bottom-up energy system model (AIM/endues) and a power dispatch model (AIM/power). Energy system results are taken as input into the other two models, and results are presented back as input to the energy system model.

This kind of study is demanding, since it requires insight in very different model types, covering broad sets of expertise. The analysis is based on repeated runs of the different models, exploring demanding challenges involving projections over several future decades.

The paper is interesting! Solid work has been done. I still think the manuscript may be improved on some aspects.

Thank you very much for your positive comments.

First, I have some general comments/questions:

- The paper focuses on mitigation costs, measured by GDP loss rates. The authors compare estimated costs from different model runs. It would be interesting to know if policy recommendations differ from an integrated model instead of stand-alone models (or alternatively whether policy assumptions or setup interpretation would differ in a stand-alone versus integrated setup). If policies do not differ between the model runs and different policy recommendations could not be inferred from model results, the question is which solution would be realized in real life. This does not depend on the model estimates, but on real-life action and real-life politics. The analysis focuses on reduced “model” mitigation costs from one model setup compared to another, and could be viewed as disconnected from real-life. If policy recommendations are similar, then real life outcomes following model result recommendations would not differ, even though different models estimate different costs. I am not convinced from the paper that the assertion in line 15 is justified, saying: “this type of integrated approach would be highly beneficial for setting national climate policies”. This might indeed be true, but the paper does not substantiate the assertion.

This is an excellent point. We are basically taking the position that this study does not explicitly recommend any policies, but rather provides new estimates of policy costs of climate change mitigation. Even though the Paris Agreement has been made, there is still political discussion

suggesting that climate mitigation costs negatively affect economic growth. We think that the major contribution of this study is to end such arguments by showing that the macroeconomic adverse side effects are not as large as previously thought. We can interpret the macroeconomic costs derived from this study as being small enough to support taking climate policy more seriously. In that context, we amended the abstract, as follows.

“this type of integrated approach would provide new insights by providing one of the key elements for setting national climate policies”

- Models are usually meant to bring new insights. CGE models are criticized for being black-box models. The paper handles the models rather as black-box models. Instead of explaining cause and effects by utilizing the models, the paper uses statistical analysis of model results in order to explain model relations. I think the authors should comment or discuss whether assumptions for using a regression model are justified (distribution of variables, independent residuals, ...). Performing statistical analysis on model results and discussing statistical significance of impact from economic sectors seems unconventional.

Example:

Line 172: “The transport sector’s effect (on GDP loss) is ambiguous, and its t-value is too small to reject the null hypothesis.”

Line 382: “We proportionally change the transport demand based on changes in GDP.”

If there is a clear model relation between transport demand and GDP, it might seem inappropriate to perform a statistical analysis on this relation.

Another reviewer recommended that we should not use regression analysis, so we decided to remove it from the paper. With respect to the black-box issue, we tried to make our method as transparent as possible. Moreover, full documentation is available in Fujimori et al. (2012)³, which we cited, and a wiki-based model description can also be seen on the website. As the wiki page was not cited in the previous manuscript, we have now added it.

https://www.iamcdocumentation.eu/index.php/IAMC_wiki

- There is unfortunately no common agreement on classifications of hybrid models. The term “integrated model” could be interpreted by some modelers as if underlying models run together (simultaneously) as one integrated mathematical model (and possibly solved simultaneously by one solver). The term integrated model is used differently in this application. It would be interesting to know whether results are exchanged automatically (this might be referred to as hard-linked models) or manually (this might be referred to as soft-linked models). The low number of iterations (two

iterations) could indicate soft-linking, but the high number of scenario setups (32 scenarios) could indicate hard-linking.

Agreed. Based on this comment, we considered the option of renaming “integrated model” as “coupled model”, but it seems that the same issue would remain. Thus, we decided to keep the terminology “integrated model” and clarify the definition, in which the soft-linking approach is mentioned explicitly.

“We call this new soft-linking modelling framework an “integrated model”.”

Regarding the number of iterations, we iterated five times and found the original iteration number, two, to be sufficient. We show how the main variables converged and remained stable after two iterations in the Supplementary Information by showing the baseline and mitigation scenarios in 2030 and 2050 for each. Moreover, no nuclear and no CCS scenario results are shown, either. The figure below is one such example.

Figure S7. Main energy, emissions and economic indicators of the baseline scenario in 2030 by iteration. Each panel illustrates an individual variable, the codes and units of which are listed in

Table S5.

Table S5. List of variables, codes and units.

Code	Variable	Unit
Fin_Ene	Final Energy	EJ/yr
Prc_Car	Price Carbon	US\$2005/t CO2
Prm_Ene	Primary Energy	EJ/yr
Prm_Ene_Coa	Primary Energy Coal	EJ/yr
Prm_Ene_Fos	Primary Energy Fossil Fuel	EJ/yr
Prm_Ene_Gas	Primary Energy Gas	EJ/yr
Prm_Ene_Nuc	Primary Energy Nuclear	EJ/yr
Prm_Ene_Oil	Primary Energy Oil	EJ/yr
Sec_Ene_Ele	Secondary Energy Electricity	EJ/yr
Sec_Ene_Ele_Bio	Secondary Energy Electricity Biomass	EJ/yr
Sec_Ene_Ele_Coa	Secondary Energy Electricity Coal	EJ/yr
Sec_Ene_Ele_Gas	Secondary Energy Electricity Gas	EJ/yr
Sec_Ene_Ele_Nuc	Secondary Energy Electricity Nuclear	EJ/yr
Sec_Ene_Ele_Oil	Secondary Energy Electricity Oil	EJ/yr
Sec_Ene_Liq	Secondary Energy Liquids	EJ/yr
Fin_Ene_Ele	Final Energy Electricity	EJ/yr
Fin_Ene_Gas	Final Energy Gases	EJ/yr
Fin_Ene_Ind	Final Energy Industry	EJ/yr
Fin_Ene_Liq	Final Energy Liquids	EJ/yr
Fin_Ene_Res_and_Com	Final Energy Residential and Commercial	EJ/yr
Fin_Ene_Solids	Final Energy Solids	EJ/yr
Fin_Ene_Tra	Final Energy Transportation	EJ/yr
Prm_Ene_Hyd	Primary Energy Hydro	EJ/yr
Prm_Ene_Solar	Primary Energy Solar	EJ/yr
Prm_Ene_Win	Primary Energy Wind	EJ/yr
Sec_Ene	Secondary Energy	EJ/yr
Fin_Ene_Res	Final Energy Residential	EJ/yr
Fin_Ene_Com	Final Energy Commercial	EJ/yr
Emi_CO2_Ene	Emissions CO2 Energy	Mt CO2/yr

Furthermore, I have some detailed, minor comments.

Line 12-13: A shortened sentence would read: “The GDP losses were lower by more than 100%”.

This wording is unfortunate, and the sentence should be reformulated. (The percentage comparison might come from lines 134-135, since 2.5% is more than 100% larger than 1.2 %.)

Agreed. We revised this part.

Line 24: NDC  I think it is “nationally” (not “national”).

Thank you. We have modified this.

Line 39: I think lines 39-44 should be further clarified/rewritten. According to the text, the first examples are global energy system models, and the last are CGE models. I don't think GCAM should be classified as an energy system model (line 39). MESSAGE could be classified as an energy system model, but MESSAGE-MACRO (line 40) represents a linking between a top-down macroeconomic model and a bottom-up energy system model. Line 42 reads: National models have applied “similar approaches”. At this point global “energy system” models have apparently been presented, while the model examples following this point do not apply “similar approaches” as such models.

We concur with this point. The core argument of this paper is coupling a multi-sectoral CGE model to an energy system model and therefore, we should classify the multi-sectoral CGE model or other models as economic and energy system models, respectively.

“Although there are other ways to classify the integrated assessment models, in this paper, we define “economic model” as the model that includes the multi-sectoral CGE model within the integrated assessment modelling framework, and “energy system model” as the model that does not.”

Line 48: “Some argue that ...” – please give references or describe sources.

We have added the relevant references.

Line 76: Consider a reformulation of the sentence.

Thank you. We have modified this as below.

“Similarly, the technological representation of power supply in the power dispatch model is more reliable than that in the energy system model.”

Line 108: The mitigation scenario exhibits ... strong electrification (Figure 2f). How/where is the strong electrification evident? The level of power generation in the baseline (figure 2b) and

mitigation scenario (figure 2f) seems rather similar, around 4 EJ/year in 2050 in both scenarios.

The term strong electrification would be exaggerated. On the other hand, the electricity share in the final energy demand is higher in the mitigation scenario than in the baseline scenario because the total final energy demand decreases more in the mitigation scenario than in the baseline scenario. Thus, we modified the corresponding text slightly.

Line 120: Is the heat legend relevant for figure 2c and 2g? The color is not visible in the figures. The same comment applies to figures S2 and S9.

Thank you. The heat usage in Japan is very limited and may not be visible in those figures. Therefore, we may not be able to deal with this issue.

Line 137-138: “The mitigation costs under such deep emissions reductions are usually not as low as this study’s estimates from the CGE stand-alone model.” This sentence says that the CGE stand-alone model estimates very low mitigation costs. Do I read the sentence wrong? I do not follow the logic in the paragraph.

The previous sentence should be revised. Thank you for pointing this out. We changed it as follows:

“The mitigation costs under such deep emissions reductions from CGE studies are usually not as low as our estimates”

Line 149: “We see systematic overestimates in the stand-alone model.” The authors are comparing results from models, but can hardly conclude that the stand-alone model overestimates mitigation costs. How do the authors know the true mitigation costs?

Great point. We will never know the true mitigation cost. All we can do is relative comparison. Thus, we changed the text accordingly.

“we again see systematically higher costs in the stand-alone model than in the integrated model”

Line 149: Is the word “this” correct? (Comparison of this integrated model’s ...)

No, it was wrong. It should be these

Line 154 - Figure 4: Scenarios “no nuclear” and “no carbon capture and storage (CCS)” are introduced.

It seems counter-intuitive that “no CCS” has lower policy cost than the default mitigation scenario in the CGE stand-alone (Figure 4b). Why should policy mitigation costs decrease by removing technological options?

Why does the “no nuclear” scenario have lower policy costs than the default mitigation scenario in the integrated model (Figure 2b and 2c)? This seems counter-intuitive (but not impossible).

This is a really excellent point. We took a close look at this behaviour of the model.

With respect to the no-CCS scenario, the CCS increases the intermediate inputs of each sector that utilizes CCS.

Regarding the no-nuclear scenarios, such tendencies, in which GDP loss rates in no-nuclear scenarios are lower than in default scenarios, can be seen in the latter years (2040s) and it seems that the investment in renewable energy in the period before 2040 is costly, but that investment contributes to lowering costs in the 2040s. These results are due to our modelling framework, in which all models use a recursive dynamic approach.

Line 168: I suggest that the estimates for industry and service sectors mitigating GDP loss rates by 0,40% and 0,50% should be commented also in view of the intercept estimated at 0.92% (and higher in Table S4).

We received a critical view of the regression analysis from reviewer #1, who recommended that we drop it. Therefore, we decided to exclude it from our analysis.

Line 211: Is decarbonizing the energy system harmful to macroeconomic growth? Ultimately, we found that “this will not occur” if energy system information is appropriately reflected in the economic model. First: The authors mix model results and real-world matters. Second: “This will not occur” is imprecise.

We fully concur that the sentence was inappropriate. We have modified as follows:

“we found that the macroeconomic costs are not as high as previously thought when energy system information is appropriately reflected in the economic model.”

Line 227: The authors state that “the elasticity parameter is normally assumed to be uniform, but it should differ among sectors, and probably regions”. I may misunderstand the substitution elasticity

of energy and value-added, but GTAP has sector specific elasticities and Koesler and Schymura (2012) estimates sector specific elasticities (Koesler, S. and M. Schymura (2012) "Substitution Elasticities in a CES Production Framework." Discussion Paper No 12-007).

Yes, our statement is not always the case. We modified the text accordingly.

Line 261: The meaning is unclear to me.

We agree with that this sentence is unclear. We modified it as follows.

“This notation is important when interpreting household results derived from integrated model results, where some may select economically irrational technologies and non-monetary factors are present”

Line 274: “Because the discrepancy improvements were sufficiently small, we stopped the calculation after the second iteration.” Was this the case for all 32 scenarios described in Table S3? Were the number of iterations predefined for the 32 scenarios, or was the convergence assessed in the same manner in all 32 scenarios – and all scenarios converged in two iterations? These questions matter with regards to the statistical analyses (ref Table 1 and Table S4).

We did not iterate for the 32 scenarios. These scenarios are systematic diagnostic scenarios by simply changing the on/off conditions. The reason that we did not run them is that the first and second iteration results did not change very much. We added a description to this effect. Furthermore, we decided to drop the statistical analysis, as suggested by the other reviewer, and instead added simple GDP loss information and discussed the sectoral contribution.

Line 285: Gross domestic product (GDP) appears to be an input to the CGE model, and a GDP development is reported as an assumption in Figure S1. Still GDP losses are reported as results from the CGE model. This seems contradictory, and should be clarified.

We fully agree with the reviewer’s comment. What we really input is TFP (total factor productivity), thus we amended that sentence.

Line 287: Should the sentence read “One characteristic of OUR industrial classification is ...”?

Indeed. Thank you.

Line 291: Incomplete reference? Reference 39 has three authors.

We have corrected the reference.

Line 321: ... considering the balances of electricity supply and demand in 3-h steps ... ?

It should be three hours.

Line 344: "... and increased capacity of demand ..." ?

We amended the corresponding text

"the power-dispatch stand-alone model does not determine the total electricity consumption and installed capacity by technology, which are given parameters"

Line 356: The sentence could be interpreted as if the energy demand is decided in AIM/Enduse, even though this model assumes energy service demand as exogenously provided. Is the point that AIM/endues provides energy consumption by energy types?

This sentence intends to describe the input to the CGE model. Thus, we modified the sentence as follows.

"Final energy consumption is classified into four sectors (industry, transport, service and residential) and fed into the CGE model."

Line 419: Where is the set SJ defined?

Line 420-422: Is this description correct? Is a_j a dummy parameter representing a set of years and sectors? Is X_j the estimated variable?

Thank you for pointing this out. This does not exist anymore because we removed the regression based on multiple reviewers' comments.

Supplementary information

Line 1: Typo in the heading.

Thank you.

Line 27: “End1_CGE1” is not apparent in Table S2, there are two columns with heading CGE1_End1. Probably line 27 is correct and the table headings are wrong?

The reviewer is correct. We modified the table heading accordingly.

Line 58: What does “XXX results for each model” mean?

We fixed the incorrect sentence in the caption of Table S2 and added a sentence to clarify the meaning of the heading in the table.

Line 69: You could explain the MER abbreviation (Market Exchange Rates) used in Figure S1b.

Thank you. We have defined this abbreviation.

Line 72 Figure S2h: Why does the “no nuclear“ scenario have baseline CO2 prices from 2025? Why are there no “no nuclear” mitigation CO2 prices? If the reported CO2 prices are mitigation prices, it seems counter-intuitive that they are lower in the “no nuclear” scenario than in the default scenario. Please clarify.

We apologize that the figure showed old data. Here, we show the latest results, which show the carbon prices start from 2025 in both scenarios. The reason why the CO₂ price is lower in without nuclear scenario than the default scenario is that the earlier investments for solar and wind occur in without nuclear option which eventually lowers the carbon price in the later period (2040-2050) while the former period (2025-2035) exhibits relatively high carbon price in the no nuclear scenario.

References

1. Oshiro K, Kainuma M, Masui T. Assessing decarbonization pathways and their implications for energy security policies in Japan. *Climate Policy* 2016, **16**(sup1): S63–S77.
2. Sugiyama M, Fujimori S, Wada K, Endo S, Fujii Y, Komiyama R, *et al.* Japan’s long-term climate mitigation policy: Multi-model assessment and sectoral challenges. *Energy* 2019, **167**: 1120–1131.

3. Fujimori S, Masui T, Matsuoka Y. AIM/CGE [basic] manual: Center for Social and Environmental Systems Research, National Institute Environmental Studies: 2012. Report No. : 2012-01.

Reviewers' comments:

Reviewer #1 (Remarks to the Author):

Dear authors,

Thank you very much for paying attention to my comments. Many questions have been answered. According to my judgement, the paper improved, but still needs revision.

In a nutshell, (i) the maths must be communicated more clearly and precisely, (ii) the central mechanism still needs some more explanation, precision, and intuition, (iii) now that the authors provided me with a better understanding of the model, they must devise some efforts to convince the readership that their finding is robust and not an artefact of parameter choices, (iv) the paper needs some editorial polishing concerning language and write up. I am willing to give a complete list of my editorial issues after the next round. I think that there are still some changes to the text to come first. I hope that this is okay for the process. (v) Relating to code and data availability, would it be possible to provide a "ready-to-use" version that contains exactly the model specifications you used for this paper? This would increase transparency and reproducibility.

Please find my detailed reply in the attached file. I replied to your rebuttal point by point in the file that was provided to me. You find my comments in red.

Dear authors,

Thank you very much for paying attention to my comments. Many questions have been answered. According to my judgement, the paper improved, but still needs revision. Please find my comments and responses in red. As I replied to your comments point by point, some of my points may be redundant. Please apologize for that. According to code and data availability, would it be possible to provide a “ready-to-use” version that contains exactly the model specifications you used for this paper?

Reviewer #1 (Remarks to the Author):

Summary and major claim of the paper

The paper aims to assess future economic costs of climate change mitigation. To this end, it devises a new integrated approach and applies it to Japanese 80% decarbonization pathways to 2050. The new integrated approach combines/soft-links an economic (top-down) CGE model with a (bottom-up) energy sector model and a technically detailed power system dispatch model. The central result states that economic costs of decarbonization – measured in percentage losses in GDP compared to a baseline without any decarbonization measures – are smaller when derived using the new integrated approach than when derived from a standard stand-alone economic CGE model. The reason for that would be the following: a standard CGE model employs a constant elasticity of substitution between the input factors energy and all others. This value is derived based on historical data and may not capture conceivable future developments in the energy sector. The new approach determines the elasticity of substitution endogenously and could better represent future changes in the energy system and their impact on the economic production process.

Thank you very much for your concise and comprehensive summary of this study.

You are welcome

Interest to others in the community and the wider field

The proposed approach is of high interest to the community and wider field. Ex-ante “evaluations” of the economic consequences of policy measures are of great relevance to the research community and policymakers when it comes to assess climate change mitigation measures. Also the identification of mechanisms that affect GDP loss estimates is timely and relevant.

Thank you very much for your positive assessment.

Overall evaluation

Overall, I like the idea of the paper. It touches a relevant point but suffers from various weaknesses. It leaves me somewhat puzzled.

- In a nutshell, too many points are opaque to assess both the novelty compared to the existing literature and the soundness of the proposed integrated model.
- There is also a range of conceptual issues, especially concerning model iteration and convergence as well as the “econometric” analysis

- Moreover, the external validity is highly doubtful as the application is driven by a number of (partly unstated) assumptions. A focus on the internal validity and the methodological contribution how and why the soft-linking decreases GDP loss estimates would help to make the paper more credible.

The authors must devise good efforts to make the paper clearer in both methods and communication and satisfactorily address all major points. Only then, I can soundly judge the paper and potentially recommend publication. Please find my detailed review below.

Thank you very much for your overall evaluation. We will respond to each comment in detail below, but here we present a brief response to the above comments.

First, regarding opaqueness, we added substantial text and mathematical equations to make the approach as transparent as possible in this revision. Second, we made further iterations, instead of just two, and showed how the variables converged more clearly than before. Third, econometric regression analysis may not be the best option for our purposes, based on the reviewers' comments. Hence, we dropped these analyses. Fourth, the external validity cannot be proven exclusively based on this paper, although we have tried to present our methodology as clearly as possible. This approach will hopefully be tested by other modelling teams. Finally, we made substantial modifications to our explanation of our results, focusing on their internal validity and the mechanism of how the soft-linking approach affects the GDP loss compared to the conventional approach.

Fine! In my view, the paper improved, but still needs revisions. In a nutshell, (i) the maths must be communicated more clearly, (ii) the mechanism still needs some more explanation, precision, and intuition, (iii) now that the authors provided me with a better understanding of the model, they must devise some good efforts to convince the readership that their finding is robust and not an artefact of parameter choices, (iv) the paper needs some editorial polishing concerning language and write up. I am willing to give a complete list of my editorial issues after the next round. I think that there are still some changes to the text to come first. I hope that this is okay for the process. Please find my detailed reply below.

Major points

Concerning structure

The authors could should do a better job in coming to the core of the paper (as I understand it, the endogenization of the substitution elasticity between energy and other inputs in the economic production process by using an energy system model).

Of course, the authors can structure the paper as they like. While I think that the paper would benefit from a more clearly visible red thread, I do not insist.

- The application to Japan is illustrative and should stay in the paper, but I see the contribution in the method. I highly doubt the external validity as numerical results are driven by numerous assumptions on functional forms and input data.

Thank you for pointing this out. We fully concur that the external validity depends on the functional forms and parameter assumptions. We have tried to make the assumptions and functional forms as transparent as possible in this revision. Meanwhile, we think that it will be difficult to prove the external validity within this paper because attempts by other modelling teams using the same (or similar) approach would be required. We added discussion related to this validity issue to the corresponding section.

“For now, this study’s approach, and the derived implications, are only applicable to Japan and within our modelling framework. We need further application fields and similar attempts from other teams, which can eventually show the external validity of our findings.”

Of course, you cannot substantiate or prove external validity here. But most importantly, the paper should not suggest so. Models can help in uncovering relationships, identifying sensitive parameters, illustrate relevant tradeoffs, and give an estimation of orders of magnitude, which is what the paper does. I am basically fine with that formulation. Yet I think that the approach as such is applicable to any country, but numerical findings may be rather specific to Japan. Please consider formulating accordingly.

- Likewise, the authors devote a lot of space to describing the differences between stand-alone CGE and integrated model. This can be shortened or longer parts relegated to an appendix. More space should be devoted to the central mechanism. The why is more relevant than the what

The revised version has improved. However, the central mechanism (the why) is still not sufficiently clear. The result as such (“lower GDP losses when using integrated model vs standalone model”) is interesting, but it is more relevant why this occurs and whether this is robust across a range of plausible parameter choices. You can basically generate a range of numbers by picking one or another plausible assumption on functional forms or parameter values. However, leaving as much as possible constant and then convincingly explaining why results are different is central here. Now that I (and other readers potentially also) better understand the forces that drive results, it remains to convince the audience that the difference is not an effect of mere parameter choices (e.g., numerical value of the substitution elasticity btwn energy and the capital-labor composite in the standalone CES), but robust and rooted in the methodological advancement.

- Therefore, authors should streamline the paper and feature the central mechanism more prominently. In the current manuscript, it becomes clear only in lines 222 - 232

This is an excellent comment and we really appreciate it. The central mechanism for changing the macroeconomic implications is adjusting the productivity of the primary factors (labour and capital) which consist of a major part of value-added.

Okay, thanks, but please get the econ terminology clear and aligned with standard wording. After talking to a fellow macroeconomist, I have some doubts whether the wording “value added” applies here. To the best of my knowledge, value added basically means value of output (quantity times price) minus value of inputs. If this is the standard terminology, please clearly explain in the paper to which concept “value added” refers here. Please also check the term “productivity” here.

This is because the total primary factor inputs are constrained exogenously each year. Please relate this assumption to the related (economic) literature.

The reason why the GDP losses are lower in the integrated model than in the stand-alone model is that the primary factor productivity is higher in the integrated model than in the stand-alone model.

Okay, this is important and it is good that you make it explicit. However, the central point to your analysis is why “primary factor productivity” (check term) is higher in the integrated model.

Hence, the productivity shifts are mainly driven by two factors. One is the productivity decreases associated with emissions reductions in energy end-use sectors such as manufacturing, transport and service sectors (e.g. capital replacement by expensive but energy efficient ones).

Word “hence” does not apply here. I understand that tighter emission caps may reduce productivity

(check term) in energy end-use sectors. But why are these decreases lower in the integrated model?

The other factor is sectoral allocation changes in primary factors.

Do you mean that sectoral allocation of capital and labor due to tighter emission caps occurs in a different manner for the integrated and the standalone model? Do sectoral output levels also change, i.e., does demand decrease?

For the first factor, Figure 5(a) illustrates the capital input efficiency of major industrial sectors (top 10 industries, which account for 95% of GDP in the base year) in the mitigation scenario compared to the baseline scenario for stand-alone and integrated models in 2050. Here, we define the capital input efficiency as capital input per output for each sector.

Terminology “relative ratio of capital efficiency (-)” in 5a unclear. Efficiency is a ratio. So you mean a ratio of ratios here? Please clarify! What about “labor input efficiency”?

Higher values indicate that additional capital inputs are needed in the mitigation scenario compared to the baseline scenario.

Understood. Please communicate that this is not an assumption on the production process, but an outcome of your model, as I understand it.

In general, the stand-alone model requires larger capital inputs than the integrated model in the mitigation scenario.

Understood! But why?

We can roughly compute to what extent the GDP losses are associated with these capital productivity losses by multiplying the value-added with the capital efficiency changes (Figure 5 (b, c)).

1) It may be redundant, but for me, as an economist, the concept of “value added share in GDP” is not entirely clear here. I also talked about this term to a fellow macro-economist who could not unanimously interpret it. Please explain carefully and clearly!

2) The sector “Other services” seems to be driving your results on GDP in the stand-alone model and thus of your paper. Please clarify what this sector is; also please clarify how reliable assumptions of the production process and the “value added” of such an aggregate sector are. Please also clarify why it is of such large importance concerning “value-added” changes in the stand-alone model and of no importance in the integrated model.

3) Figure 5c is nice. However, I do not get why the blocks are located differently concerning the zero line. As far as I understand it the total change to GDP from emission caps is negative. Figure 5c seems to devise a sectoral composition of these GDP changes, right? Please clarify. And why does it

appear positive in the upper bar and negative in the lower bar? Please also explain what the dots in the bars represent!

They eventually account for 1.3 % of the total value-added (GDP). Hence, the productivity differences between the stand-alone and integrated model are mainly caused by the differences in the function form and parameters, particularly for the value-added and energy bundle.

If this is the central explanation for **why** the integrated model shows lower GDP losses from tight emission caps, then this is somewhat unsatisfactory. Functional forms and parameter choices are quite debatable, e.g., the 0.4 for the elasticity of substitution below. Maybe, if you increase this value to 0.5, which may also be supported by the literature, then results change, eventually, also qualitatively? Please (i) clearly explain whether my assertion is correct or wrong; and (ii) if it is wrong, defend the choice of (your functional forms and) parameter values! I think I am not picky here - to me, this is central to understand how much you exactly add to the literature and how relevant your results are!

Here, we use a CES function wherein the substitution elasticity is 0.4 for the stand-alone model.

Substitution elasticity between capital and labor or x and energy? Please defend the choice of the value 0.4 or devise sensitivity calculations.

The future autonomous energy efficiency is adopted.

Please clarify what you mean by that! See also my comment below.

The integrated model uses a function of the same form, but the additional investment and energy inputs are provided by AIM/Enduse, by endogenizing the CES shift parameters (sector-wise additional investments are shown in Supplementary Table 9).

This seems to be at the heart of your methodological contribution; a new/improved way of linking CGE and energy system modelling. Please provide an explanation/intuition for the CES shift parameter here. In my view, this would communicate the central mechanism at play here more clearly. Moreover, I cannot link your comment to Table S9, which contains specific investment cost, lifetime, and OM costs for electricity generation plants. Also, I have some problems in fitting the “additional investments” into the story. Could you please clarify this? See also my comment below (where the maths is).

The second factor, effects of sectoral primary factor allocation change, is mainly driven by the power generation sector, where the electricity generation in the mitigation scenario compared to the baseline scenario is 20% higher in the stand-alone model, whereas it is almost same in the integrated model (Figure 5).

Figure 5d), right? Then I do not understand the label on the vertical axis. Please also carefully check all other text in Figure 5d)!

For me to get it right: electricity generation in the mitigation scenario is higher in the stand-alone model. This seems to be the second main driver of the central result. More electricity required due to mitigation → higher GDP effect because the exogenously limited production factors need to be allocated to electricity generation, right? In any case, it remains the question why this is the case. This is not sufficiently clear: why does sectoral primary factor allocation change? Why is electricity generation higher? It becomes somewhat better understandable in the maths in the appendix. But please write it clearly into the main text as this drives your central result!

There are certainly differences in power technological shares between the stand-alone and integrated models, but, in summary, it seems that the difference in total electricity generation between the models is the dominant factor.

Okay! How can you be sure that these “certain” differences are not a dominant part of the story?

The stand-alone model requires incremental capital and labour inputs that account for 0.4% of GDP compared to the integrated model (Figure 5 (d)).

Please explain more clearly. Do you mean that more factors (capital, labor) must be allocated to electricity generation in the standalone model, and because factors are exogenously constrained (i.e., no endogenous investment/depreciation, no endogenous population growth/labor supply decisions) GDP losses are higher? If this is true, why is this the case? According to my judgment, you must provide an intuitive, clear, and understandable explanation for that. Analogous to my comment above.

The total electricity demand is determined by the energy consumption represented in the energy end-use sectors. The total energy consumption in each energy end-use sector is represented by a CES function, as mentioned above, and the fuel-wise share is determined by a logit function in both the stand-alone and integrated models. The share parameters in the logit function are endogenously determined by the integrated model obtained from the AIM/Enduse results, whereas they are exogenous parameters in the stand-alone model.

After reading the maths below, I get an understanding for the difference. It seems to me that the central difference between integrated and standalone is that these share parameters are exogenous vs endogenous. Could you perhaps provide a table containing this and all other decisive differences (appendix)? In any case, the choice of the exogenous parameters for the shares of technologies in electricity generation are highly sensible for the results. You must defend this choice and ideally provide sensitivity calculations.

In addition to the two main mechanisms mentioned above, productivity changes and sectoral shifts in other sectors certainly occur, but are relatively minor.

You say “certainly”. From this I infer that you did not explicitly check that they, and their effect, are relatively minor. Please clarify or do so. Again, I do not want to be picky, but the central mechanisms deserves this great clarity and must be bullet proof!

In summary, we added one section to the results to discuss the points raised above and described the corresponding equations and parameter assumptions in more detail in the Supplementary Notes.

Thank you, fine. However, requires revision.

“Mechanism causing the differences in macroeconomic implications between the stand-alone and integrated models

The central mechanisms for changing the macroeconomic implications are changes to the productivity of primary factors (labour and capital) which constitute value-added. This is because the primary factor inputs are constrained exogenously for each year in our economic model. The reason that GDP losses are lower in the integrated model than in the stand-alone model is that the primary factor productivity is higher in the integrated model than in the stand-alone model. Hence, the differences in productivity are mainly driven by two factors. One is the productivity decreases associated with emissions reductions in energy end-use sectors, such as industry, transport and service sectors (e.g. capital replacement by expensive but energy efficient ones). The other factor is sectoral allocation changes in primary factors.

In the case of the first factor, Figure 5(a) illustrates the capital input efficiency of major industrial sectors (top 10 industries, which account for 95% of GDP in base year) in the mitigation scenario compared to the baseline scenario for stand-alone and integrated models in 2050. Here, we define the capital input efficiency as capital input per output for each sector. Higher values indicate that additional capital inputs are needed in the mitigation scenario compared to the baseline scenario. In general, the stand-alone model requires larger capital inputs than the integrated model in the mitigation scenario. We can roughly compute the value-added losses associated with these capital productivity losses by multiplying the value-added of each sector (Figure 5 (b, c)). These eventually account for 1.3 % of the total value-added (GDP). Then, the productivity differences between the stand-alone and integrated model are mainly caused by differences in the functional form and parameters particular to the value-added and energy bundle. Here, we use a CES function in which the substitution elasticity, share parameters and future autonomous energy efficiency are defined in the stand-alone model. The integrated model uses a function of the same form, but the additional investment and energy inputs are exogenously given by AIM/Enduse, whereas the CES shift parameters are determined endogenously (sector-wise additional investments are shown in Supplementary Table 9).

The second factor, namely the effect of sectoral primary factor allocation change, is mainly driven by the power generation sector. The electricity generation in the mitigation scenario compared to the baseline scenario is about 20% higher in the stand-alone model, but almost the same in the integrated model (Figure 5). There are certainly differences in technological shares between the stand-alone and integrated models, but, in summary, it seems that the difference in total electricity generation between the models is the dominant factor, where the stand-alone model requires additional capital and labour inputs, accounting for 0.4% of GDP, relative to the integrated model (Figure 5(d)). With respect to the representation of electricity demand, the total electricity demand is determined by energy consumption in the energy end-use sectors, which are represented by a CES function, as mentioned above. The fuel-wise share is determined using a logit function in both the stand-alone and integrated models. A parameter representing the preferences or technological choices in the logit function is determined endogenously in the integrated model, based on the AIM/Enduse results, whereas they are exogenous parameters in the stand-alone model. We describe the detailed mathematical formation and assumptions in the Supplementary information.

In addition to the two main mechanisms mentioned above, the productivity changes and sectoral shifts in other sectors certainly occur, but are relatively minor.”

Thank you; please revise according to my comments.

Figure 3. Valued-added differences between baseline and mitigation scenarios. Panel **a** illustrates sectoral capital input efficiency for the top 10 industrial activities in 2050. Panel **b** shows sectoral value-added share in the baseline scenario for the top 10 industrial activities. Panel **c** represents the capital productivity value-added effects compared to the total value-added for the top 10 industrial activities. Panel **d** illustrates the value-added share of power sectors in terms of total economy-wide value-added.

The mathematical description and parameter assumptions below are now included in the Supplementary Information.

3. Mathematical formula of CGE model

In this section, we describe the mathematical formula in the AIM/CGE model, which is particularly relevant to energy consumption and power generation.

3.1. Energy end-use sectors other than household sector in the CGE model

We begin with the representation of the energy end-use sectors (Table S5).

S6, right? Are all (non-household) energy end-use sectors production sectors according to the right column of new table S6 and vice versa? What about agricultural sectors?

The value-added and energy composite inputs are determined by multiplying a coefficient by the output from the energy end-use sectors (Equation 1). Then, value-added and energy composite are combined with the CES function (Equation 2). Labour and capital inputs are further nested in the CES function, as shown in Equation 3.

Generally, please motivate the nest structure (cf. van de Weerf, Production functions for climate policy modelling: an empirical analysis, Energy Economics 30, 2008); please also motivate the choice of parameter $ivae$ – assumption or calibration? Please also indicate how production activities a are combined to determine GDP which is the final outcome of interest.

$$QVAE_a^B = ivae_a \cdot QA_a^B \quad (1)$$

$$QVAE_a^B = \alpha ae_a (\beta ae_a \cdot QVA_a^B^{-\rho ae_a} + (1 - \beta ae_a) \cdot (at_a \cdot QENE_a^B)^{-\rho ae_a})^{\frac{1}{-\rho ae_a}} \quad (2)$$

$$QVA_a^B = tfp \cdot \alpha f_a (\beta f_a \cdot QF_{capital,a}^B^{-\rho f_a} + (1 - \beta f_a) \cdot QF_{labor,a}^B)^{\frac{1}{-\rho f_a}} \quad (3)$$

Check exponents in (3)

where

aCE is a set of production activities,

Please, clearly list all of them; the relation to the table S6 remains unclear. Also check your use of symbols in MS Word. On my computer, I have some strange notation, e.g., “aCE” here

QA_a^B is an output of sector a in the baseline scenario,

Please clarify the units of the variables (for me). Is it monetary or “real”?

$QVAE_a^B$ is the composite of the value-added and energy of sector a in the baseline scenario,

QVA_a^B is the value-added of sector a in the baseline scenario,

$QENE_a^B$ is the energy input of sector a in the baseline scenario,

It is unclear to me whether this energy input can be any energy type (oil, gas electricity)

$QF_{f,a}^B$ is the primary factor input of factor f and sector a in baseline scenarios (f = capital or labour),

$ivae_a$ is an input coefficient of the output of sector a ,

Seems a somewhat arbitrary choice to me; assumption or calibrated?

αae_a is the scale parameter of the CES function for value-added and energy aggregates,

βae_a is the share parameter of the CES function for value-added and energy aggregates,

ρae_a is the exponent parameter of the CES function for value-added and energy aggregates,

αf_a is the scale parameter of the CES function for primary factor aggregates,

βf_a is the share parameter of the CES function for primary factor aggregates,

ρf_a is the exponent parameter of the CES function for primary factor aggregates,

at_a is an autonomous energy efficiency improvement parameter, and

tfp is a parameter that represents the economy-wide total factor productivity, which is calibrated in the baseline scenarios by hitting the target GDP. The calibrated values are adopted in the mitigation scenarios.

Is the calibration base year the year 2010?

Energy consumption by fuel type in energy end-use sectors is determined by logit sharing, as follows:

$$SHENE_{c,a} = \frac{ae_c \cdot \delta_{c,a}^{en} \cdot PQ_{c,a}^{-\beta^{el}}}{\sum_{c \in CENE} ae_{cp} \cdot \delta_{cp,a}^{en} \cdot PQ_{cp,a}^{-\beta^{el}}} \quad c \in CENE, a \in AEnd \quad (4)$$

Please motivate and defend this logit sharing approach!

Where

$aCEEnd$ is a subset of the production activity and energy end-use production activity (e.g. industry, transport and so on),

Please clarify and be precise, you mean it is a subset of a union of two sets? What differentiates production activities from energy end-use production activities?

$cCEENE$ is set of energy commodities (coal, petroleum products and so on), which is a subset of

all commodities,

For clarification, does it also include electricity from the power generation “logit module” below?

$SHENE_{c,a}$ is the energy consumption share of energy commodity c and production activity a ,

For clarification: do you mean “in” or “and” production activity a ?

$PQ_{c,a}$ is the price of commodity c in production activity a ,

$\delta_{c,a}^{en}$ and β^{el} are parameters for logit selection, and

ae_c is the fuel-wise energy preference change parameter of commodity c .

This parameter seems rather arbitrary to me; what does “preference” mean here? Please motivate or, ideally, refer to some literature you base this approach on.

For the stand-alone model, βae_a^B and $\delta_{c,a}$ are calibrated by base year information. Autonomous Energy Efficiency Improvement (AEEI) in the stand-alone model at_a is one of the critical parameters determining energy consumption. We adopted a uniform AEEI across energy end-use sectors for each year, which is associated with GDP growth. For the years that assume more than 1% of GDP annual growth, the AEEI is 1%, and half of the annual GDP growth rates are assumed for the other the years. The fuel-wise energy preference change parameter ae_c is set annually to 1%, 0.5% and -0.5% for electricity, gas and coal, respectively, which represent the fuel shift from conventional solid and liquids to gas and electricity carriers.

Please be precise! AEEI is not 1%; rather its growth rate, right? In this case, the starting value is missing. Please also defend the choice of the parameter values for AEEI. This appears to be central for the difference in your numerical results. Please also better explain the preference parameter. Why can you exogenously pick values? Why is it in percent? Is this related to decarbonization requirements? If so, how?

Another point: how does SHENE relate to QENE in the standalone model?

With respect to the integrated model, the $SHENE_{c,a}$ and $QENE_a^B$ are fixed based on the AIM/Enduse model results and endogenise $\delta_{c,a}^{en}$ and at_a .

Please either describe the AIM/Enduse in equations (roughly) or refer to a citable source and explain in more detail how the model determines SHENE and QENE!

Then, we can derive the input coefficients of capital in the baseline scenarios after obtaining the simulation results of the baseline scenarios, as shown in Equation (5).

Are these input coefficients (capital input over output) equal to the “relative ratio of capital efficiency (-)” in Figure 5a? It seems. Please be also clear about your terminology! If so, how do you use the ifa to arrive at GDP changes?

For the mitigation scenarios in the integrated model, the derived input coefficients of the primary factors shown above are used to estimate primary factor inputs. For the capital inputs, additional investment costs associated with mitigation policy, which is given by the AIM/Enduse model, are added to the baseline inputs, as shown in Equation (6).

$$ifa_{f,a}^B = QF_{f,a}^B / QA_a^B \quad f \in Fcap \quad (5)$$

$$QF_{f,a}^M = ifa_{f,a}^B * QA_a^M + AddInv_{f,a} \quad f \in Fcap \quad (6)$$

where

$fCEcap$ is a set of capital and a subset of primary factors

I cannot find $fCEcap$ in eqs 5 and 6 and I do not want to guess. Please be precise and exact! Does set of capital and a subset of primary factors mean $fCEcap$ is a set union? Which primary factors do you mean?

ifa_a^B is an input coefficient of primary factor f and sector a in baseline scenarios.

$AddInv_{f,a}$ is the additional investment cost associated with mitigation costs provided by AIM/Enduse,

Please clarify the nature of these additional investments (to me)! Investments into what?

$QF_{f,a}^M$ is the primary factor input of factor f and sector a in the mitigation scenarios (f = capital or labour), and

QA_a^M is an output of sector a in mitigation scenarios.

3.2. Power generation in the CGE model

The total consumption of electricity is determined by the demand side representation shown in the previous subsection (and the latter for household).

Please apologize, but where do I find the total consumption of electricity above?

Then, the power generation is determined based on logit sharing, as below.

$$SHAC_{c,a} = \frac{\delta_{c,ap}^{el} \cdot PXAC_{c,a}^{-\beta^{el}}}{\sum_{ap \in AEly} \delta_{c,ap}^{el} \cdot PXAC_{c,ap}^{-\beta^{el}}} \quad c \in CEly, a \in AEly \quad (7)$$

where $aCEly$ is a subset of production activity and electricity production activity (e.g. coal, solar PV and so on),

Please motivate/defend the use of logit sharing in representing electricity generation.

$cCEly$ is a subset of commodity (electricity),

Subset containing what?

$SHAC_{c,a}$ is the electricity generation share of production activity a ,

I am sorry, but what exactly do you mean? This formulation suggests that each production activity a (wood products, iron and steel, ... according to Table S6 if you mean the same “a”s as above)

generates electricity. I guess you mean the share of an electricity generation technology in total electricity generation.

$PXAC_{c,a}$ is the price of commodity c produced by production activity a ,

$\delta_{c,a}^{el}$ and β^{el} are parameters for the logit selection of power general technologies.

For the stand-alone model, $\delta_{c,a}^{el}$ is calibrated using base year information.

Please explain (to me) how you decarbonize electricity generation in the standalone model; it is solely using the CO2 cap, right?

The integrated model fixes the $SHAC_{c,a}$ as the AIM/Enduse model results and endogenises $\delta_{c,a}^{el}$. This treatment is the same between the baseline and mitigation scenarios. As the absolute amount of power generation is determined by the demand side, here we specify only the share. The transmission losses are also considered. Battery capacity is input as an absolute amount.

3.3. Household consumption in CGE model

Household consumption is formulated using a LES function and further nested in a CES function for energy and other manufacturing goods. Then, the total energy consumption is finally split out into fuel-wise consumption using a logit function

$$QCH_{ch} = \mu_{ch} + \theta_{ch} \cdot \left(\frac{EH}{PCH_{ch}} - \sum_{chp \in CH} \mu_{chp} \right) \quad ch \in CH \quad (8)$$

$$QCH_{ch} = \alpha h_{ch} \left(\beta h_{ch} \cdot QHE_{ch}^{-\rho h_{ch}} + (1 - \beta h_{ch}) \cdot QHM_{ch}^{-\rho h_{cp}} \right)^{\frac{1}{-\rho h_{ch}}} \quad (9)$$

$$SHHENE_{c,ch} = \frac{ae_c \cdot \delta_{c,ch}^h \cdot PQ_{c,h}^{-\beta h}}{\sum_{cp \in CENE} ae_{cp} \cdot \delta_{cp,ch}^h \cdot PQ_{cp,h}^{-\beta h}} \quad c \in CENE, ch \in CHE \quad (10)$$

$$QH_c = \frac{\sum_{(c,ch) \in mapCCH} QCH_{ch} \quad c \in CNENE}{\sum_{(c,ch) \in mapCCH} QHM_{ch} \quad c \in COMF} \quad (11)$$

where

$chCEH$ is a set of household consumption goods, of which mappings with goods c are shown in Table S10,

$chCEHE$ is a set of energy-related household consumption goods (car usage and other energy-related consumption)

$(c, ch) \in mapCCH$ is a mapping from household consumption goods ch to general goods c , shown in **Error! Reference source not found.**,

PCH_{ch} and QCH_{ch} are the quantity and price of household consumption goods ch ,

$SHHENE_{c,ch}$ is the share of energy fuel c of household consumption goods ch ,

QHE_{ch} and QHM_{ch} are the quantity of energy and other manufacturing goods, respectively, for

household consumption goods ch (car usage and other energy-related consumption),
 θ_{ch} and μ_{ch} are LES function parameters,
 α_{ch} is the scale parameter of the CES function for energy and other manufacturing goods aggregates,
 β_{ch} is the share parameter of the CES function for energy and other manufacturing goods aggregates,
 ρ_{ch} is the exponent parameter of the CES function for energy and other manufacturing goods aggregates,
 $\delta_{c,ch}^h$ and β^h are parameters for logit selection of household energy fuel.

Household LES function parameters in the stand-alone model are updated recursively based on income elasticity. Electricity and biofuel used in transport are not accounted for in the base year social accounting matrix and, thus, we introduce the initial parameters for $\delta_{c,ch}^h$ and $\delta_{c,a}^{en}$ by calibrating 0.1 % of the share in each energy consumption in 2015 and 2020. They are then updated afterwards, to one-third of the value of petroleum products in 40 years. For the integrated model, the parameters β_{ch} and ρ_{ch} were determined endogenously based on the energy consumption and investment needs for other manufacturing goods computed by AIM/Enduse.

Please (i) motivate using this household representation (literature), (ii) explain whether there is any market clearing between household consumption and production of goods and how it is implemented, and (iii) related to that if there is import and export of goods and services.

Concerning analysis

The exposition of the models is opaque and must be considerably improved. The model is highly complex but under-explained. It is difficult to deeply understand what drives the results. Please apologize that I go into details here, but all details together constitute a major issue.

- Overall language must be more precise. Especially, central concepts must have a clear and immutable name. E.g.

We highly appreciate you pointing this out. We have amended the text accordingly and read through the entire manuscript once again.

Thank you, fine

o “value-added”, “energy service demand”, ...

We use the terms “value-added” and “energy service demand” immutably rather than using “value added” and “service demand”

Okay

- CGE and energy system model:

o Clearly state which sectors you analyze. I was puzzled throughout the paper

- CGE model

o The description must be clearer. I have some experiences with CGE models, but lines 292 to 310 are opaque

→ Which production sectors? Is there a final consumption good or multiple?

→ Which energy transformation sectors? What do you mean here by “value-added” as an input? Do you mean capital?

→ Which energy end-use sectors? What do you mean here by “value-added” as an input?

We have added a list of production sectors to the Supporting Information, which we quote here. We also clarified the meaning of value-added.

Okay, you mean Table S6, right? Please clarify and pay attention to the typos.

→ Please explain the concepts of energy and material balances in power generation more clearly!

→ Lines 299-300: Which share do you mean? Which calibrated information do you mean?

Our intended meaning was the share of each technological power generation method out of the total electricity generation. We have clarified this point.

Okay. The point of the energy and material balances is still open

- Energy system model Enduse

o Clearly state the scope of the model!

o Line 314: Be more specific on energy end use and supply sectors! Are they the same as in the CGE? Are they a disaggregated representation of the sectors in the CGE?

Regarding the scope and granularity of sectors in AIM/Enduse, we added the following sentences.

“The model covers energy-related GHG emissions from both energy end-use and energy supply sectors. The end-use sectors are composed of industry, buildings and transportation sectors, and they are disaggregated into several subsectors with respect to types of products, buildings, and transportation mode based on the IEA energy balances.”

Okay, thanks. It is fair that you cannot describe the entire model here. You state that it is a “recursive, dynamic partial equilibrium model” (line 394). Could you please provide some more information here? Is it an optimization model (minimization of total system costs = partial equilibrium as outcome of perfectly competitive market) or an equilibrium model featuring multiple agents with distinct objective functions?

As the difference between the sectoral granularity of AIM/Enduse and AIM/CGE should be elaborated in the data exchange section rather than in each model description, we added the corresponding sentences to the “Information from AIM/Enduse provided to AIM/CGE” section.

“Because the sectoral disaggregation of AIM/Enduse basically complies with the IEA energy balance, there are inconsistencies in the AIM/CGE, which is based on an input-output table. Thus, in terms of data exchange from AIM/CGE to AIM/Enduse, the subsectors are aggregated so that the granularity of the sectors is in agreement. Nevertheless, given the large share of industrial GHG emissions in Japan’s long-term low carbon scenarios, iron, chemical, paper, non-metallic minerals, and non-ferrous metals are exempted from the sector aggregation.”

Okay, thanks

o Line 318: Why are energy prices a constraint in the model? Which energy do you mean? Electricity? In my understanding of an energy system model, they are a result.

As Japan’s energy supply largely depends on imported fossil fuels, primary energy prices must be defined exogenously in this model. We edited the text as follows.

“Mitigation options are selected based on linear programming to minimize total energy system costs that include investments for mitigation options and energy costs subject to exogenous parameters such as cost and efficiency of technology, primary energy prices, energy service demands and emission constraints.”

Thank you, fine

o Line 325: Which energy-demanding sectors? Those in Table S3?

Yes, it is consistent with those in Table S3. However, we avoid using this term to clarify the model description. We have edited the text as shown below.

“In the industry, building and transportation sectors, wide mitigation options are included, such as energy-efficient devices and fuel switching.”

Okay, thanks

- Power dispatch model

o Clearly state whether investment decisions are part of that model! It seems not, but in line 332 you call it a generation expansion model

As for the second sentence of your comment, unfortunately, we could not find the words “generation

expansion model”. We used the term “generation planning model” on line 332. We think that the term “generation planning” can be used for planning operation schedules, frequency management, and storage management, which are modelled in AIM/Power. Thus, we did not modify the term used on line 332.

My mistake, please apologize!

As for the first sentence of your comment, we added the following description of the investment decisions made by the power dispatch model in this study after the first paragraph of the section on the AIM/Power model.

“Note that as AIM/Enduse provides power generation installed capacity for AIM/Power, AIM/Power does not make investment decisions, except for making additional investments in storage and power plants aimed at hourly and within hourly power demand-supply management”

Thank you, fine

- Lines 336-337: What do you mean by “or both”? Is it possible that the model satisfies either only electricity demand or only emission reduction targets? This appears strange to me.

Thank you very much for your comment. It was a misleading sentence. It has been modified as follows.

“several constraints, including satisfying electricity demand and CO₂ emission reduction targets”.

Thank you, fine

- Also explain the difference in the scopes of the power dispatch and energy system models more clearly! What do they cover?

Thank you very much for your comment. The following sentence has been added.

“In other words, unlike the AIM/Enduse model covering all energy-related sectors, the AIM/Power model only covers the power generation sector.”

Thank you, fine

The explanation of the communication between models is insatisfactory and must be improved.

- Exchange from Enduse to CGE

o You state that both AIM/Enduse and AIM/Power decide on battery investments. Please clarify!

We stated that AIM/Power determines the capacity factors of the batteries, and that capacity is determined by AIM/Enduse. Then, CGE incorporates both data. We have clarified this point.

Okay. Can you please clarify for me what you mean by capacity factors of batteries? To the best of my knowledge, the capacity factor is the average hourly use of a technology over a year (or other timeframe), defined between zero and one. I know it for wind power or PV, for which it is given as exogenous data. For batteries a higher capacity factor may trigger higher investment, but higher

investments may also trigger a lower/(higher) capacity factor. Please explain (to me). Thank you.

o Line 367: What do you mean by “baseline case”? What do you mean by the implicit assumption “that the baseline investment cost is inherent in the CES substitution”?

We meant to say that the energy investment of energy end-use sectors in the baseline scenario is modelled by a typical CES nested function. We have modified the text to clarify this point.

Ok

o Line 368: What do you exactly mean when you say that “capital input coefficients are fixed at baseline levels”? I think that this is the central point of the paper where you replace the exogenously assumed substitution elasticity by results from the energy system model. If so, please explain way more clearly and understandably. This is at the heart of the contribution of the paper!

Thank you. We think that it would be better to use a mathematical formulation to provide a precise explanation. We have added this to the supplementary information, as mentioned above.

Thanks, answered in your comments above

- Exchange from AIM/CGE to AIM/Enduse

o Line 382: What do you mean by “industry service demand”?

Thank you very much for pointing this out. We added a more detailed explanation, as indicated below.

“The energy service demand in the industrial sectors, such as steel and cement production, and outputs of other industrial sectors, is altered by the outputs from AIM/CGE.”

Thank you, fine

o Lines 384-386: Tentative guess of substitution elasticities being unity based on some Swedish econometric analysis is doubtful. Please more explanation here or argumentation why this is okay.

Thank you very much for raising this important point. We think that not asserting this tentative assumption is okay because the elasticity between monetary and physical units is uncertain. However, we believe that the impact of this uncertainty on the major findings of this paper is negligible, because the GDP losses estimated in this paper are low enough (3% at maximum) in the CGE stand-alone model. To test the uncertainty in the elasticity, we ran scenarios in which we varied the elasticity from 0.5 to 2.0. Our results indicate that the elasticity assumption affects the numbers, but the same qualitative conclusion holds. We have edited the sentences as below.

“According to the Swedish econometric analysis⁴⁸, elasticity between monetary and physical units of energy services can be assumed to be approximately 1.0. Furthermore, the GDP losses indicated in this study are relatively small, less than 3 %, in the CGE stand-alone model, as shown in Figure 4a. Thus, we tentatively applied an elasticity value of 1.0. Meanwhile, we varied the elasticity from 0.5 to 2.0 and observed that the policy costs change slightly, but the qualitative conclusion still holds (Supplementary Figure 5).”

Supplementary Figure 5. Climate mitigation policy costs associated with variations in the elasticity in monetary outputs and physical energy service demand.

Thank you. The sensitivity and the figure are convincing. Could you please clarify what exactly you mean by the “elasticity between monetary and physical units”? I do not fully get it.

- Exchange from AIM/Enduse to AIM/Power and vice versa

o Please clarify in which model you determine which battery capacities!

Although we calculated the battery capacities in both AIM/Enduse and AIM/Power, we used the results from AIM/Enduse in our analysis. This is because the decision to invest in batteries for

long-term fluctuations is part of capacity planning, as are investment decisions relating to pumped hydro power plants. Battery capacities for short-term fluctuations were determined by AIM/Power, which can model hourly electricity demand-supply balances.

Thank you. I think you can assume this and that this is uncritical. Can you please clarify how you differentiate between long-term and short-term batteries? Different investment costs for energy and power? Different technologies, i.e., li-ion vs redox flow or the like? Can you briefly comment on E/P ratios (endogenous/exogenous)?

The following sentence was added to the last section of “Information from AIM/Power provided to AIM/Enduse”.

“Note that generation capacity, although not directly related to balancing short-term fluctuations, is decided by AIM/Enduse. Thus, the battery capacity is determined by AIM/Enduse when considering long-term electricity fluctuations, whereas that for short-term fluctuations is provided by AIM/Power.”

Thanks, fine; please potentially adapt to my comment above.

o Please clarify the importance of the capacity factors! If you have a three-hour granularity in the Enduse model, why are they important (for investment decisions)?

Even AIM/Enduse has a three-hour granularity, and this time resolution is insufficient for assessing demand-supply balances for electricity (and this is the reason why we incorporated AIM/Power in the model integration framework). Curtailment is one effective method for increasing the time resolution of the model, because the curtailment ratio directly affects the capacity factor of each generation plant. Specifically, X% of curtailment ratio is equivalent to an X% reduction in the capacity factor, which is a crucial factor for investment decisions.

To clarify these points, we added the following sentence.

“Moreover, the power system would respond to large-scale renewable energy installations by adjusting the capacity factor for conventional power generation systems (e.g. coal-fired power) in addition to curtailing the output from variable renewables. These measures for balancing short-term fluctuations reduce the electricity output per installed capacity, and thus affect investment decisions. It is necessary to consider this feedback from AIM/Power to AIM/Enduse.”

Okay, thanks

- Iteration and convergence

o Line SI25: How do you apply an RMSE? If you compare two values (for the same variable in the two models), then the concept is not applicable. If you compare more than two values, please

clarify!

- o Table S2: In any case, the table is difficult to understand, please explain more clearly what the percentage numbers mean!
- o Table S2: I do not see convergence. Basically, results for almost all variables differ by a large degree between CGE and Enduse, and only the magnitude of the difference stays roughly constant after the second iteration. In my understanding, convergence must imply that variables take on identical values between models.
- o Related to that, two iterations seem unrealistically few.
- o Please also comment on the computational burden
- o You state that “the absolute value of energy consumption is not fully harmonized between” the CGE and energy system model (line 362). Do you refer to Table S4 here? If no, explain. If yes, this seems problematic for me.

Regarding the iterations, there are three points for discussion here. First, the definition of convergence depends on the study, and within this study, we define it as the iteration point where differences between models stop improving.

Sure, this is convergence. But convergence to a (potentially substantial) difference.

Technically speaking, we cannot force all AIM/CGE variables into AIM/Enduse. At the same time, we reconsidered the method for inputting AIM/Enduse data into AIM/CGE and it turned out that we could use a slightly different approach, in which the absolute final energy consumption information is input directly (previously, the fuel share was input). The differences between AIM/Enduse and AIM/CGE were much smaller than in the previous approach, and we updated the description of our method accordingly.

Fine

Non-energy use and LPG accounts are post-processed and agricultural energy is not fully harmonized. Consequently, small gaps remain, but these are small enough to stop iterating.

I am not too much worried about that.

Second, as the differences would not improve, it would not be meaningful to iterate further. However, we did carry out further iterations, up to five, and present all of our results. The results confirm that most variables converge after two iterations. We did this for the default case as well as the no CCS and no nuclear cases for the baseline and mitigation scenarios, respectively. In the Supplementary Information, we show all comprehensive indicators by iteration for each scenario. In this response letter, we selected two examples, namely the 2050 baseline and mitigation scenarios, for default technological assumptions (see Supplementary Information for more scenarios and years).

Okay, looks quite convincing; but I am still a bit sceptical. Could you provide some intuition (for me and the reader) why convergence is achieved so fast? I also work with models that need to converge, but this usually requires more iterations.

Third, we admit that the root mean square error was not the exact metric used, and we define the error indicator below. Here, we regard the AIM/Enduse and AIM/CGE information as the true and estimated values, respectively. Although there is no true value for the future, we think that the metric itself is valuable. According to the comment, we changed the name of that indicator.

The indicator shown below is adopted. i , t and s are sets of variables of model (e.g. energy demand), years and scenarios, respectively. $X_{i,t,s}$ and $Y_{i,t,s}$ are AIM/CGE and AIM/Enduse outputs, respectively.

$$\text{Error indicator}_{i,s} = \frac{\sqrt{\sum_t (X_{t,i,s} - Y_{t,i,s})^2}}{\sqrt{\left[\sum_t \left(\frac{X_{t,i,s} + Y_{t,i,s}}{2}\right)^2\right]}}$$

Figure S8. Main energy, emissions and economic indicators of the baseline scenario in 2050 by iteration. Each panel illustrates the individual variables, the codes and units of which are listed in Table S5.

Figure S9. Main energy, emissions and economic indicators in the mitigation scenario in 2050 by iteration. Each panel illustrates individual variables, the codes and units of which are listed in Table S5.

Table S1. List of variables, codes and units.

Code	Variable	Unit
Fin_Ene	Final Energy	EJ/yr
Prc_Car	Price Carbon	US\$2005/t CO2
Prm_Ene	Primary Energy	EJ/yr
Prm_Ene_Coa	Primary Energy Coal	EJ/yr
Prm_Ene_Fos	Primary Energy Fossil Fuel	EJ/yr
Prm_Ene_Gas	Primary Energy Gas	EJ/yr
Prm_Ene_Nuc	Primary Energy Nuclear	EJ/yr
Prm_Ene_Oil	Primary Energy Oil	EJ/yr
Sec_Ene_Ele	Secondary Energy Electricity	EJ/yr
Sec_Ene_Ele_Bio	Secondary Energy Electricity Biomass	EJ/yr
Sec_Ene_Ele_Coa	Secondary Energy Electricity Coal	EJ/yr
Sec_Ene_Ele_Gas	Secondary Energy Electricity Gas	EJ/yr
Sec_Ene_Ele_Nuc	Secondary Energy Electricity Nuclear	EJ/yr
Sec_Ene_Ele_Oil	Secondary Energy Electricity Oil	EJ/yr
Sec_Ene_Liq	Secondary Energy Liquids	EJ/yr
Fin_Ene_Ele	Final Energy Electricity	EJ/yr
Fin_Ene_Gas	Final Energy Gases	EJ/yr
Fin_Ene_Ind	Final Energy Industry	EJ/yr
Fin_Ene_Liq	Final Energy Liquids	EJ/yr
Fin_Ene_Res_and_Com	Final Energy Residential and Commercial	EJ/yr
Fin_Ene_Solids	Final Energy Solids	EJ/yr
Fin_Ene_Tra	Final Energy Transportation	EJ/yr
Prm_Ene_Hyd	Primary Energy Hydro	EJ/yr
Prm_Ene_Solar	Primary Energy Solar	EJ/yr
Prm_Ene_Win	Primary Energy Wind	EJ/yr
Sec_Ene	Secondary Energy	EJ/yr
Fin_Ene_Res	Final Energy Residential	EJ/yr
Fin_Ene_Com	Final Energy Commercial	EJ/yr
Emi_CO2_Ene	Emissions CO2 Energy	Mt CO2/yr

Concerning novelty

As the analysis has many unclear points, it is hard to judge genuine novelty. The literature features a

range of attempts to combine top-down CGE with bottom-up energy system models; among others:

- The Bohringer-Rutherford stream (e.g., Energy Economics 30 (2008), 574-596, Journal of Economic Dynamics and Control 33(9) (2009), 1648-1661 and further papers by the same author)
- Tuladhar et al. (Energy Economics 31(2), (2009), S223-S234)
- Abrell and Rausch (Journal of Environmental Economics and Management 79 (2016), 87-113)
- Tapia-Ahumada et al. (Economic Modelling 51 (2015), 242-262)
- Hwang and Lee (Energy Policy 83 (2015), 69-81)
- The IAM community (e.g., REMIND (<https://www.pik-potsdam.de/research/sustainable-solutions/models/remind/remindequations.pdf>))

Several of the above papers also explicitly analyze macro-economic effects (Tuladhar, Abrell and Rausch, Hwang and Lee). I did not carry out an exhaustive search, but there are, confidently, more.

We deeply appreciate you pointing this out. We agree that some references were missing from the previous manuscript and we have, therefore, made a table exhaustively listing studies linked to multi-sectoral CGE and energy system models in the Supporting Information. The above literature pointed out by the reviewer is either included in the list or discussed within the main text.

Thanks, fine.

Meanwhile, we would like to remind the reviewers that the novelty of this paper is that the policy cost estimates are smaller than previously thought. We do think that this has not been discussed before, as we clearly compared the conventional stand-alone CGE and integrated models. To clarify this point, we modified the text as below.

“we identify the magnitude of the differences in macroeconomic costs for climate change mitigation using values derived from this newly integrated model and the conventional economic model approach, and determine which sector’s representation is an influential factor. This is the novelty of this study.”

Thanks, fine.

As the exposition of the model in this manuscript is shallow and not sufficiently clear, it is hard to judge whether its contribution is sufficiently novel. While energy system-macro model integration is not entirely new, I would judge the integration of two highly detailed models sufficient.

Likewise, the observation that CGE and energy system models differ in results concerning their GDP impact is known, as the authors state. The new insight of this paper would be why they differ. The explanation given is plausible (“kind of” endogenous substitution elasticity), but does not become sufficiently clear from the analysis. This is a main point the authors must address.

Thank you very much. We fully concur. As mentioned above, the key message of this paper is that the macroeconomic implications differ between the integrated and standalone models rather than methodological advances, although the three models (CGE, energy system and power dispatch model integration) could be new.

As suggested above, we significantly modified and supplemented the text, as described above, to explain why they differ.

Ok

Further major points

Diagnostic scenarios

As such, a good idea, but the regression analysis is awkward (Tables 1 and S4). Drop it.

- You have a small sample (N=32 in Table S4). Estimators cannot be assumed to be distributed normally. Did you account for that?
- Regression analysis is an appropriate method to estimate population effects from samples. Your data is deterministic. You know the statistical population. There is no need for regression analysis.
- You can just compare means. Take the mean of the GDP result of all scenarios with a certain feature (16 when for a single year) and the mean of the GDP result without this feature and compare.
- It would also be interesting to see whether the detected effect (e.g. representation of industry is relevant) is more pronounced in combination with other features. You could populate a Table like S3 with your GDP results. This would be more insightful
- All tables must state the sample size and the R-squared
- Line 175: as far as I see you do not have a fixed effect (=time-constant unobserved heterogeneity in the cross-sections)
- In any regression, you estimate parameters with given data for variables; adjust terminology

Based on the suggestion that there is no need for regression analysis, we decided to remove it. It was also recommended that we show a table like S3 with GDP changes, and the corresponding table has been added. We discussed the table and, although it may not be easy to say deterministically that industry and service sector energy use representation is the main driver affecting GDPs, we observed this tendency.

“We ran further diagnostic scenarios with and without incorporating energy system information by sector (see Methods for more details) to investigate the extent to which the energy system model’s output information for each sector contributes to mitigation cost differences compared to the

stand-alone CGE. Comparing scenarios that include a single sector’s information from AIM/Enduse and the stand-alone model (yellows and red in Table S, respectively), the inclusion of the industry and service sector information from AIM/Enduse makes a remarkable difference in the GDP loss rate (Scenarios 9 and 5 in Table S, respectively). From the opposite side, the scenarios taking out the AIM/Enduse information for each sector (green in Table S) show that excluding the industry and service sectors similarly generates GDP loss differences compared to the integrated model (scenario 32 in Table S). Conversely, the incorporation of residential, transport and energy supply sector information given by AIM/Enduse has a small impact on GDP losses, or even has the opposite effect in some cases. Further cross sectoral diagnostic scenarios are shown in Table S4. However, the overall insights are clear, that the industry and service sectors are key in determining macroeconomic implications.

Table S4. Full list of diagnostic scenarios and their GDP loss rates in 2050. Column names are sectors, and ‘on’ and ‘off’ refer to whether AIM/Enduse information is incorporated. The red and blue rows indicate the stand-alone and integrated models, respectively. Yellow and green rows indicate scenarios that include and exclude information from a single sector given by AIM/Enduse, respectively.

	Energy Supply	Industry	Service	Transport	Residential	GDP loss rate (%)	
						2030	2050
scenario 1	off	off	off	off	off	1.1	2.4
scenario 2	off	off	off	off	on	0.9	2.3
scenario 3	off	off	off	on	off	1.1	2.4
scenario 4	off	off	off	on	on	1.0	2.3
scenario 5	off	off	on	off	off	0.6	1.7
scenario 6	off	off	on	off	on	0.5	1.5
scenario 7	off	off	on	on	off	0.7	1.6
scenario 8	off	off	on	on	on	0.6	1.5
scenario 9	off	on	off	off	off	0.4	0.8
scenario 10	off	on	off	off	on	0.3	0.6
scenario 11	off	on	off	on	off	0.5	0.7
scenario 12	off	on	off	on	on	0.4	0.6
scenario 13	off	on	on	off	off	0.1	0.4
scenario 14	off	on	on	off	on	0.0	0.3
scenario 15	off	on	on	on	off	0.2	0.3
scenario 16	off	on	on	on	on	0.1	0.2

scenario 17	on	off	off	off	off	0.9	2.2
scenario 18	on	off	off	off	on	0.9	2.3
scenario 19	on	off	off	on	off	1.1	2.4
scenario 20	on	off	off	on	on	0.9	2.5
scenario 21	on	off	on	off	off	0.5	2.0
scenario 22	on	off	on	off	on	0.4	1.8
scenario 23	on	off	on	on	off	0.7	2.3
scenario 24	on	off	on	on	on	0.5	2.2
scenario 25	on	on	off	off	off	0.4	1.2
scenario 26	on	on	off	off	on	0.3	1.0
scenario 27	on	on	off	on	off	0.6	1.3
scenario 28	on	on	off	on	on	0.4	1.2
scenario 29	on	on	on	off	off	0.0	0.8
scenario 30	on	on	on	off	on	-0.1	0.6
scenario 31	on	on	on	on	off	0.1	0.8
scenario 32	on	on	on	on	on	0.0	0.8

”

Thank you. I am basically fine, but (i) this text here does not coincide with the text in the revised paper; (ii) it is also striking that excluding energy supply makes a larger difference, and (iii) I do not understand the color coding in Table S4; what do the uncolored rows indicate?

Throughout the paper, do not call it mitigation of GDP loss. Neither results from the CGE model nor results from the integrated model are the truth. Please consistently call it something like difference or wedge between results.

Agreed. We have changed this throughout the manuscript.

Thank you!

Line 166: It is unclear what you mean by “with and without incorporating energy system information by sector”. Do you switch off the entire sector in Enduse or do you suppress transferring the information to CGE?

Thank you very much for pointing this out. We should have specified these things. We meant to say that the information obtained from AIM/Enduse includes data on all sectors, but here we generated scenarios that partially incorporate such information. For example, one scenario only uses the transport energy-related information provided by AIM/Enduse, and the rest of the sector’s model representation was maintained as in the conventional CGE, while another used residential and

industrial energy information. We clarified this point by changing the text as below.

“we ran diagnostic scenarios with and without incorporating energy system information generated by AIM/Enduse by sector (e.g. one scenario only uses data on the transport sector’s energy use obtained from AIM/Enduse)”

Okay, thank you

Lines 221-235: You clearly conclude that it is the industry and service sectors whose representation causes model results to be different. The question remains why. You mention that it is caused by the production function form and its parameters. Again, this central point must become crystal clear.

This point has already been discussed above. We would like to refer you to it.

Ok, fine; see my points above, then

Model

How do you implement the temporal structure?

- Do you solve each model for 2010 until convergence and then proceed to 2015, 2020 and so on? Or do you solve Enduse for 2010 – 2050 and then hand over to CGE and iterate?

Thank you for pointing this out. We solved each model for the entire period and then passed the model outputs to the other models. We have added a description to this effect.

“The individual models were solved from 2010 to 2050, then the results from each were input to the other models.”

Thank you, I am fine with that. It remains (i) to defend this choice, and (ii) plausibly argue that results would be similar if you solved year 2010 until convergence, then year 2015, and so on. At least, it appears somewhat unusual to me.

- Please be more specific on your CO2 budget. Is it one budget for the entire 2015-2050 period so that banking and borrowing, e.g., delayed decarbonization is possible?

Our approach is recursive dynamic and there is only an annual emissions constraint. We modified the corresponding text as follows.

“The CO₂ emissions constraint is assumed for every simulation year of the AIM/Enduse model under the mitigation scenario.”

I am fine with that; however, you should (i) related this approach to the literature, and (ii) try to give

an interpretation how allowing for banking and borrowing may impact results. Such distinction is not uncommon in the energy systems literature, to the best of my knowledge.

- Please be more specific on (social) discounting

There is no social discounting rate because our models are solved as recursive dynamic. To compute the annualized cost, we use the discount rate for individual energy use devices. We added the following detail to the supporting information and methods.

“As the models used in this study were solved as recursive dynamic, we did not consider discounting of the energy system costs. Nevertheless, the AIM/Enduse model annualizes the capital costs of energy technologies using a discount rate in the range 5–33 % (Oshiro et al. 2016)¹. The sectoral discount rate represents 5 % for power and industry, 10 % for transportation, and 33 % for other sectors.”

I am fine with that; however, please motivate (at least for me) why you can assume such high discount rates.

- Do you start with the existing Japanese power system? What are your assumption on capital depreciation?

Yes, we begin with the existing Japanese power system. For large power generation plants, we included data on each individual power plant, such as year constructed, capacity, efficiency, *etc.* We have added corresponding text, as below.

“In the AIM/Enduse model, the residual capacities of the existing power plants in operation today were calculated based on individual power plant information, such as year constructed, capacity of each plant, and expected lifetime. The total capacity was calculated based on the capacity of newly installed power plants, which was determined endogenously, as well as that of existing plants.”

Thank you. Some minor remarks: (i) please replace “today” by “in 2015” if your starting system data is from 2015; (ii) please provide all parameter assumptions, as long as they are not protected by any legal or commercial boundary, in a supplementary file for transparency and reproducibility. This would also comprise assumptions on fuel costs.

Please explain clearly why you do not account for climate damages on the economy, i.e., both production process and household utility (which hides behind linear expenditure system)!

First, we did not want to mix all of the information in a single study. Our primary focus was to

identify the macroeconomic effects associated with climate change mitigation policies, and if we were to include these impacts, we would need to isolate these two effects.

Fair point. Thanks.

Second, the degree of global climate change and its impact is not determined exclusively by Japanese climate policy but is dependent on other countries' future emissions. It would add more complexity to this study and make our primary focus unclear.

Fair point.

Third, it is quite ordinary to exclude climate damage from climate change mitigation studies and, for example, most studies referred to in Chapter 6 of IPCC AR5 do not include climate damage. Thus, inclusion of climate damage is not a necessary condition for climate change mitigation studies. We have added the following sentences related to these discussions.

Fair point.

“In this study, we excluded the effect of climate change damage on the economy to avoid complexity (e.g. isolating mitigation effects from the mixture of climate change mitigation and damage impact, and additional assumptions on other countries' emissions situations.”

Thanks; I am fine with that.

When I work on forward-looking analyses, technological change is a key driver for results

- In both an energy-system and a macro-economic perspective, technological innovation and learning curves, i.e., decreasing costs are, of utmost importance. Think of solar PV or, currently, batteries
- Your technology cost assumptions are missing. Be transparent on all of them (appendix)
- Do you incorporate decreasing technology costs? If not, defend result!
- Sector coupling is key for (very) low-carbon energy systems (e.g. see the literature by M.Z. Jacobson or C. Breyer). Please make it clearer in your model description how you deal, for instance, with hydrogen. Is it somehow included in your sectors? This would also be a credible long-term storage.

Thank you very much. We added cost information with respect to power generation, which also includes decreasing technology costs, to the Supporting Information.

Thanks, this is valuable! Can you please also add the data sources? Can you please also add data on fuel costs and other components of variable costs if there are any?

Beyond that, I have some concerns about the cost data you use as they seem somewhat outdated. Capital costs are overnight investments, right? Basically, solar PV is at 3k dollar in 2020 and 1.8k dollar per kWp in 2050. This is high. Current (2019) costs are well below (< 1.5k), future costs presumably well below 1000 and I think that also WACC in Japan should be low. Check some

random IRENA, NREL (for US) or Fraunhofer (for Germany) studies, e.g.,

https://www.irena.org/-/media/Files/IRENA/Agency/Publication/2018/Jan/IRENA_2017_Power_Costs_2018.pdf

<https://www.nrel.gov/docs/fy17osti/68925.pdf>

<https://www.ise.fraunhofer.de/content/dam/ise/en/documents/publications/studies/recent-facts-about-photovoltaics-in-germany.pdf>

Analogous for wind onshore, which is below 2k today. Offshore is a bit trickier due to idiosyncrasies and deep/shallow connection charges. Also costs for coal seem to be somewhat high for me. I understand that you need some source, preferably one consistent one for all/most cost data. At least provide some explanatory words. Ideally, update the cost data to a more recent source or run a sensitivity calculation.

Regarding hydrogen, we included it in the same manner as other energy carriers for energy demand sectors. On the production side, we also assumed that hydrogen production sectors will be introduced and provide input energy. However, within the Japanese context, our results indicate that hydrogen consumption will occupy a very small share of the final energy consumption (e.g. 0.02 %) in the mitigation scenarios up to 2050.

Fine, thanks, this is a result. Nonetheless, could you please provide some information and an interpretation which role renewable electricity plays for heating and transportation; if not through hydrogen, then maybe through battery-electric vehicles, heat pumps or synthetic fuels?

Variable	Unit	2010	2015	2020	2025	2030	2035	2040	2045	2050
Capital Cost Biomass w/ CCS	US\$2010/kW	7540	8063	8063	8063	8063	8063	8063	8063	8063
Capital Cost Biomass w/o CCS	US\$2010/kW	3813	4336	4336	4336	4336	4336	4336	4336	4336
Capital Cost Coal w/ CCS	US\$2010/kW	4338	4338	4338	4338	4338	4338	4338	4338	4338
Capital Cost Coal w/o CCS	US\$2010/kW	2704	2704	2704	2704	2704	2704	2704	2704	2704
Capital Cost Gas w/ CCS	US\$2010/kW	2174	2174	2174	2174	2174	2174	2174	2174	2174
Capital Cost Gas w/o CCS	US\$2010/kW	1122	1122	1122	1122	1122	1122	1122	1122	1122
Capital Cost Geothermal	US\$2010/kW	8716	8607	8607	8607	8607	8607	8607	8607	8607
Capital Cost Hydro	US\$2010/kW	8716	8716	8716	8716	8716	8716	8716	8716	8716
Capital Cost Nuclear	US\$2010/kW	4506	4721	4721	4721	4721	4721	4721	4721	4721
Capital Cost Solar PV	US\$2010/kW	5704	4031	3035	2588	2141	2065	1990	1914	1838
Capital Cost Wind Offshore	US\$2010/kW	5443	5704	5449	5194	4940	4797	4655	4513	4370
Capital Cost Wind Onshore	US\$2010/kW	3046	3145	2918	2725	2531	2531	2531	2490	2449
Lifetime Biomass w/ CCS	years	40	40	40	40	40	40	40	40	40
Lifetime Biomass w/o CCS	years	40	40	40	40	40	40	40	40	40
Lifetime Coal w/ CCS	years	40	40	40	40	40	40	40	40	40
Lifetime Coal w/o CCS	years	40	40	40	40	40	40	40	40	40
Lifetime Gas w/ CCS	years	40	40	40	40	40	40	40	40	40
Lifetime Gas w/o CCS	years	40	40	40	40	40	40	40	40	40
Lifetime Geothermal	years	40	40	40	40	40	40	40	40	40
Lifetime Hydro	years	80	80	80	80	80	80	80	80	80
Lifetime Nuclear	years	40	40	40	40	40	40	40	40	40
Lifetime Solar PV	years	15	15	15	15	15	15	15	15	15
Lifetime Wind Offshore	years	15	15	15	15	15	15	15	15	15
Lifetime Wind Onshore	years	15	15	15	15	15	15	15	15	15
OM Cost Fixed Biomass w/ CCS	US\$2010/kW/yr	982	982	982	982	982	982	982	982	982
OM Cost Fixed Biomass w/o CCS	US\$2010/kW/yr	290	290	290	290	290	290	290	290	290
OM Cost Fixed Coal w/ CCS	US\$2010/kW/yr	303	303	303	303	303	303	303	303	303
OM Cost Fixed Coal w/o CCS	US\$2010/kW/yr	106	106	106	106	106	106	106	106	106
OM Cost Fixed Gas w/ CCS	US\$2010/kW/yr	159	159	159	159	159	159	159	159	159
OM Cost Fixed Gas w/o CCS	US\$2010/kW/yr	34	34	34	34	34	34	34	34	34
OM Cost Fixed Geothermal	US\$2010/kW/yr	354	354	354	354	354	354	354	354	354
OM Cost Fixed Hydro	US\$2010/kW/yr	379	379	379	379	379	379	379	379	379
OM Cost Fixed Nuclear	US\$2010/kW/yr	200	200	200	200	200	200	200	200	200
OM Cost Fixed Solar PV	US\$2010/kW/yr	39	39	39	39	39	39	39	39	39
OM Cost Fixed Wind Offshore	US\$2010/kW/yr	241	241	241	241	241	241	241	241	241
OM Cost Fixed Wind Onshore	US\$2010/kW/yr	64	64	64	64	64	64	64	64	64

Minor points

Language must be improved

- Abstract: Try to shorten to at most 100-125 words

The Nature Communications guideline for authors says that abstracts should be approximately 150 words, and we think that our current abstract meets this standard.

Okay, that's fair. I did not know that.

- Abstract: What do you mean by 100% lower losses? Are they zero?

It should be 50% and we revised this accordingly.

Thanks!

- Line 24: Nationally

Thank you. We revised this.

Thanks!

- Line 33: 5th IPCC reports states 2-6%

Thank you. We revised this.

Thanks!

- Line 48: "Some argue..." Who?

We have added the corresponding reference.

Gherzi F. Hybrid Bottom-up/Top-down Energy and Economy Outlooks: A Review of IMACLIM-S Experiments. *Frontiers in Environmental Science* 2015, 3(74).

Thanks!

- Figure 4c: Parentheses do not match

Changed.

Thanks!

- Line 359: word "its" not appropriate

Thank you. We revised this.

Thanks!

- Line SI57: check table heading

Thank you. We revised this.

Ok, thanks

Further points

- Line 22: climate change is a particular challenge for developing countries

Revised

Thanks!

- Line 69: what about seasonal storage, which should become relevant under 80% decarbonization?

We agree that seasonal storage is an essential element. The actual results are, however, that there are buffers to deal with seasonal fluctuations, such as fossil fuel CCS thermal plants, in the 80 % reduction scenario, and thus, even if we consider battery storage for seasonal fluctuations, it would remain unused due to the cost competitiveness. The no CCS scenario also uses gas thermal plants to adjust for seasonal differences.

Thanks, this is plausible. Could you nevertheless please provide a short explicit clarification of this point in the main text (as in this comment here), potentially also in connection to flexible electrification (if there is any in the model)? Could also be footnote/endnote. I suppose that many readers may wonder about that.

- Figure 1: the right column with arrows is hard to understand. Find a better format for representing exchange between models

We added numbers to the arrows so that we can specify what each arrow means.

Thanks, this helps! Remark: didn't you alter the power generation shares to another variable in (1)?

- Line 103 and Figure S1 state that economic growth is a model input. How does this conform this GDP being the central output of interest?

This model input should be specified for the baseline scenario only.

Okay

- Line 109/Figure 2f: I don't see strong electrification in the figure

The term "strong electrification" is exaggerated. On the other hand, the electricity share in the final energy demand is higher in the mitigation scenario than in the baseline scenario because, with

respect to the baseline scenario, the total final energy demand decreases in the mitigation scenario. Thus, we slightly modified the corresponding text.

Fine, thanks!

- Figure 2c: How can final energy demand be larger primary energy? Imports? Heat pumps?

Indeed, secondary energy trade was not shown in that figure, but we have now added it.

Fine, thanks

- Figure 2a: What drives changes in the baseline?

As mentioned in the main text, the change in the total primary energy in the baseline scenario is due to decreasing population and modest economic growth. As for the share by fuel type, the phase-out of nuclear and lower energy price of coal drive changes to the baseline scenario. We added the following text.

“The main differences relative to the base year in the baseline 2050 model are the lower price of coal relative to other fossil fuels, and the share of nuclear energy, which reflects the current societal attitude toward nuclear power that limits new construction (Figure 2b).”

Okay, thanks

- Figure 2b: Why is there a marked kink in 2015 Oil w/o CCS?

This is due to the suspension of nuclear power generation after the accident at the Fukushima Daiichi plant. After 2020, power generation from oil will be replaced by restarting nuclear power and new construction of natural gas and coal plants, which are less costly than oil.

Okay, thanks, that's plausible

- Figure 2g: What are the low-carbon liquids, gases, and solids?

We have split these out.

Okay, thanks

- Figure 2h: Please explain the extreme high CO₂ price in 2050. Model-ending effects?

This is due to the technological availability in the AIM/Enduse model. We added an explanation of this.

“The price of carbon in the mitigation scenario increased over time and reaches approximately \$1000/tCO₂ in 2050. This high carbon price is due to the technological availability in the AIM/Enduse model”

Thanks, but still not entirely clear to me. You mean the availability of certain technologies in Enduse? Which ones do you specifically mean?

- Figure 2: Please use one clearly distinguishable color for one data point!

Agreed.

Fine

- Figure 3: Curtailment numbers are quite high; please comment on connection to (conceivable) sector coupling!

We have added a comment.

Ok, thanks

- Lines 136- 137: Please state the “usual” numbers!

We have included these.

Okay, thanks

- Line 211-212: If you include an energy system model in a CGE model, your results indicate lower GDP losses. This does not mean that GDP losses will be in this range. As said, external validity is doubtful for me.

We concur with this point and appreciate it. We changed the text accordingly. As discussed above, external validity is impossible to prove in this paper and we need further studies based on results obtained by other modelling teams. As we have already responded we have added the following statement

“For now, this study’s approach, and the derived implications are only applicable to Japan and within our modelling framework. We need further application fields and similar attempts from other teams, which can eventually show the external validity of our findings.”

Thanks, fine; see my comment above

- Table S1: Sample size, R-squared

We have added these.

Thanks

- Table S3: Dispensable in this form; can be condensed to text

It might be possible to explain by using only text, but we prefer to keep this table to show precisely what we did. There are no strong disadvantages to keeping the table.

You are right, okay

Reviewer #2 (Remarks to the Author):

GENERAL COMMENTS

The authors have adequately addressed specific reviewer comments but have not fully addressed general comments, especially the limitations and contributions of this paper:

- The primary contribution of this paper remains model integration. One goal of model integration is to provide more realistic cost estimates, whether they be higher or lower than a stand-alone economic model. In this application, reported cost estimates are reduced with model integration.
- If the cost of greenhouse gas mitigation can be calculated from energy models alone, then why is an economy-wide economic model needed? The authors could (and should) make a stronger case for the benefits of using a CGE model.
- The new section starting at line 173, "Mechanism causing the differences in macroeconomic implications between the stand-alone and integrated models", has created additional confusion, even though its intent was to clarify. I am not convinced that differences in primary factor productivity cause the difference in GDP cost between the stand-alone model and the integrated model. Implying causality is too strong. It seems plausible that changes in primary factor productivity and GDP losses are both caused by some other exogenous input to the models. The authors have decomposed total change in GDP into sector-specific components, which is useful, but I don't see a causal relationship. I suggest backing off the claim of causality, but still report results in Figure 5.
- What are next steps for model integration? An ambitious research agenda would replace the electricity sector in a CGE model with an electricity dispatch model. This would provide full simultaneity in solving model equations, and a well-defined interface between the dispatch model and other sectors of the economic model.

This remains an interesting paper, primarily because of linking an electricity dispatch model with a general equilibrium economic model, and the requirement to match hourly electricity demand and supply. The use of an electricity dispatch model, with hourly load shapes, is a big step beyond modeling the electricity market with load duration curves. This is especially important for the intermittent supply of wind and solar power.

ADDITIONAL SPECIFIC COMMENTS

Please define and clarify "curtailment" in the paper. I think the authors are referring to excess wind or solar power that cannot be used or stored. Or it could mean electricity demand that cannot be met and load must be reduced, requiring some form of demand side management.

The revised sentence at lines 48-49 is helpful: "Traditionally, CGE models tend to project policy costs that are higher than those of energy system models." Please make the same modification to the fourth sentence in the Abstract.

Reviewer #3 (Remarks to the Author):

Review of revision 1 "Energy Transformation Cost for the Japanese Mid-century Strategy: Energy System Feedback Effects in an Economic Model"

The authors have responded competently, and I think the paper has been significantly improved. I

am a little in doubt regarding the degree of novelty of this work, but I recommend publishing the article provided that editorial adjustments are made.

I have one possible major issue: I notice that the results have changed (Figure 4, Figure 6, Table S3). Generally this could indicate that results are not robust, but I assume changes are due to the change in the method for presenting AIM/Enduse data as input to AIM/CGE. However, Figure 4a now indicate that policy cost is zero in 2020, as opposed to the previous version with more than 1% GDP loss in 2020. How do you explain such a change?

Minor comments:

Abstract line 15-16: This sentence should be clarified. Do you mean "GDP loss" as "one of the key elements"? Do you mean something like: "Our findings suggest that this type of integrated approach would provide new insights by providing improved estimates of GDP losses - one of the key elements for setting national climate policies."?

Line 39-41: The text suggests two models while you use three models – you may try to reformulate.

Line 59: "In IAMs, they are represented in some way" – you may want to reformulate the sentence.

Line 60 Suggestion: "Further literature list"  "An extended literature list".

Line 97: "Application of this framework would be beneficial for Japan's climate policies" – why? Because "necessary" policies would be easier to implement if new model estimates show lower GDP losses? The reader might think that an integrated model results in lower GDP loss, which touches upon the external validity problem. Please state why the application of the framework would be beneficial for Japan's climate policies.

Line 117: Sentence mixes past tense (increased) and present tense (reaches).

Figure 4: You may want to label the Figure 4a and 4b Y-axis differently, to improve readability (GDP loss versus equivalent variation). Perhaps you could connect the scatter points in figure 4c and 4d to simplify the interpretation by indicating the order of the points?

Figure 4 c) seems surprising, since the more constrained "no CCS" observations have lower GDP loss. The authors have commented on this earlier. It would still be interesting to check if household utility levels have decreased in the "no CCS" scenario, and to report equivalent variation.

Figure 5: I am not sure what the black arrows are indicating?

Figure 6: New results for the integrated model. The sign of the service sector is reversed in 2030 compared to the previous version, and the output change is relatively larger in 2050 than in the previous version.

Line 346: "We executed the model for two iterations" – but you have run the model for 5 iterations (Figure S8-S13). Do you report the results after 2 or 5 iterations? Please clarify.

Line 364: "Dietary preferences" sounds strange – do you mean "consumer preferences"?

Line 535: Please correct the sentence, and explain set i.

Supplementary: line 34: "We further run the iterations until 5" – please reformulate.

Supplementary: line 35 in the supplementary information refers to the supplementary information. Needs to be corrected.

Supplementary: line 44: "causing the RMSE rate" – you have stated that you don't use RMSE, and have supplied the error rate formula you have used. Please correct.

Figure S11: The caption is identical on Figure S10 and S11. I assume S11 shows the mitigation scenario, not the baseline scenario. Please correct the caption.

To all reviewers

Thank you very much for giving us another chance to improve our manuscript. We appreciate the reviewers' comments and have made substantial revisions throughout the manuscript. The main changes are as follows.

- 1) We have conducted an additional sensitivity analysis of a key parameter, which showed the robustness of the results. We believe that the added material strengthens our results and supports our conclusions.
- 2) We have explained, and edited existing text concerning, the central mathematical mechanism.
- 3) We have amended the justification for using the economy-wide economic model, and the scientific contribution of this study.

We believe that the revised manuscript meets the standards of *Nature Communications*. The responses to each comment are shown below. The blue text is our responses to the second-round of reviewer comments. For some of the comments, we felt that it was better to attach the previous round of comments (in green) and our responses (in pink). The second round of reviewer comments are in black.

Reviewers' comments:

Reviewer #1 (Remarks to the Author):

Dear authors,

Thank you very much for paying attention to my comments. Many questions have been answered. According to my judgement, the paper improved, but still needs revision.

In a nutshell, (i) the maths must be communicated more clearly and precisely, (ii) the central mechanism still needs some more explanation, precision, and intuition, (iii) now that the authors provided me with a better understanding of the model, they must devise some efforts to convince the readership that their finding is robust and not an artefact of parameter choices, (iv) the paper needs some editorial polishing concerning language and write up. I am willing to give a complete list of my editorial issues after the next round. I think that there are still some changes to the text to come first. I hope that this is okay for the process. (v) Relating to code and data availability, would it be possible to provide a “ready-to-use” version that contains exactly the model specifications you used for this paper? This would increase transparency and reproducibility.

Thank you for these additional thoughtful comments, which helped us to improve the manuscript. Our responses to each comment are shown below. Regarding comment (v), which will not be touched upon later, we are not ready to release the full model code. We have queried this with the editor, and were told that it is not mandatory to do so. The code still needs cleaning and the user's manual to operate the model is not finished. Nevertheless, if the reviewer needs the code for this review, we could provide it for review purposes only, although we do not know how best to deal with any licensing issues regarding the statistics.

Comment 1

- The application to Japan is illustrative and should stay in the paper, but I see the contribution in the method. I highly doubt the external validity as numerical results are driven by numerous assumptions on functional forms and input data.

Thank you for pointing this out. We fully concur that the external validity depends on the functional forms and parameter assumptions. We have tried to make the assumptions and functional forms as transparent as possible in this revision. Meanwhile, we think that it will be difficult to prove the external validity within this paper because attempts by other modelling teams using the same (or similar) approach would be required. We added discussion related to this validity issue to the corresponding section.

“For now, this study’s approach, and the derived implications, are only applicable to Japan and within our modelling framework. We need further application fields and similar attempts from other teams, which can eventually show the external validity of our findings.”

Of course, you cannot substantiate or prove external validity here. But most importantly, the paper should not suggest so. Models can help in uncovering relationships, identifying sensitive parameters, illustrate relevant tradeoffs, and give an estimation of orders of magnitude, which is what the paper does. I am basically fine with that formulation. Yet I think that the approach as such is applicable to any country, but numerical findings may be rather specific to Japan. Please consider formulating accordingly.

We have reformulated the text as follows: “For now, this study’s approach and the implications thereof are applicable only to Japan, within the context of our modelling framework. Application to other fields by different modelling teams is needed to demonstrate that our findings can be generalised.”

Comment 2

The revised version has improved. However, the central mechanism (the why) is still not sufficiently clear. The result as such (“lower GDP losses when using integrated model vs standalone model”) is interesting, but it is more relevant why this occurs and whether this is robust across a range of plausible parameter choices. You can basically generate a range of numbers by picking one or another plausible assumption on functional forms or parameter values. However, leaving as much as possible constant and then convincingly explaining why results are different is central here. Now that I (and other readers potentially also) better understand the forces that drive results, it remains to convince the audience that the difference is not an effect of mere parameter choices (e.g., numerical value of the substitution elasticity btwn energy and the capital-labor composite in the standalone CES), but robust and rooted in the methodological advancement.

We believe that this has already been explained (see also Figure 5). However, although it might be repetitious, we will explain again presently. There are two major points; firstly, capital efficiency is higher in the integrated model than in the stand-alone model, because emissions reduction is cheaper using AIM/Enduse versus the typical CES function. Since the parameters of the CES function of the AIM/CGE stand-alone model rely on the existing literature, they represent only historical technological changes and energy saving trends. By contrast, the parameters in the integrated model rely on the energy system model outputs; therefore, it can represent unexperienced changes in the technological parameters assumed in the energy system model. Second, the amount of electricity generated differs between the approaches; less is produced with AIM/Enduse than with the AIM/CGE stand-alone model. Moreover, the difference in total electricity consumption between the baseline and mitigation scenarios is much smaller in AIM/Enduse than in AIM/CGE. The higher power generation requires additional capital usage in the stand-alone model compared with the integrated model. Regarding the parameters chosen for the CES function, we performed a sensitivity

analysis by considering variation in the substitution elasticity; the results seemed robust, as we will discuss later. In addition, we have edited the explanation of Figure 5.

“We conducted a sensitivity analysis, varying the elasticity substitution between energy and value-added from 0.2 to 0.8, taking the range from the literature¹. The results showed that the cost differences associated with variation in the substitution elasticity parameter are much smaller than the differences between the integrated and stand-alone models, although the climate policy costs varied (see Figure S4). This implies that even if the wide range of values for the substitution elasticity parameter (as seen historically in the literature) is considered, future technological changes cannot be expressed. Therefore, integration of the energy system and economic models improves the representation of technological change.”

- Therefore, authors should streamline the paper and feature the central mechanism more prominently. In the current manuscript, it becomes clear only in lines 222 - 232

This is an excellent comment and we really appreciate it. The central mechanism for changing the macroeconomic implications is adjusting the productivity of the primary factors (labour and capital) which consist of a major part of value-added.

Okay, thanks, but please get the econ terminology clear and aligned with standard wording. After talking to a fellow macroeconomist, I have some doubts whether the wording “value added” applies here. To the best of my knowledge, value added basically means value of output (quantity times price) minus value of inputs. If this is the standard terminology, please clearly explain in the paper to which concept “value added” refers here. Please also check the term “productivity” here.

We think that our use of value-added is correct. The constant price in GDP measured by the production side is referred to as value-added in the context of national accounts (e.g. UN national accounts).

Regarding “productivity”, it is also common that the productivity of A (e.g. capital) is taken as the measure of how much output is generated by a unit amount of A. We use this terminology as it is (e.g. productivity of capital). We have added an explanation of value-added to the main text.

This is because the total primary factor inputs are constrained exogenously each year.

Please relate this assumption to the related (economic) literature.

We refer to the assumptions of the well-known global CGE models, EPPA and ENVISAGE. We have extended the explanation of the capital and labour assumptions as follows:

“The capital and labour inputs change dynamically with population and demographic changes, and GDP growth.”

The reason why the GDP losses are lower in the integrated model than in the stand-alone model is that the primary factor productivity is higher in the integrated model than in the stand-alone model.

Okay, this is important and it is good that you make it explicit. However, the central point to your analysis is why “primary factor productivity” (check term) is higher in the integrated model.

This is explained in the text on lines around line 190-220 where we added and amended some text.

Hence, the productivity shifts are mainly driven by two factors. One is the productivity decreases associated with emissions reductions in energy end-use sectors such as manufacturing, transport and service sectors (e.g. capital replacement by expensive but energy efficient ones).

Word “hence” does not apply here. I understand that tighter emission caps may reduce productivity (check term) in energy end-use sectors. But why are these decreases lower in the integrated model?

We agree that “hence” is not appropriate here and have changed this.

The lower decreases in the integrated model are due to the AIM/Enduse outputs. AIM/Enduse can reduce the emissions more cost-effectively than the CES function in the conventional CGE model. However, this is not directly derived from the results, because the parameters for the CES function of the AIM/CGE stand-alone model rely on the existing literature, which can only represent historical technological changes and energy saving trends. In contrast, the parameters in the integrated model rely on the energy system model outputs. Therefore, the integrated model can represent unexperienced changes in the technological parameters of the energy system model.

The other factor is sectoral allocation changes in primary factors.

Do you mean that sectoral allocation of capital and labor due to tighter emission caps occurs in a different manner for the integrated and the standalone model? Do sectoral output levels also change, i.e., does demand decrease?

Yes, they differ across sectors and demand decreased. The sectoral allocations are shown in Figure 5.

For the first factor, Figure 5(a) illustrates the capital input efficiency of major industrial sectors (top 10 industries, which account for 95% of GDP in the base year) in the mitigation scenario compared to the baseline scenario for stand-alone and integrated models in 2050.

Here, we define the capital input efficiency as capital input per output for each sector.

Terminology “relative ratio of capital efficiency (-)” in 5a unclear. Efficiency is a ratio. So you mean a ratio of ratios here? Please clarify! What about “labor input efficiency”?

Thank you for pointing this out. As described, the indicator shown in Figure 5a takes the ratio of capital efficiency between baseline and mitigation. We have revised the caption accordingly.

Here, the main concern is capital efficiency rather than labour efficiency because the investment information derived by AIM/Enduse is incorporated into the integrated model.

Higher values indicate that additional capital inputs are needed in the mitigation scenario compared to the baseline scenario.

Understood. Please communicate that this is not an assumption on the production process, but an outcome of your model, as I understand it.

You are absolutely correct; they are the model results. The stand-alone model is based on the conventional CES function, whereas the integrated model relies on AIM/Enduse outputs. We have added an explanation to clarify this point.

In general, the stand-alone model requires larger capital inputs than the integrated model in the mitigation scenario.

Understood! But why?

We explained this in the original manuscript; the integrated model relies on the AIM/Enduse outputs, which differs from the conventional CES. Since the parameters for the CES function of the AIM/CGE stand-alone model are informed by the existing literature, they only represent historical technological changes and energy saving trends. By contrast, the parameters in the integrated model rely on the energy system model outputs, and can therefore represent unexperienced changes in the technological parameters assumed in the energy system model. We have clarified this in the main text.

We can roughly compute to what extent the GDP losses are associated with these capital productivity losses by multiplying the value-added with the capital efficiency changes (Figure 5 (b,c)).

1) It may be redundant, but for me, as an economist, the concept of “value added share in GDP” is not entirely clear here. I also talked about this term to a fellow macro-economist who could not unanimously interpret it. Please explain carefully and clearly!

Thank you for this comment. As we have already stated, the term “value-added” is based on national accounts and is also used in UN national accounts (<https://unstats.un.org/unsd/snaama/Basic>)

2) The sector “Other services” seems to be driving your results on GDP in the stand-alone model and thus of your paper. Please clarify what this sector is; also please clarify how reliable assumptions of the production process and the “value added” of such an aggregate sector are. Please also clarify why it is of such large importance concerning “value-added” changes in the stand-alone model and of no importance in the integrated model.

We very much appreciate this comment. We would not say that value-added is less important in the integrated model. In this paper, we are pointing out that the conventional CES function in the energy end-use sectors differs from what an energy system model simulates and the

CES function might not be suitable for representing energy consumption. To this end, we are proposing that the outputs of energy system models be used. We discuss the differences in value-added to better understand why the GDP varies between the stand-alone and integrated models.

We now provide definitions of the other service sectors in Table S-6.

3) Figure 5c is nice. However, I do not get why the blocks are located differently concerning the zero line. As far as I understand it the total change to GDP from emission caps is negative. Figure 5c seems to devise a sectoral composition of these GDP changes, right? Please clarify. And why does it appear positive in the upper bar and negative in the lower bar? Please also explain what the dots in the bars represent!

Thank you very much. We now describe the meaning of the dots in Figure 5c. It is important to define both negative and positive; we show the GDP loss (positive) associated with capital productivity and, to avoid confusion, have defined the dots more clearly.

They eventually account for 1.3% of the total value-added (GDP). Hence, the productivity differences between the stand-alone and integrated model are mainly caused by the differences in the function form and parameters, particularly for the value-added and energy bundle.

If this is the central explanation for why the integrated model shows lower GDP losses from tight emission caps, then this is somewhat unsatisfactory. Functional forms and parameter choices are quite debatable, e.g., the 0.4 for the elasticity of substitution below. Maybe, if you increase this value to 0.5, which may also be supported by the literature, then results change, eventually, also qualitatively? Please (i) clearly explain whether my assertion is correct or wrong; and (ii) if it is wrong, defend the choice of (your functional forms and) parameter values! I think I am not picky here - to me, this is central to understand how much you exactly add to the literature and how relevant your results are!

Here, we use a CES function wherein the substitution elasticity is 0.4 for the stand-alone model.

Substitution elasticity between capital and labor or x and energy? Please defend the choice of the value 0.4 or devise sensitivity calculations.

This is an excellent question. It has already been demonstrated that the substitution elasticity parameter can affect the mitigation cost (Jacoby, 2006).² Therefore, we conducted a sensitivity analysis varying the elasticity substitution between energy and value-added from 0.2 to 0.8; the range is taken from Edwin van der Werf (2008).¹ The results show that the cost differences associated with the substitution elasticity parameter are much smaller than the differences between the integrated and stand-alone models, although the climate policy costs varied (see the figure below). We have added a figure to the Supporting Information and additional discussion to the main text.

Figure S4 The GDP loss rates associated with variation in the substitution elasticity between energy and value-added in the stand-alone model

The future autonomous energy efficiency is adopted.

Please clarify what you mean by that! See also my comment below.

The integrated model uses a function of the same form, but the additional investment and energy inputs are provided by AIM/Enduse, by endogenizing the CES shift parameters (sector-wise additional investments are shown in Supplementary Table 9).

This seems to be at the heart of your methodological contribution; a new/improved way of linking CGE and energy system modelling. Please provide an explanation/intuition for the CES shift parameter here. In my view, this would communicate the central mechanism at play here more clearly. Moreover, I cannot link your comment to Table S9, which contains specific investment cost, lifetime, and OM costs for electricity generation plants. Also, I have some problems in fitting the “additional investments” into the story. Could you please clarify this? See also my comment below (where the maths is).

We apologise; Table S9 should have been Table S8. This is part of the central mechanism, which is shown in Figure 5 and discussed accordingly.

The second factor, effects of sectoral primary factor allocation change, is mainly driven by the power generation sector, where the electricity generation in the mitigation scenario

compared to the baseline scenario is 20% higher in the stand-alone model, whereas it is almost same in the integrated model (Figure 5).

Figure 5d), right? Then I do not understand the label on the vertical axis. Please also carefully check all other text in Figure 5d)!

For me to get it right: electricity generation in the mitigation scenario is higher in the stand-alone model. This seems to be the second main driver of the central result. More electricity required due to mitigation higher GDP effect because the exogenously limited production factors need to be allocated to electricity generation, right? In any case, it remains the question why this is the case. This is not sufficiently clear: why does sectoral primary factor allocation change? Why is electricity generation higher? It becomes somewhat better understandable in the maths in the appendix. But please write it clearly into the main text as this drives your central result!

Your understanding is correct.

As discussed, the GDP losses associated with the sectoral allocation of primary factor changes are mostly attributable to the electricity sector, so we limit the discussion here to electricity.

The electricity generation is higher in the mitigation scenario of the stand-alone model; this is because of how the CES function and fuel prices behave in the stand-alone model, and how cost minimisation works for AIM/Enduse in the integrated model. We have slightly edited the main text for clarity. As explained above, the higher electricity generation is due to a limitation of the conventional CES function, which relies on historical data.

Regarding the figure axis label, it has been modified as suggested.

There are certainly differences in power technological shares between the stand-alone and integrated models, but, in summary, it seems that the difference in total electricity generation between the models is the dominant factor.

Okay! How can you be sure that these “certain” differences are not a dominant part of the story?

This is because the total value-added is given by macroeconomic indicators, like GDP. Figure 5d clearly shows that the value-added share of each technology in the power sector differs; that is why we used the word “certain”.

The stand-alone model requires incremental capital and labour inputs that account for 0.4% of GDP compared to the integrated model (Figure 5 (d)).

Please explain more clearly. Do you mean that more factors (capital, labor) must be allocated to electricity generation in the standalone model, and because factors are exogenously constrained (i.e., no endogenous investment/depreciation, no endogenous population growth/labor supply decisions) GDP losses are higher? If this is true, why is this the case? According to my judgment, you must provide an intuitive, clear, and understandable explanation for that. Analogous to my comment above.

We are afraid that the reviewer's understanding is incorrect; more production factors must be allocated because the total electricity demand is higher in the stand-alone model.

The total electricity demand is determined by the energy consumption represented in the energy end-use sectors. The total energy consumption in each energy end-use sector is represented by a CES function, as mentioned above, and the fuel-wise share is determined by a logit function in both the stand-alone and integrated models. The share parameters in the logit function are endogenously determined by the integrated model obtained from the AIM/Enduse results, whereas they are exogenous parameters in the stand-alone model.

After reading the maths below, I get an understanding for the difference. It seems to me that the central difference between integrated and standalone is that these share parameters are exogenous vs endogenous. Could you perhaps provide a table containing this and all other decisive differences (appendix)? In any case, the choice of the exogenous parameters for the shares of technologies in electricity generation are highly sensible for the results. You must defend this choice and ideally provide sensitivity calculations.

That is part of the story, as we have illustrated using Figure 5. The total energy consumption for each sector, and the investment needed to save energy or shift the energy carrier, is important, as is how the power generation is determined. We present the information related to power generation in Tables S9 and S11.

In addition to the two main mechanisms mentioned above, productivity changes and sectoral shifts in other sectors certainly occur, but are relatively minor.

You say "certainly". From this I infer that you did not explicitly check that they, and their effect, are relatively minor. Please clarify or do so. Again, I do not want to be picky, but the central mechanisms deserves this great clarity and must be bullet proof!

We have checked, but the remaining sectors contribute less than 5% of the total GDP, so changes therein would not influence the GDP data overall. If we included all of the information, it would merely confuse the readers. Therefore, we have limited the number of sectors shown in Figure 5 and state that they account for 95% of GDP.

In summary, we added one section to the results to discuss the points raised above and described the corresponding equations and parameter assumptions in more detail in the Supplementary Notes.

Thank you, fine. However, requires revision.

We have tried to answer all of your questions, and hope you are satisfied with our explanations.

Comment 2 (cont.) about formula

3.1. Energy end-use sectors other than household sector in the CGE model

We begin with the representation of the energy end-use sectors (Table S5).

S6, right? Are all (non-household) energy end-use sectors production sectors according to the right column of new table S6 and vice versa? What about agricultural sectors?

Thank you very much. This should have been Table S6, which shows agricultural sectors as well. In Table S12, we can see that cereal corresponds to the aggregate of rice, wheat, and other grains for household consumption goods. Other goods are considered separately.

The value-added and energy composite inputs are determined by multiplying a coefficient by the output from the energy end-use sectors (Equation 1). Then, value-added and energy composite are combined with the CES function (Equation 2). Labour and capital inputs are further nested in the CES function, as shown in Equation 3.

Generally, please motivate the nest structure (cf. van de Weerf, Production functions for climate policy modelling: an empirical analysis, Energy Economics 30, 2008); please also motivate the choice of parameter α – assumption or calibration? Please also indicate how production activities a are combined to determine GDP which is the final outcome of interest.

Thank you for this comment. We have modified the text accordingly. The α is the calibrated parameter. Total GDP is accounted for by the final consumption amount; we have added this to the text.

“As with many other CGE models, AIM/CGE uses a KL-E type multi-nested CES production function (van de Weerf 2008).”

“The GDP is accounted by the total final consumption.”

Check exponents in (3)

Please, clearly list all of them; the relation to the table S6 remains unclear. Also check your use of symbols in MS Word. On my computer, I have some strange notation, e.g., “aCE” here.

Please clarify the units of the variables (for me). Is it monetary or “real”?

Thank you for this comment. We have added units for each variable.

$QENE^B_a$ is the energy input of sector a in the baseline scenario,

It is unclear to me whether this energy input can be any energy type (oil, gas electricity)

Thank you for this comment. This is the total energy, comprising all energy types; we have clarified this.

$ivae_a$ is an input coefficient of the output of sector a ,

Seems a somewhat arbitrary choice to me; assumption or calibrated?

Is the calibration base year the year 2010?

It is 2005. Thank you for this comment; we have added explanatory text. This is calibrated by base year social accounting matrix and energy balance table.

Please motivate and defend this logit sharing approach!

We have added the following text:

“The energy carrier shares in the energy end-use sectors are calculated using McFadden’s (1981)³ logit share equation. The concept underlying the logit is that the decision makers determine the share of each element under a certain probability distribution function. This function form is applied in some other IAMs (GCAM⁴ and IMAGE⁵). For further theoretical discussion of the logit sharing mechanism, see Clarke and Edmonds (1993)⁶.”

$a \in AEnd$ is a subset of the production activity and energy end-use production activity (e.g. industry, transport and so on),

Please clarify and be precise, you mean it is a subset of a union of two sets? What differentiates production activities from energy end-use production activities?

The production functions differ between energy end-use and energy transformation activities (shown in Table S6). We have clarified this point.

$c \in CENE$ is set of energy commodities (coal, petroleum products and so on), which is a subset of all commodities,

For clarification, does it also include electricity from the power generation “logit module” below?

CENE includes electricity. However, c is a set of commodities different from the production sectors. Therefore, the power generation logit not involved.

$SHENE_{c,a}$ is the energy consumption share of energy commodity c and production activity a ,

For clarification: do you mean “in” or “and” production activity a ?

Agreed. We have changed “and” to “in”. Thank you.

aec is the fuel-wise energy preference change parameter of commodity c .

This parameter seems rather arbitrary to me; what does “preference” mean here? Please motivate or, ideally, refer to some literature you base this approach on.

For the stand-alone model, $beta_{ae}$ and $deltac,a$ are calibrated by base year information.

Autonomous Energy Efficiency Improvement (AEEI) in the stand-alone model ata is one of the critical parameters determining energy consumption. We adopted a uniform AEEI across

energy end-use sectors for each year, which is associated with GDP growth. For the years that assume more than 1% of GDP annual growth, the AEEI is 1%, and half of the annual GDP growth rates are assumed for the other the years. The fuel-wise energy preference change parameter aec is set annually to 1%, 0.5% and -0.5% for electricity, gas and coal, respectively, which represent the fuel shift from conventional solid and liquids to gas and electricity carriers.

Please be precise! AEEI is not 1%; rather its growth rate, right? In this case, the starting value is missing. Please also defend the choice of the parameter values for AEEI. This appears to be central for the difference in your numerical results. Please also better explain the preference parameter. Why can you exogenously pick values? Why is it in percent? Is this related to decarbonization requirements? If so, how?

This is correct, it is the AEEI growth rate; we have amended the text accordingly. The starting value is in accordance with the base year social accounting matrix. The preference change parameters are intended to represent the fact that energy carriers have shifted from solids to liquids, gases, and electricity, due to environmental regulations related to air quality or the need for cleaner energy. The values are set arbitrarily, but the parameter assumptions need to be discussed in the context of the validity of the baseline energy consumption pattern; we have added explanatory text accordingly.

Another point: how does SHENE relate to QENE in the standalone model?

If the *SHENE* is the sum of all energy commodities, c , it takes a value of 1, which means that the total energy consumption is *QENE*. We have clarified this point.

“*QENE* shown above is the sum of the individual energy carriers determined by this equation.”

With respect to the integrated model, the $SHENE_{c,a}$ and $QENEB_a$ are fixed based on the AIM/Enduse model results and endogenise $\delta_{c,a}$ and at_a .

Please either describe the AIM/Enduse in equations (roughly) or refer to a citable source and explain in more detail how the model determines SHENE and QENE!

The equations used for AIM/Enduse are all available in Kainuma et al. (2003),⁷ although some of these were updated, specifically pertaining to power generation, by Oshiro et al. (2015)⁸. Simple cost-minimising linear programming was used, as already explained in the Method.

Then, we can derive the input coefficients of capital in the baseline scenarios after obtaining the simulation results of the baseline scenarios, as shown in Equation (6).

Are these input coefficients (capital input over output) equal to the “relative ratio of capital efficiency (-)” in Figure 5a? It seems. Please be also clear about your terminology! If so, how do you use the ifa to arrive at GDP changes?

No, they are not. The “relative ratio of capital” means capital efficiency in the mitigation scenario relative to the baseline scenario. As stated, ifa is applicable only to the integrated model. The information in Figure 5a can be represented by QF/QA .

$f \in Fcap$ is a set of capital and a subset of primary factors

I cannot find $fCEcap$ in eqs 5 and 6 and I do not want to guess. Please be precise and exact! Does set of capital and a subset of primary factors mean $fCEcap$ is a set union? Which primary factors do you mean?

Thank you very much for pointing this out and we apologise for the error; that should be $Fcap$ and we have modified the text accordingly.

$AddInv_{f,a}$ is the additional investment cost associated with mitigation costs provided by AIM/Enduse,

Please clarify the nature of these additional investments (to me)! Investments into what?

Capital input was already specified, but we further clarify the text to avoid misunderstanding, as follows:

“For the capital inputs of the energy end-use sector, additional investment costs (associated with mitigation scenarios) relative to baseline scenarios, which are given by the AIM/Enduse model, have been added to the baseline inputs, as shown in Equation (7).”

3.2. Power generation in the CGE model

The total consumption of electricity is determined by the demand side representation shown in the previous subsection (and the latter for household).

Please apologize, but where do I find the total consumption of electricity above?

Multiplying $QENE$ by $SHENE$ yields the electricity consumption; this has been clarified.

where $aCEEly$ is a subset of production activity and electricity production activity (e.g. coal, solar PV and so on),

Please motivate/defend the use of logit sharing in representing electricity generation.

This was done in accordance with the logit literature.

$cCEEly$ is a subset of commodity (electricity),

Subset containing what?

Only electricity, as described.

$SHAC_{c,a}$ is the electricity generation share of production activity a ,

I am sorry, but what exactly do you mean? This formulation suggests that each production activity a (wood products, iron and steel, ... according to Table S6 if you mean the same “a”s

as above) generates electricity. I guess you mean the share of an electricity generation technology in total electricity generation.

SHAC is used in equation 7, where the activities a are limited to the electricity generation sectors.

For the stand-alone model, $\delta_{a,c}$ is calibrated using base year information.

Please explain (to me) how you decarbonize electricity generation in the standalone model; it is solely using the CO2 cap, right?

The carbon price shifts the share.

Household LES function parameters are updated recursively based on income elasticity. Electricity and biofuel used in transport, are not accounted in the base year social accounting matrix and, thus, we introduce the initial parameters for $\delta_{c,ch}^h$ and $\delta_{c,a}^{en}$ by calibrating 0.1% of the share in each energy consumption in 2015 and 2020. They are then updated afterwards to one third of the value of petroleum products in 40 years. They are same as used in the SSPs quantification. For the integrated model, the parameters and were determined endogenously based on the energy consumption and investment needs for other manufacturing goods computed by AIM/Enduse.

Please (i) motivate using this household representation (literature), (ii) explain whether there is any market clearing between household consumption and production of goods and how it is implemented, and (iii) related to that if the is import and export of goods and services.

The function is derived from a function originally defining how spending on individual commodities has a linear relation with total consumption spending, based on the assumption that each household maximises a “Stone–Geary” utility function subject to consumption expenditure constraints. The parameters of the LES function other than food were calibrated based on income elasticity values (Nganou, 2005)⁹. The income elasticity of food demand for each region and commodity was from Bruinsma (2010)¹⁰.

We have added the above text.

The market clearing is simply formulated as “consumption = supply”, which consists of production – exports + imports + depreciation – stock change.

Comment 3

→ Please explain the concepts of energy and material balances in power generation more clearly!

→ Lines 299-300: Which share do you mean? Which calibrated information do you mean? Our intended meaning was the share of each technological power generation method out of the total electricity generation. We have clarified this point.

Okay. The point of the energy and material balances is still open

As described herein, only the energy balance is considered; we have corrected this.

Comment 4

- Energy system model Enduse

o Clearly state the scope of the model!

o Line 314: Be more specific on energy end use and supply sectors! Are they the same as in the CGE? Are they a disaggregated representation of the sectors in the CGE?

Regarding the scope and granularity of sectors in AIM/Enduse, we added the following sentences.

“The model covers energy-related GHG emissions from both energy end-use and energy supply sectors. The end-use sectors are composed of industry, buildings and transportation sectors, and they are disaggregated into several subsectors with respect to types of products, buildings, and transportation mode based on the IEA energy balances.”

Okay, thanks. It is fair that you cannot describe the entire model here. You state that it is a “recursive, dynamic partial equilibrium model” (line 394). Could you please provide some more information here? Is it an optimization model (minimization of total system costs = partial equilibrium as outcome of perfectly competitive market) or an equilibrium model featuring multiple agents with distinct objective functions?

AIM/Enduse is a cost-minimising partial equilibrium model, as already stated.

Comment 5

The explanation of the communication between models is unsatisfactory and must be improved.

- Exchange from Enduse to CGE

o You state that both AIM/Enduse and AIM/Power decide on battery investments. Please clarify!

We stated that AIM/Power determines the capacity factors of the batteries, and that capacity is determined by AIM/Enduse. Then, CGE incorporates both data. We have clarified this point.

Okay. Can you please clarify for me what you mean by capacity factors of batteries? To the best of my knowledge, the capacity factor is the average hourly use of a technology over a year (or other timeframe), defined between zero and one. I know it for wind power or PV, for which it is given as exogenous data. For batteries a higher capacity factor may trigger higher investment, but higher investments may also trigger a lower/(higher) capacity factor. Please explain (to me). Thank you.

You are right. For batteries, the capacity factor is the total battery use (in hours) divided by 1 year. We have added this definition to the manuscript.

Comment 7

o Lines 384-386: Tentative guess of substitution elasticities being unity based on some Swedish econometric analysis is doubtful. Please more explanation here or argumentation why this is okay.

Thank you very much for raising this important point. We think that not asserting this tentative assumption is okay because the elasticity between monetary and physical units is uncertain. However, we believe that the impact of this uncertainty on the major findings of this paper is negligible, because the GDP losses estimated in this paper are low enough (3% at maximum) in the CGE stand-alone model. To test the uncertainty in the elasticity, we ran scenarios in which we varied the elasticity from 0.5 to 2.0. Our results indicate that the elasticity assumption affects the numbers, but the same qualitative conclusion holds. We have edited the sentences as below.

“According to the Swedish econometric analysis⁴⁸, elasticity between monetary and physical units of energy services can be assumed to be approximately 1.0. Furthermore, the GDP losses indicated in this study are relatively small, less than 3%, in the CGE stand-alone model, as shown in Figure 4a. Thus, we tentatively applied an elasticity value of 1.0. Meanwhile, we varied the elasticity from 0.5 to 2.0 and observed that the policy costs change slightly, but the qualitative conclusion still holds (Supplementary Figure 5).”

Thank you. The sensitivity and the figure are convincing. Could you please clarify what exactly you mean by the “elasticity between monetary and physical units”? I do not fully get it.

The elasticity accounts for the percentage change in physical output caused by a 1% change in monetary outputs; we have added text explaining this.

Comment 8

- Exchange from AIM/Enduse to AIM/Power and vice versa

o Please clarify in which model you determine which battery capacities!

Although we calculated the battery capacities in both AIM/Enduse and AIM/Power, we used the results from AIM/Enduse in our analysis. This is because the decision to invest in batteries for long-term fluctuations is part of capacity planning, as are investment decisions relating to pumped hydro power plants. Battery capacities for short-term fluctuations were determined by AIM/Power, which can model hourly electricity demand-supply balances.

Thank you. I think you can assume this and that this is uncritical. Can you please clarify how you differentiate between long-term and short-term batteries? Different investment costs for energy and power? Different technologies, i.e., li-ion vs redox flow or the like? Can you briefly comment on E/P ratios (endogenous/exogenous)?

Long-term batteries are assumed to mitigate VRE fluctuations between 1 and 24 hours.

To set the parameters for long-term batteries, we assume NaS batteries were used, as these are already in use in power generation systems. The E/P ratios for the long-term batteries were set as 7.2; thus, we do not differentiate the investment costs for energy and power. Short-term batteries are assumed to mitigate VRE fluctuations between a few minutes and 1 hour. The Li-ion battery was assumed to be used when setting parameters for short-term batteries. The E/P ratios were set to 1.

We have added text clarifying the above.

- Iteration and convergence

Second, as the differences would not improve, it would not be meaningful to iterate further. However, we did carry out further iterations, up to five, and present all of our results. The results confirm that most variables converge after two iterations. We did this for the default case as well as the no CCS and no nuclear cases for the baseline and mitigation scenarios, respectively. In the Supplementary Information, we show all comprehensive indicators by iteration for each scenario. In this response letter, we selected two examples, namely the 2050 baseline and mitigation scenarios, for default technological assumptions (see Supplementary Information for more scenarios and years).

Okay, looks quite convincing; but I am still a bit sceptical. Could you provide some intuition (for me and the reader) why convergence is achieved so fast? I also work with models that need to converge, but this usually requires more iterations.

There are two possible reasons for the rapid convergence. First, on the AIM/CGE side, the energy consumption is forced to be AIM/Enduse by endogenising parameters that are exogenous in the conventional CGE formula. Our understanding of other model coupling attempts would not change the variable and parameter relationships. Second, the major information provided by AIM/CGE to AIM/Enduse that changes the AIM/Enduse response is the energy service changes (output of sectors and total household consumption), but the difference from the previous iteration is less than 1%, which would not change the AIM/Enduse results in terms of carbon price or power generation.

We have added text to clarify this.

“We achieved relatively fast convergence compared with existing studies. There are two possible reasons for the rapid convergence. First, on the AIM/CGE side, the energy consumption is forced to be AIM/Enduse by endogenising parameters that are exogenous in the conventional CGE formula. Second, the major information provided by AIM/CGE to AIM/Enduse that changes the AIM/Enduse response is the energy service changes (output of sectors and total household consumption), but the difference from the previous iteration is less than 1%, which would not change AIM/Enduse results in terms of carbon price or power generation.”

Comment 9

About Table S4

Thank you. I am basically fine, but (i) this text here does not coincide with the text in the revised paper; (ii) it is also striking that excluding energy supply makes a larger difference, and (iii) I do not understand the color coding in Table S4; what do the uncolored rows indicate?

As discussed with respect to Figure 5, power generation accounts for 1~2% of GDP; if it was halved, in the figure would be 0.5~1% GDP. The uncoloured rows do not appear in Table 1. We have added text to the caption of Table S4.

Comment 10

How do you implement the temporal structure?

- Do you solve each model for 2010 until convergence and then proceed to 2015, 2020 and so on? Or do you solve Enduse for 2010 – 2050 and then hand over to CGE and iterate?

Thank you for pointing this out. We solved each model for the entire period and then passed the model outputs to the other models. We have added a description to this effect.

“The individual models were solved from 2010 to 2050, then the results from each were input to the other models.”

Thank you, I am fine with that. It remains (i) to defend this choice, and (ii) plausibly argue that results would be similar if you solved year 2010 until convergence, then year 2015, and so on. At least, it appears somewhat unusual to me.

If the models interact with each other on a yearly basis, convergence could occur much more rapidly, since the current approach results in gaps among the models throughout the study period that may widen, particularly during the latter period. Fortunately, however, we have already obtained good convergence with fewer iterations. We have added text to clarify this.

Comment 11

- Please be more specific on your CO2 budget. Is it one budget for the entire 2015-2050 period so that banking and borrowing, e.g., delayed decarbonization is possible?

Our approach is recursive dynamic and there is only an annual emissions constraint. We modified the corresponding text as follows.

“The CO2 emissions constraint is assumed for every simulation year of the AIM/Enduse model under the mitigation scenario.”

I am fine with that; however, you should (i) related this approach to the literature, and (ii) try to give an interpretation how allowing for banking and borrowing may impact results. Such distinction is not uncommon in the energy systems literature, to the best of my knowledge.

In this study, inter-temporal optimisation would not change the results because the carbon price trajectory is almost exponential. However, this might not be the case for other carbon constraints; we have added text regarding this.

“Within this study, the carbon price trajectory is almost exponential as a consequence. Therefore, even if we adopt the inter-temporal optimisation scheme, it would not markedly affect the results. However, this might not be the case for other carbon constraints.”

Comment 12

- Please be more specific on (social) discounting

There is no social discounting rate because our models are solved as recursive dynamic. To compute the annualized cost, we use the discount rate for individual energy use devices. We added the following detail to the supporting information and methods.

“As the models used in this study were solved as recursive dynamic, we did not consider discounting of the energy system costs. Nevertheless, the AIM/Enduse model annualizes the capital costs of energy technologies using a discount rate in the range 5–33% (Oshiro et al. 2016)¹. The sectoral discount rate represents 5% for power and industry, 10% for transportation, and 33% for other sectors.”

I am fine with that; however, please motivate (at least for me) why you can assume such high discount rates.

The discount rate for the individual sector (hereafter, individual discount rate) tends to be higher than the social discount rate. For example, the US EIA assumes 15% discounting for the commercial sector in their Annual Energy Outlook analysis. Additionally, in the case of Japan, since the previous study assumed a higher discount rate (33%) based on a questionnaire survey of households and companies in Japan (Hanaoka et al. 2008), we also set a 33% discount rate for the building sector.

In this study, the individual discount rate of the energy system model might not influence the main findings significantly, as the energy investment data fed into the economic model are not discounted by individual discount rates. We have added the following explanatory text.

“As the models used in this study were recursive dynamic, we did not consider discounting the energy system costs. Nevertheless, the AIM/Enduse model annualises the capital costs of energy technologies using a discount rate in the range 5–33% (Oshiro et al. 2016).¹¹ The sectoral discount rate is 5% for power and industry, 10% for transportation, and 33% for other sectors. These individual discount rates are only applied to simulate technology selection in the energy system model. Consequently, the energy investment data fed into the economic model are not discounted by these rates.”

References:

EIA. Assumptions to AEO2019 (2019). <https://www.eia.gov/outlooks/aeo/assumptions/>
Hanaoka et al. Global Greenhouse Gas Emissions Reduction Potentials and Mitigation Costs in 2020 (2008). CGER Report I081.
<http://www.cger.nies.go.jp/publications/report/i081/i081.pdf>

Comment 13

- Do you start with the existing Japanese power system? What are your assumption on capital depreciation?

Yes, we begin with the existing Japanese power system. For large power generation plants, we included data on each individual power plant, such as year constructed, capacity, efficiency, etc. We have added corresponding text, as below.

“In the AIM/Enduse model, the residual capacities of the existing power plants in operation today were calculated based on individual power plant information, such as year constructed, capacity of each plant, and expected lifetime. The total capacity was calculated based on the capacity of newly installed power plants, which was determined endogenously, as well as that of existing plants.”

Thank you. Some minor remarks: (i) please replace “today” by “in 2015” if your starting system data is from 2015; (ii) please provide all parameter assumptions, as long as they are not protected by any legal or commercial boundary, in a supplementary file for transparency and reproducibility. This would also comprise assumptions on fuel costs.

We have added the following table of power plant capacity data by construction year, as represented in the energy system model, to the Supplementary Information.

Table. Assumed power plant capacity by construction year

Construction year	-1970	1971	1976	1981	1986	1991	1996	2001	2006
		-75	-80	-85	-90	-95	-00	-05	-10
Coal w/o CCS	0.2	4.4	2.5	4.1	4.1	8.3	10.1	10.7	2.8
Gas w/o CCS	0.6	14.5	9.0	6.7	7.9	4.5	12.9	4.3	9.6
Oil w/o CCS	12.8	15.0	7.8	8.1	3.3	2.3	3.4	2.3	1.5
Hydro	16.1	1.0	1.0	1.1	0.6	0.4	0.2	0.2	0.1
Nuclear		12.1	5.7	3.4	6.5	11.1	4.7	2.2	3.2
Biomass w/o CCS									0.2
Geothermal	0.0	0.0	0.0	0.1	0.1	0.2	0.1		0.0
Solar PV						0.0	0.2	0.9	1.9

Wind, onshore 0.0 0.0 0.1 0.9 1.4

Units: G

The text has also been edited according to comment (i) as follows:

“In the AIM/Enduse model, the residual capacity of the existing power plants in operation in 2010 was calculated based on information for individual power plants, such as year of construction, capacity, and expected lifetime.”

The following table of assumed fuel costs has been also added to the Supplementary Information.

Table Assumed fuel prices

Fuel	Unit	2010	2020	2030	2040	2050
Price Coal	US\$2010/GJ	4.1	3.9	3.8	3.4	3.3
Price Gas	US\$2010/GJ	9.9	9.3	11.2	11.3	11.4
Price Oil	US\$2010/GJ	12.8	14.9	17.3	15.9	14.7

Comment 14

When I work on forward-looking analyses, technological change is a key driver for results
 - In both an energy-system and a macro-economic perspective, technological innovation and learning curves, i.e., decreasing costs are, of utmost importance. Think of solar PV or, currently, batteries

- Your technology cost assumptions are missing. Be transparent on all of them (appendix)
- Do you incorporate decreasing technology costs? If not, defend result!
- Sector coupling is key for (very) low-carbon energy systems (e.g. see the literature by M.Z. Jacobson or C. Breyer). Please make it clearer in your model description how you deal, for instance, with hydrogen. Is it somehow included in your sectors? This would also be a credible long-term storage.

Thank you very much. We added cost information with respect to power generation, which also includes decreasing technology costs, to the Supporting Information.

Thanks, this is valuable! Can you please also add the data sources? Can you please also add data on fuel costs and other components of variable costs if there are any?

Beyond that, I have some concerns about the cost data you use as they seem somewhat outdated. Capital costs are overnight investments, right? Basically, solar PV is at 3k dollar in 2020 and 1.8k dollar per kWp in 2050. This is high. Current (2019) costs are well below (<

1.5k), future costs presumably well below 1000 and I think that also WACC in Japan should be low. Check some random IRENA, NREL (for US) or Fraunhofer (for Germany) studies, e.g.,

<https://www.irena.org/->

/media/Files/IRENA/Agency/Publication/2018/Jan/IRENA_2017_Power_Costs_2018.pdf

<https://www.nrel.gov/docs/fy17osti/68925.pdf>

<https://www.ise.fraunhofer.de/content/dam/ise/en/documents/publications/studies/recent-facts-aboutphotovoltaics-in-germany.pdf>

Analogous for wind onshore, which is below 2k today. Offshore is a bit trickier due to idiosyncrasies and deep/shallow connection charges. Also costs for coal seem to be somewhat high for me. I understand that you need some source, preferably one consistent one for all/most cost data. At least provide some explanatory words. Ideally, update the cost data to a more recent source or run a sensitivity calculation.

The cost information is based on the latest survey by the Ministry of Energy, Trade and Industry of Japan (METI 2015), in which the values can be somewhat higher compared to other countries, as is well known. More detailed information is provided in a supplementary file from METI (2015), which can be obtained through the following link, although it is available only in Japanese.

https://www.enecho.meti.go.jp/committee/council/basic_policy_subcommittee/mitoshi/cost_wg/pdf/cost_wg_03.pdf

We use the METI assumption in this paper for two main reasons. First, as Japan's NDC is based on the cost assumption of METI, we use consistent assumptions to allow assessment in the context of national climate policy. Second, one of the main objectives of this study is to assess how much the GDP losses from the economic model can be changed by integrating energy system information. Therefore, we use the METI assumption, which is relatively conservative, to avoid underestimating the energy system costs and thus ensure robust results. We have added the following explanatory text to the Supplementary Materials.

“The cost information is based on METI data (2015),¹² as it is consistent with the assumptions in Japan's NDC. Note that the estimated mitigation cost may become much lower under more optimistic assumptions regarding future cost reductions, especially for renewable energies.”

Regarding hydrogen, we included it in the same manner as other energy carriers for energy demand sectors. On the production side, we also assumed that hydrogen production sectors will be introduced and provide input energy. However, within the Japanese context, our results indicate that hydrogen consumption will occupy a very small share of the final energy consumption (e.g. 0.02%) in the mitigation scenarios up to 2050.

Fine, thanks, this is a result. Nonetheless, could you please provide some information and an interpretation which role renewable electricity plays for heating and transportation; if not through hydrogen, then maybe through battery-electric vehicles, heat pumps or synthetic fuels?

In our analysis, electrification in energy end-use sectors is the major driver of renewable energy. As electricity is supplied by various low-carbon sources, such as nuclear and fossil with CCS, as well as renewables, it is difficult to provide information on how much renewable energy is consumed in each demand sector. We provide a figure showing the final energy demand by sector in the Supplementary Material. As shown in the figure, the share of electricity increases mainly in the building and transport sectors, due to the introduction of electric heat-pump technologies (e.g. air-conditioners and HP water heating) and battery-powered electric vehicles.

Figure. Final energy demand in the a) industry, b) buildings, and c) transportation sectors according to the mitigation scenario based on the energy system model.

Comment 15

- Line 69: what about seasonal storage, which should become relevant under 80% decarbonization?

We agree that seasonal storage is an essential element. The actual results are, however, that there are buffers to deal with seasonal fluctuations, such as fossil fuel CCS thermal plants, in the 80% reduction scenario, and thus, even if we consider battery storage for seasonal fluctuations, it would remain unused due to the cost competitiveness. The no CCS scenario also uses gas thermal plants to adjust for seasonal differences.

Thanks, this is plausible. Could you nevertheless please provide a short explicit clarification of this point in the main text (as in this comment here), potentially also in connection to flexible electrification (if there is any in the model)? Could also be footnote/endnote. I suppose that many readers may wonder about that.

Nature Communications does not allow footnotes; therefore, we have added text.

Comment 16

- Figure 1: the right column with arrows is hard to understand. Find a better format for representing exchange between models

We added numbers to the arrows so that we can specify what each arrow means.

Thanks, this helps! Remark: didn't you alter the power generation shares to another variable in (1)?

The power generation shares were delineated in (1) in the initial manuscript.

Comment 17

- Figure 2h: Please explain the extreme high CO₂ price in 2050. Model-ending effects?

This is due to the technological availability in the AIM/Enduse model. We added an explanation of this.

“The price of carbon in the mitigation scenario increased over time and reaches approximately \$1000/tCO₂ in 2050. This high carbon price is due to the technological availability in the AIM/Enduse model”

Thanks, but still not entirely clear to me. You mean the availability of certain technologies in Enduse? Which ones do you specifically mean?

Most of the technologies are used; other technologies with a high carbon price include insulation, etc.

Reviewer #2 (Remarks to the Author):

GENERAL COMMENTS

The authors have adequately addressed specific reviewer comments but have not fully addressed general comments, especially the limitations and contributions of this paper:

- The primary contribution of this paper remains model integration. One goal of model integration is to provide more realistic cost estimates, whether they be higher or lower than a stand-alone economic model. In this application, reported cost estimates are reduced with model integration.
- If the cost of greenhouse gas mitigation can be calculated from energy models alone, then why is an economy-wide economic model needed? The authors could (and should) make a stronger case for the benefits of using a CGE model.

Thank you very much. This framework allows us to determine the ultimate macroeconomic implications, which are more relevant for policy makers than the energy system cost. In this study, the macroeconomic and energy system costs ultimately showed similar tendencies, which was not expected before we began this research. The other advantage of using this framework is that the sectoral impacts of climate policy can be clarified. We have added text to this effect to the Abstract and Introduction.

- The new section starting at line 173, “Mechanism causing the differences in macroeconomic implications between the stand-alone and integrated models”, has created additional confusion, even though its intent was to clarify. I am not convinced that differences in primary factor productivity cause the difference in GDP cost between the stand-alone model and the integrated model. Implying causality is too strong. It seems plausible that changes in primary factor productivity and GDP losses are both caused by some other exogenous input to the models. The authors have decomposed total change in GDP into sector-specific components, which is useful, but I don’t see a causal relationship. I suggest backing off the claim of causality, but still report results in Figure 5.

Thank you very much. The reviewer’s understanding that the model inputs change productivity and ultimately affect GDP is correct. The parameter used in the stand-alone CGE model can easily be seen, and the AIM/Enduse model results, which are the inputs for the integrated model, are now shown. The GDP differences are not simple to understand; multiple factors are involved, as Figure 5 shows. Ultimately, the results rely on the assumptions of the stand-alone model parameters, and on the AIM/Enduse model outputs in the integrated model. For the former, we tried to make the assumptions as transparent as possible. In addition, the AIM/Enduse model assumptions and outputs are also now shown. As the reviewer suggested, we changed the text slightly to soften our stance regarding direct causality.

- What are next steps for model integration? An ambitious research agenda would replace the electricity sector in a CGE model with an electricity dispatch model. This would provide full simultaneity in solving model equations, and a well-defined interface between the dispatch model and other sectors of the economic model.

This remains an interesting paper, primarily because of linking an electricity dispatch model with a general equilibrium economic model, and the requirement to match hourly electricity demand and supply. The use of an electricity dispatch model, with hourly load shapes, is a big step beyond modeling the electricity market with load duration curves. This is especially important for the intermittent supply of wind and solar power.

Absolutely; that would be ambitious, but interesting. We have added text pertaining to that idea. We also think that certain other economic analyses, regarding employment implications, for example, are relevant to policy making; this is now discussed in the paper.

ADDITIONAL SPECIFIC COMMENTS

Please define and clarify “curtailment” in the paper. I think the authors are referring to excess wind or solar power that cannot be used or stored. Or it could mean electricity demand that cannot be met and load must be reduced, requiring some form of demand side management.

Thank you for this comment. The curtailment information shown in the figure is the discarded power energy divided by the total power generation for each technology. We have added an explanation of this to the figure caption.

The revised sentence at lines 48-49 is helpful: “Traditionally, CGE models tend to project policy costs that are higher than those of energy system models.” Please make the same modification to the fourth sentence in the Abstract.

This change has been made.

Reviewer #3 (Remarks to the Author):

Review of revision 1 "Energy Transformation Cost for the Japanese Mid-century Strategy: Energy System Feedback Effects in an Economic Model"

The authors have responded competently, and I think the paper has been significantly improved. I am a little in doubt regarding the degree of novelty of this work, but I recommend publishing the article provided that editorial adjustments are made.

I have one possible major issue: I notice that the results have changed (Figure 4, Figure 6, Table S3). Generally this could indicate that results are not robust, but I assume changes are due to the change in the method for presenting AIM/Enduse data as input to AIM/CGE. However, Figure 4a now indicate that policy cost is zero in 2020, as opposed to the previous version with more than 1% GDP loss in 2020. How do you explain such a change?

We apologise for this confusion. We noticed that an incorrect figure was shown for Figure 2 (the others are all correct). The carbon price data should have been from 2025 and this has now been corrected.

Minor comments:

Abstract line 15-16: This sentence should be clarified. Do you mean "GDP loss" as "one of the key elements"? Do you mean something like: "Our findings suggest that this type of integrated approach would provide new insights by providing improved estimates of GDP losses - one of the key elements for setting national climate policies."?

Yes, you are right; we have revised the text accordingly.

Line 39-41: The text suggests two models while you use three models – you may try to reformulate.

The power dispatch model is not usually classified as an integrated assessment model and therefore we do not explicitly discuss it here. However, we have added text mentioning the power dispatch model.

Line 59: "In IAMs, they are represented in some way" – you may want to reformulate the sentence.

This has been changed to “represented to some degree”.

Line 60 Suggestion: "Further literature list"  "An extended literature list".

This change has been made.

Line 97: "Application of this framework would be beneficial for Japan’s climate policies" – why? Because "necessary" policies would be easier to implement if new model estimates show lower GDP losses? The reader might think that an integrated model results in lower GDP loss, which touches upon the external validity problem. Please state why the application of the framework would be beneficial for Japan’s climate policies.

In Japan, it is still argued that the costs of implementing climate policies harm economic growth. However, if the cost information is updated based on this study, this argument should be resolved. We have modified the text accordingly.

Line 117: Sentence mixes past tense (increased) and present tense (reaches).

We have revised this sentence.

Figure 4: You may want to label the Figure 4a and 4b Y-axis differently, to improve readability (GDP loss versus equivalent variation). Perhaps you could connect the scatter points in figure 4c and 4d to simplify the interpretation by indicating the order of the points?

Figure 4 c) seems surprising, since the more constrained "no CCS" observations have lower GDP loss. The authors have commented on this earlier. It would still be interesting to check if household utility levels have decreased in the "no CCS" scenario, and to report equivalent variation.

We have renamed the y-axis labels and have connected the points in 4c and 4d.

Regarding the “no CCS” scenario results, these results are due to the methodology, in which we assumed the same carbon prices for the stand-alone and integrated models. Thus, the

emissions differ among models and the “no CCS” scenario has relatively higher emissions in the stand-alone model. However, we realise that this assumption would not be appropriate for the “no CCS” and “no Nuc” scenarios, which can show unacceptable differences in emissions from default technology scenarios. Therefore, we have decided to change the assumptions for the “no CCS” and “no Nuc” scenarios, where we imposed the same CO₂ emissions trajectories as in the default mitigation scenario. The results in Figure 4 have been updated and the default scenario now shows the lowest GDP losses compared with the “no CCS” and “no Nuc” scenarios. For your reference, the equivalent variation is also shown below; the trend is similar to the GDP loss rates.

Figure 5: I am not sure what the black arrows are indicating?

They indicate that Figure 5c is derived from the information in Figures 5a and 5b.

Figure 6: New results for the integrated model. The sign of the service sector is reversed in 2030 compared to the previous version, and the output change is relatively larger in 2050 than in the previous version.

The carbon price has changed, which underlies these results.

Line 346: "We executed the model for two iterations" – but you have run the model for 5 iterations (Figure S8-S13). Do you report the results after 2 or 5 iterations? Please clarify.

We report the results for the second iteration; we have clarified this point.

Line 364: "Dietary preferences" sounds strange – do you mean "consumer preferences"?

Yes, thank you very much; we have revised this.

Line 535: Please correct the sentence, and explain set i .

i is a set of variables and we have revised the sentence accordingly.

Supplementary: line 34: "We further run the iterations until 5" – please reformulate.

We have revised this text, as we ran five iterations.

Supplementary: line 35 in the supplementary information refers to the supplementary information. Needs to be corrected.

We have corrected this.

Supplementary: line 44: "causing the RMSE rate" – you have stated that you don't use RMSE, and have supplied the error rate formula you have used. Please correct.

We have revised this text.

Figure S11: The caption is identical on Figure S10 and S11. I assume S11 shows the mitigation scenario, not the baseline scenario. Please correct the caption.

You are correct; we have revised this text.

References)

1. van der Werf E. Production functions for climate policy modeling: An empirical analysis. *Energy Economics* 2008, **30**(6): 2964-2979.
2. Jacoby HD, Reilly JM, McFarland JR, Paltsev S. Technology and technical change in the MIT EPPA model. *Energy Economics* 2006, **28**(5-6): 610-631.
3. McFadden D. Econometric models of probabilistic choice. *Structural analysis of discrete data with econometric applications* 1981, **198272**.
4. Brenkert AL, Smith SJ, Kim SH, Pitcher HM. Model Documentation for the MiniCAM. PNNL; 2003.

5. de Vries BJM, van Vuuren DP, den Elzen MGJ, Janssen MA. The Targets IMage Energy Regional (TIMER) model Technical Documentation. Department of International Environmental Assessment National Institute of Public Health and the Environment (RIVM); 2001.
6. Clarke JF, Edmonds JA. Modelling energy technologies in a competitive market. *Energy Economics* 1993, **15**(2): 123-129.
7. Kainuma M MY, Morita T. *Climate policy assessment: Asia-Pacific integrated modeling*. Springer: Japan, 2003.
8. Oshiro K, Masui T. Diffusion of low emission vehicles and their impact on CO2 emission reduction in Japan. *Energy Policy* 2015, **81**: 215-225.
9. Nganou J-P. Estimation of the parameters of a linear expenditure system (LES) demand; 2005.
10. Bruinsma J. The resource outlook to 2050: by how much do land, water and crop yields need to increase by 2050?, Expert meeting on how to feed the world in 2050; 2010.
11. Oshiro K, Kainuma M, Masui T. Assessing decarbonization pathways and their implications for energy security policies in Japan. *Climate Policy* 2016, **16**(sup1): S63-S77.
12. METI. Report on analysis of generation costs, etc. for subcommittee on long-term energy supply and demand outlook. 2015.

Reviewers' comments:

Reviewer #1 (Remarks to the Author):

Dear authors,

Thank you very much for paying attention to my second round of comments. My open points are satisfactorily addressed. I think that especially the new sensitivity on substitution elasticities is highly valuable. After some editorial revisions (some points below), I recommend publishing the article.

Editorial comments

- L. 41-42: consider reformulating the new text in parentheses
- L. 57-58: [...] For policy makers, [...] would be more meaningful [...]
- L. 66: "power energy sources", term correct?
- L. 67: balance demand and supply
- L. 82: as energy supply and demand
- L. 92: compare; generally, tenses are mixed (e.g. "we confirm" in l. 88); consider using only present (or past) tense for describing what you do
- Ll. 99-100: across IAMs
- Fig. 1: You begin with enduse; appears strange if you denote it "CGE results 2"
- L. 121: carbon price increases over time
- Ll. 147-148: hereafter, GDP is accounted for by total final consumption
- Figure 4: consider having equally sized boxes
- L. 182: value-added, which is the GDP measure on the production side
- L. 183: because the capital and labour inputs change dynamically with population development and GDP growth, but [...]
- L. 187: [...] the latter on [...]
- L. 196: which is a model outcome
- L. 231: revise comma
- L. 239: which sectors
- L. 295: explain abbreviation AEEI upon first appearance
- L. 313: to this end, we
- L. 351: This notion – if I am not mistaken
- L. 357: on the AIM/CGE side
- l. 366: for future research
- ll. 371 – 374: consider revising the sentence
- l. 379: We executed five model iterations
- ll. 386 – 388: sentence unclear, consider revising
- l. 400: consumer preferences on diet
- l. 416: were similar
- l. 441: adopted an
- ll. 457 – 463: new sentence is present two times
- l. 469: as they are consistent

Reviewer #2 (Remarks to the Author):

I am satisfied with author responses to the latest round of review comments. However, I have a few suggestions for improvement, mostly for readability.

1. In line 121, replace "declines" with "increases."
2. Consider shortening the section "Mechanism causing the differences in macroeconomic implications between the stand-alone and integrated models." The first paragraph (lines 179-192)

is clear, but the technical detail that follows is difficult to understand. You could delete figures 5c and 5d, and the related discussion. I think most readers would skip this part.

3. The paper needs minor editing for English grammar.

Reviewer #3 (Remarks to the Author):

I am in favor of publishing the paper, but I have some comments that I consider important. Minor revisions are necessary.

Most important comments:

I am having trouble with figure 4.

1) The authors have changed the assumptions for the "no CCS" and "no nuclear" scenarios, they state that they now impose the same CO2 emissions trajectories as in the default mitigation scenario. I strongly agree with this change.

Figures 4c and 4d indicate that the more constrained "no CCS" scenario has lower GDP loss than the default mitigation scenario in the integrated model. However, figure S3 shows that the "no CCS" scenario has a different CO2 trajectory than the default mitigation and the "no-nuclear" scenarios. The "no CCS" has higher emissions in 2025, and a zero carbon price in 2025. This differs from the other scenarios. Is this one reason that the "non CCS" has lower GDP loss than default mitigation? Shouldn't the CO2-trajectory have been the same as for the other scenarios?

The authors have kindly supplied a chart in the rebuttal showing the equivalent variation from the three scenarios. This chart shows results that are as one would expect. However, this chart apparently shows equivalent variation from the CGE stand-alone model. I would like to see a chart showing equivalent variation from the integrated model (for the three scenarios "Default mitigation", "Mitigation_NoCCS" and "Mitigation_NoNuc").

2) The authors have connected the points in 4c and 4d, which I applaud. However, it seems to me that Figure 4c and 4d are not showing the yearly observations in the same order. This would mean that there are errors in the figures.

Example a: The red triangles for the default mitigation scenario do show increasing GDP losses in the integrated model in Figure 4c, while figure 4d shows a sudden decrease in the observation for year 2045 (I guess). One explanation could be that 2040 and 2045 have changed places in figure 4d.

Example b: The green "no nuclear" circles show a sudden drop (in 2035) in figure 4d for the integrated model. Such a drop is not found in 4c.

Example c: The blue "no CCS" diamonds show a decrease from 2045 to 2050 in the integrated model in figure 3c, while they show an increase in figure 4d from 2045 to 2050.

Figures 4c and 4d seem to share the same y-axis, in which case the development in figure 4d seems not compatible with figure 4c for any of the scenarios. The figure text says that 4d shows GDP losses in the CGE stand-alone model and additional energy system costs in AIM/Enduse. This is in conflict with the y-axis label saying "Integrated model (%/year)". If the y-axis is "CGE stand-alone (%/year) instead, then the green "no nuclear" and the red "Default" series are still not compatible with figure 3c, since they have sudden decreases (measured by figure 4c y-axis) that are not visible in figure 3c (measured by the x-axis).

Minor comments

line 23: The word "counties" should be "countries".

line 121: The word "declines" seems wrong. The price of carbon in the mitigation scenario increases over time.

line 170, Figure 4 b: There is no indication for equivalent variation result for CGE stand alone in 2050. Is this point missing?

line 201: Percentage points are different than percentages. Do you rather mean percentage points  "These eventually account for 1.3 percentage points of the total value-added (GDP)"?
Suggestion: "These eventually account for 1.3 percentage points of the change in total value-added (GDP)".

line 214: Do you rather mean percentage points  "... accounting for 0.4 percentage points of GDP"?

line 443-446: Text from lines 436-439 is repeated.

line 461-463: Text from lines 457-459 is repeated.

line 469: Correct the sentence: ".., as it is are consistent with ..."

Line 476: There is an underscore in the middle of the line.

Line 498-503: This part is rather detailed, and introduces new and unexplained abbreviations such as NaS and E/P ratios. According to line 496; battery storage for seasonal fluctuations would remain unused. Then it seems irrelevant to go into further details about the batteries. Consider to drop this part.

Line 535-537: The authors state that subsectors are aggregated so that the granularity of the sectors is in agreement, but still "... iron, chemical ... and non-ferrous metals are exempted from the sector aggregation". This could be perceived as if information regarding these important GHG sectors are not exchanged, which I assume is not correct(?). Please clarify the consequences of exempting these sectors from sector aggregation. Is information regarding these sectors not exchanged? Or are these sectors apparent in both models - in which case sector aggregation is not necessary?

Supplementary material

Line 54-55: I suggest to avoid using "not" many times in one sentence.

Line 60: "the mathematical formula"  should be plural: "the mathematical formulas"?

Table S7: The table is hard to read due to lack of borders between sectors.

Table S12: The table is hard to read due to lack of borders between household consumption good categories.

Figure S2 (line 49): Letters indicating panels a-h are not visible.

Figure S3 (line 56): Letters indicating panels a-h are not visible.

Figure S3 (line 56): As mentioned above, the CO2 trajectory (panel d) is different from the other

scenarios, which seems to contradict the revision comment stating: "Therefore, we have decided to change the assumptions for the "no CCS" and "no Nuc" scenarios, where we imposed the same CO2 emissions trajectories as in the default mitigation scenario." Also the CO2 price in panel h is positive from 2030, instead of 2025 as in the other scenarios. I suspect that such a difference between the scenarios is unintentional?

To all reviewers

Thank you very much for giving us comments. We have made revisions on the text according to the comments and figure 4.

The responses to each comment are shown below. The blue text is our responses to the reviewer comments.

Reviewers' comments:

Reviewer #1 (Remarks to the Author):

- L. 41-42: consider reformulating the new text in parentheses
- L. 57-58: [...] For policy makers, [...] would be more meaningful [...]
- L. 67: balance demand and supply
- L. 82: as energy supply and demand
- L. 92: compare; generally, tenses are mixed (e.g. “we confirm” in l. 88); consider using only present (or past) tense for describing what you do
- Ll. 99-100: across IAMs
- L. 121: carbon price increases over time
- Ll. 147-148: hereafter, GDP is accounted for by total final consumption
- L. 182: value-added, which is the GDP measure on the production side
- L. 183: because the capital and labour inputs change dynamically with population development and GDP growth, but [...]
- L. 187: [...] the latter on [...]
- L. 196: which is a model outcome
- L. 231: revise comma
- L. 239: which sectors
- L. 295: explain abbreviation AEEI upon first appearance
- L. 313: to this end, we
- L. 357: on the AIM/CGE side
- l. 366: for future research
- l. 379: We executed five model iterations
- l. 400: consumer preferences on diet
- l. 416: were similar
- l. 441: adopted an
- ll. 457 ? 463: new sentence is present two times
- l. 469: as they are consistent

Thank you very much. We have modified the above points accordingly.

- L. 66: “power energy sources”, term correct?

Thank you. We revised as “energy sources”.

- L. 351: This notion ? if I am not mistaken

Correct and we have modified it.

- Fig. 1: You begin with enduse; appears strange if you denote it “CGE results 2”
We needed to distinguish all CGE runs and numbering from the stand-alone model is convenient, which is also related to the Supplementary Information Table S3. Thus, we decided to keep them as it is.

- Figure 4: consider having equally sized boxes

Thank you. We have equalized box sizes.

- ll. 371 ? 374: consider revising the sentence

- ll. 386 ? 388: sentence unclear, consider revising

We modified the sentences accordingly.

Reviewer #2 (Remarks to the Author):

I am satisfied with author responses to the latest round of review comments. However, I have a few suggestions for improvement, mostly for readability.

1. In line 121, replace “declines” with “increases.”

Thank you. We have modified as suggested.

2. Consider shortening the section “Mechanism causing the differences in macroeconomic implications between the stand-alone and integrated models.” The first paragraph (lines 179-192) is clear, but the technical detail that follows is difficult to understand. You could delete figures 5c and 5d, and the related discussion. I think most readers would skip this part.

Thank you very much. We completely agree to the comment which is really useful to consider the readership of Nature Communications. Meanwhile, we needed to add this part according to the reviewer 1’s strong suggestion and we, thus, decide to keep them as it has been.

3. The paper needs minor editing for English grammar.

Thank you. We have read through once again and edited if it is needed.

Reviewer #3 (Remarks to the Author):

I am in favor of publishing the paper, but I have some comments that I consider important. Minor revisions are necessary.

Most important comments:

I am having trouble with figure 4.

1) The authors have changed the assumptions for the "no CCS" and "no nuclear" scenarios, they state that they now impose the same CO2 emissions trajectories as in the default mitigation scenario. I strongly agree with this change.

Figures 4c and 4d indicate that the more constrained "no CCS" scenario has lower GDP loss than the default mitigation scenario in the integrated model. However, figure S3 shows that the "no CCS" scenario has a different CO2 trajectory than the default mitigation and the "no-nuclear" scenarios. The "no CCS" has higher emissions in 2025, and a zero carbon price in 2025. This differs from the other scenarios. Is this one reason that the "non CCS" has lower GDP loss than default mitigation? Shouldn't the CO2-trajectory have been the same as for the other scenarios?

The authors have kindly supplied a chart in the rebuttal showing the equivalent variation from the three scenarios. This chart shows results that are as one would expect. However, this chart apparently shows equivalent variation from the CGE stand-alone model. I would like to see a chart showing equivalent variation from the integrated model (for the three scenarios "Default mitigation", "Mitigation_NoCCS" and "Mitigation_NoNuc").

First of all, we apology our poor explanation in the previous response and should have been more precise. The assumption that has the same emissions trajectory is applied only for the stand-alone model. The requested figure that include equivalent variation from the integrated model (for the three scenarios "Default mitigation", "Mitigation_NoCCS" and "Mitigation_NoNuc") is shown below. The nuclear phase-out obviously influences on the near-term macroeconomic impacts, but it has benefit of the early penetration of the renewable energies in the latter period. NoCCS scenario basically exhibits higher costs than the default scenario.

Regarding the emissions in 2025, we have found that the NoCCS scenario in AIM/Enduse run had a mistake and we modified it. Now, the carbon price occurs in 2030 and tangible emissions reduction also occurs from 2025, but the overall carbon prices are very similar to the previous results which we think that we would not need to change any qualitative discussion in the main text.

2) The authors have connected the points in 4c and 4d, which I applaud. However, it seems to me that Figure 4c and 4d are not showing the yearly observations in the same order. This would mean that there are errors in the figures.

Example a: The red triangles for the default mitigation scenario do show increasing GDP losses in the integrated model in Figure 4c, while figure 4d shows a sudden decrease in the observation for year 2045 (I guess). One explanation could be that 2040 and 2045 have changed places in figure 4d.

Example b: The green "no nuclear" circles show a sudden drop (in 2035) in figure 4d for the integrated model. Such a drop is not found in 4c.

Example c: The blue "no CCS" diamonds show a decrease from 2045 to 2050 in the integrated model in figure 3c, while they show an increase in figure 4d from 2045 to 2050.

Figures 4c and 4d seem to share the same y-axis, in which case the development in figure 4d seems not compatible with figure 4c for any of the scenarios. The figure text says that 4d shows GDP losses in the CGE stand-alone model and additional energy system costs in AIM/Enduse. This is in conflict with the y-axis label saying "Integrated model (%/year)". If the y-axis is "CGE stand-alone (%/year) instead, then the green "no nuclear" and the red "Default" series are still not compatible with figure 3c, since they have sudden decreases (measured by figure 4c y-axis) that are not visible in figure 3c (measured by the x-axis).

Thank you very much. We really appreciate the comment that notifies us that there were errors in figures as well as in the captions. The reviewer is totally correct. More technically, we used “geom_line” function to draw these lines in R software, but it should had been “geom_path” function. We have revised the figures as shown below. The sudden drops are still seen in “Default” and “NoNuclear” scenarios, but they represent the behavior of the AIM/Enduse model. We believe that panel c and d are now compatible. The caption of figure d was also incorrect and it should be “integrated model” rather than “CGE stand-alone”. We have revised accordingly.

Minor comments

line 23: The word "counties" should be "countries".

Thank you. We have modified as suggested.

line 121: The word "declines" seems wrong. The price of carbon in the mitigation scenario increases over time.

Thank you. We have modified as suggested.

line 170, Figure 4 b: There is no indication for equivalent variation result for CGE stand alone in 2050. Is this point missing?

We cut the maximum value at 3.5% and indeed we missed 2050. Now we revise the figure as shown above.

line 201: Percentage points are different than percentages. Do you rather mean percentage points  "These eventually account for 1.3 percentage points of the total value-added (GDP)"? Suggestion: "These eventually account for 1.3 percentage points of the change in total value-added (GDP)".

Thank you. We have modified as suggested.

line 214: Do you rather mean percentage points  "... accounting for 0.4 percentage points of GDP"?

Thank you. We have modified as suggested.

line 443-446: Text from lines 436-439 is repeated.

line 461-463: Text from lines 457-459 is repeated.

Thank you very much. Both have been removed.

line 469: Correct the sentence: "..., as it is are consistent with ..."

Line 476: There is an underscore in the middle of the line.

Thank you. We have modified both.

Line 498-503: This part is rather detailed, and introduces new and unexplained abbreviations such as NaS and E/P ratios. According to line 496; battery storage for seasonal fluctuations

would remain unused. Then it seems irrelevant to go into further details about the batteries. Consider to drop this part.

Thank you. We have decided to drop this part.

Line 535-537: The authors state that subsectors are aggregated so that the granularity of the sectors is in agreement, but still "... iron, chemical ... and non-ferrous metals are exempted from the sector aggregation". This could be perceived as if information regarding these important GHG sectors are not exchanged, which I assume is not correct(?). Please clarify the consequences of exempting these sectors from sector aggregation. Is information regarding these sectors not exchanged? Or are these sectors apparent in both models - in which case sector aggregation is not necessary?

They are partly disaggregated in the energy system model, but the detailed sector's information is not exchanged. Instead, the total industry energy consumption by fuel types are exchanged. The analysis of the results show that service sector is a major element that changes macro-economic implications in this study.

Supplementary material

Line 54-55: I suggest to avoid using "not" many times in one sentence.

Line 60: "the mathematical formula"  should be plural: "the mathematical formulas"?

Table S7: The table is hard to read due to lack of borders between sectors.

Table S12: The table is hard to read due to lack of borders between household consumption good categories.

Figure S2 (line 49): Letters indicating panels a-h are not visible.

Figure S3 (line 56): Letters indicating panels a-h are not visible.

Thank you very much. We have all modified the above six points accordingly.

Figure S3 (line 56): As mentioned above, the CO₂ trajectory (panel d) is different from the other scenarios, which seems to contradict the revision comment stating: "Therefore, we have decided to change the assumptions for the "no CCS" and "no Nuc" scenarios, where we imposed the same CO₂ emissions trajectories as in the default mitigation scenario." Also the CO₂ price in panel h is positive from 2030, instead of 2025 as in the other scenarios. I suspect that such a difference between the scenarios is unintentional?

The CO₂ trajectories are only harmonized for the CGE stand-alone model because the energy consumption and supply in the CGE model within the integrated framework are almost determined by the energy system model and this representation prevents us to make identical emissions.

Regarding the carbon price, we have found that the NoCCS scenario in AIM/Enduse run had a mistake and we revised it. Now the carbon price occurs in 2030, but the overall carbon prices are quite similar to the previous results which we think that we would not need to change any qualitative discussion in the main text. The revised figure is shown as below.

REVIEWERS' COMMENTS:

Reviewer #3 (Remarks to the Author):

The authors have responded competently to the comments. I suggest that the article may be published.